# The Complexity of Sparse Tensor PCA

**Davin Choo**[*]
Department of Computer Science
National University of Singapore
davin@u.nus.edu

**Tommaso d'Orsi**
Department of Computer Science
ETH Zürich
tommaso.dorsi@inf.ethz.ch

## Abstract

We study the problem of sparse tensor principal component analysis: given a tensor $\boldsymbol{Y} = \boldsymbol{W} + \lambda x^{\otimes p}$ with $\boldsymbol{W} \in \otimes^p \mathbb{R}^n$ having i.i.d. Gaussian entries, the goal is to recover the $k$-sparse unit vector $x \in \mathbb{R}^n$. The model captures both sparse PCA (in its Wigner form) and tensor PCA. For the highly sparse regime of $k \leq \sqrt{n}$, we present a family of algorithms that smoothly interpolates between a simple polynomial-time algorithm and the exponential-time exhaustive search algorithm. For any $1 \leq t \leq k$, our algorithms recovers the sparse vector for signal-to-noise ratio $\lambda \geq \widetilde{\mathcal{O}}(\sqrt{t} \cdot (k/t)^{p/2})$ in time $\widetilde{\mathcal{O}}(n^{p+t})$, capturing the state-of-the-art guarantees for the matrix settings (in both the polynomial-time and sub-exponential time regimes). Our results naturally extend to the case of $r$ distinct $k$-sparse signals with disjoint supports, with guarantees that are independent of the number of spikes. Even in the restricted case of sparse PCA, known algorithms only recover the sparse vectors for $\lambda \geq \widetilde{\mathcal{O}}(k \cdot r)$ while our algorithms require $\lambda \geq \widetilde{\mathcal{O}}(k)$. Finally, by analyzing the low-degree likelihood ratio, we complement these algorithmic results with rigorous evidence illustrating the trade-offs between signal-to-noise ratio and running time. This lower bound captures the known lower bounds for both sparse PCA and tensor PCA. In this general model, we observe a more intricate three-way trade-off between the number of samples $n$, the sparsity $k$, and the tensor power $p$.

## 1 Introduction

Sparse tensor principal component analysis is a statistical primitive generalizing both sparse PCA[2] and tensor PCA[3]. We are given multi-linear measurements in the form of a tensor

$$\boldsymbol{Y} = \boldsymbol{W} + \lambda x^{\otimes p} \in \otimes^p \mathbb{R}^n \tag{SSTM}$$

for a Gaussian noise tensor $\boldsymbol{W} \in \otimes^p \mathbb{R}^n$ containing i.i.d. $N(0,1)$ entries[4] and signal-to-noise ratio $\lambda > 0$. Our goal is to estimate the "structured" unit vector $x \in \mathbb{R}^n$. The structure we enforce on $x$ is sparsity: $|\mathrm{supp}(x)| \leq k$. The model can be extended to include multiple spikes in a natural way: $\boldsymbol{Y} = \boldsymbol{W} + \sum_{q=1}^r \lambda_q x_{(q)}^{\otimes p}$, and even general order-$p$ tensors: $\boldsymbol{Y} = \boldsymbol{W} + \sum_{q=1}^r \lambda_q \mathcal{X}_{(q)}$ for $\mathcal{X}_{(q)} = x_{(q,1)} \otimes \cdots \otimes x_{(q,p)} \in \otimes^p \mathbb{R}^n$. In this introduction, we focus on the simplest single spike setting of SSTM.

It is easy to see that sparse PCA corresponds to the setting with tensor order $p = 2$. On the other hand, tensor PCA is captured by effectively removing the sparsity constraint: $|\mathrm{supp}(x)| \leq n$. In

---

[*]Part of the work was done while the author was in ETH Zürich.

[2]Often in the literature, the terms sparse PCA and spiked covariance model refer to the sparse spiked *Wishart* model. However, here we consider the sparse spiked *Wigner* matrix model.

[3]Tensor PCA is also known as the spiked *Wigner* tensor model, or simply the spiked tensor model.

[4]Throughout the paper, we will write random variables in boldface.

recent years, two parallel lines of work focused respectively on sparse PCA [JL09, AW08, BR13a, DM16, HKP$^+$17, DKWB19, HSV20, dKNS20] and tensor PCA [MR14, HSS15, MSS16, HKP$^+$17, KWB19, AMMN19], however no result captures both settings. The appeal of the *sparse spiked tensor model* (henceforth SSTM) is that it allows one to study the computational and statistical aspects of these other fundamental statistical primitives in a unified framework, understanding the computational phenomena at play from a more general perspective.

In this work, we investigate SSTM from both algorithmic and computational hardness perspectives. Our algorithm improves over known tensor algorithms whenever the signal vector is highly sparse. We also present a lower bound against low-degree polynomials which extends the known lower bounds for both sparse PCA and tensor PCA, leading to a more intricate understanding of how all three parameters ($n$, $k$ and $p$) interact.

## 1.1 Related work

Disregarding computational efficiency, it is easy to see that optimal statistical guarantees can be achieved with a simple exhaustive search (corresponding to the maximum likelihood estimator): find a $k$-sparse unit vector maximizing $\langle \boldsymbol{Y}, x^{\otimes p} \rangle$. This algorithm returns a $k$-sparse unit vector $\widehat{x}$ achieving constant squared correlation[5] with the signal $x$ as soon as $\lambda \gtrsim \sqrt{k \cdot \log(np/k)}$. That is, whenever $\lambda \gtrsim \max_{\|x\|=1, \|x\|_0=k} \langle \boldsymbol{W}, x^{\otimes p} \rangle$. Unfortunately, this approach runs in time exponential in $k$ and takes super-polynomial time when $p \lesssim k$.[6] *As such, we assume $p \leq k$ from now on.*

Taking into account computational aspects, the picture changes. A good starting point to draw intuition for SSTM is the literature on sparse PCA and tensor PCA. We briefly outline some known results here. To simplify the discussion, we hide absolute constant multiplicative factors using $\mathcal{O}(\cdot)$, $\Omega(\cdot)$, $\lesssim$, and $\gtrsim$, and hide multiplicative factors logarithmic in $n$ using $\widetilde{\mathcal{O}}(\cdot)$.

### 1.1.1 Sparse PCA (Wigner noise)

Sparse PCA with Wigner noise exhibits a sharp phase transition in the top eigenvalue of $\boldsymbol{Y}$ for $\lambda \geq \sqrt{n}$ [FP07]. In this strong signal regime, the top eigenvector[7] $v$ of $\boldsymbol{Y}$ correlates[8] with $x$ with high probability, thus the following spectral method achieves the same guarantees as the exhaustive search suggested above: compute a leading eigenvector of $\boldsymbol{Y}$ and restrict it to the top $k$ largest entries in absolute value. Conversely, when $\lambda < \sqrt{n}$, the top eigenvector of $\boldsymbol{Y}$ does not correlate with the signal $x$. In this weak signal regime, [JL09] proposed a simple algorithm known as diagonal thresholding: compute the top eigenvector of the principal submatrix defined by the $k$ largest diagonal entries of $\boldsymbol{Y}$. This algorithm recovers the sparse direction when $\lambda \gtrsim \widetilde{\mathcal{O}}(k)$, thus requiring almost an additional $\sqrt{k}$ factor when compared to inefficient algorithms. More refined polynomial-time algorithms (low-degree polynomials [dKNS20], covariance thresholding [DM16] and the basic SDP relaxation [dGJL07, dKNS20]) only improve over diagonal thresholding by a logarithmic factor in the regime $n^{1-o(1)} \lesssim k^2 \lesssim n$. Interestingly, multiple results suggest that this *information-computation gap* is inherent to the sparse PCA problem [BR13a, BR13b, DKWB19, dKNS20]. Subexponential time algorithms and lower bounds have also been shown. For instance, [DKWB19, HSV20] presented smooth trade-offs between signal strength and running time.[9]

---

[5]One could also aim to find a unit vector with correlation approaching one or, in the restricted setting of $x \in \{0, \pm 1/\sqrt{k}\}$, aim to recover the support of $x$. At the coarseness of our discussion here, these goals could be considered mostly equivalent.

[6]Note that the problem input is of size $n^p$. So when $p \gtrsim k$, exhaustive search takes $n^{\mathcal{O}(p)}$ time which is polynomial in $n^p$. Thus, the interesting parameter regimes occur when $p \lesssim k$.

[7]By "top eigenvector" or "leading eigenvector", we mean the eigenvector corresponding to "largest (in absolute value) eigenvalue".

[8]More precisely, the vector consisting of the $k$ largest (in absolute value) entries of $v$.

[9]Both works studied the single spike matrix setting. [HSV20] only considers the Wishart noise model and thus its guarantees cannot be compared to ours. [DKWB19] studied both the Wishart and Wigner noise models. In the Wishart noise model setting, both [HSV20] and [DKWB19] observe the same tradeoff between running time and signal-to-noise ratio. In the Wigner noise model setting, our algorithm and the algorithm of [DKWB19] offer the same smooth-trade off between running time and signal strength, up to universal constants.

### 1.1.2 Tensor PCA

In tensor settings, computing $\max_{\|x\|=1}\langle \boldsymbol{Y}, x^{\otimes p}\rangle$ is NP-hard already for $p = 3$ [HL13]. For even tensor powers $p$, one can unfold the tensor $\boldsymbol{Y}$ into a $n^{p/2}$-by-$n^{p/2}$ matrix and solve for the top eigenvector [MR14]. However, this approach is sub-optimal for odd tensor powers. For general tensor powers $p$, a successful strategy to tackle tensor PCA has been the use of semidefinite programming [HSS15, BGL16, HSS19]. Spectral algorithms inspired by the insight of these convex relaxations have also been successfully applied to the problem [SS17]. These methods succeed in recovering the single-spike $x$ when $\lambda \gtrsim \widetilde{\mathcal{O}}\left(n^{p/4}\right)$, thus exhibiting a large gap when compared to exhaustive search algorithms. Matching lower bounds have been shown for constant degrees in the Sum-of-Squares hierarchy [BGL16, HKP$^{+}$17] and through average case reductions [BB20].

### 1.1.3 Sparsity-exploiting algorithms and tensor algorithms

It is natural to ask how do the characteristics of sparse PCA and tensor PCA extend to the more general setting of SSTM. In particular, there are two main observations to be made.

The first observation concerns the sharp computational transition that we see for $k \lesssim \sqrt{n}$ in sparse PCA. In these highly sparse settings, the top eigenvector of $\boldsymbol{Y}$ does not correlate with the signal $x$ and so algorithms primarily based on spectral methods fail to recover it. Indeed, the best known guarantees are achieved through algorithms that crucially exploit the sparsity of the hidden signal. These algorithms require the signal strength to satisfy $\lambda \geq \widetilde{O}(\sqrt{k})$, with only logarithmic dependency on the ambient dimension. To exemplify this to an extreme, notice how the following algorithm can recover the support of $xx^{\mathsf{T}}$ with the same guarantees as diagonal thresholding, essentially disregarding the matrix structure of the data: zero all but the $k^2$ largest (in absolute value) entries of $\boldsymbol{Y}$. A natural question to ask is whether a similar phenomenon may happen for higher order tensors. In the highly sparse settings where $k \lesssim \sqrt{n}$, *can we obtain better algorithms exploiting the sparsity of the hidden vector?* Recently, a partial answer appeared in [LZ20] with a polynomial time algorithm recovering the hidden signal for $\lambda \geq \widetilde{O}(p \cdot k^{p/2})$, albeit with suboptimal dependency on the tensor order $p$.

The second observation concerns the computational-statistical gap in the spiked tensor model. As $p$ grows, the gap between efficient algorithms and exhaustive search widens with the polynomial time algorithms requiring signal strength $\lambda \geq \widetilde{O}\left(n^{p/4}\right)$ while exhaustive search succeeds when $\lambda \geq \widetilde{O}(\sqrt{n})$ [MR14]. The question here is: *how strong is the dependency on $p$ for efficient algorithms in sparse signal settings?*

In this work, we investigate these questions in the high order tensors regime $p \in \omega(1)$. We present a family of algorithms with a smooth trade-off between running time and signal-to-noise ratio. Even restricting to polynomial-time settings, our algorithms improve over previous results. Furthermore, through the lens of low-degree polynomials, we provide rigorous evidence of an *exponential gap* in the tensor order $p$ between algorithms and lower bounds.

**Remark.** *The planted sparse densest sub-hypergraph model [CPMB19, BCPS20, CPSB20] is closely related to SSTM. We discuss this model in Appendix E.*

## 1.2 Results

### 1.2.1 Single spike setting

Consider first the restricted, but representative, case where the planted signal is a $(k, A)$-sparse unit vector with $k$ non-zero entries having magnitudes in the range $\left[\frac{1}{A\sqrt{k}}, \frac{A}{\sqrt{k}}\right]$ for some constant $A \geq 1$. We say that the signal is *flat* when $A = 1$ and *approximately flat* when $A \geq 1$.

Our first result is a limited brute force algorithm – informally, an algorithm that smoothly interpolates between some brute force approach and some "simple" polynomial time algorithm – that *exactly* recovers the signal support of the planted signal[10].

**Theorem 1** (Algorithm for single spike sparse tensor PCA, Informal)**.** *Let $A \geq 1$ be a constant. Consider the observation tensor*

$$\boldsymbol{Y} = \boldsymbol{W} + \lambda x^{\otimes p}$$

---

[10]A similar algorithm was analyzed by [DKWB19] for the special case of $p = 2$ and $r = 1$.

*where the additive noise tensor $\boldsymbol{W} \in \otimes^p \mathbb{R}^n$ contains i.i.d. $N(0,1)$ entries and the signal $x \in \mathbb{R}^n$ is a $(k, A)$-sparse unit vector with signal strength $\lambda > 0$. Let $1 \leq t \leq k$ be an integer. Suppose that*

$$\lambda \gtrsim \sqrt{t \left( \frac{2A^2 k}{t} \right)^p \ln n} \,.$$

*Then, there exists an algorithm that runs in $\mathcal{O}(pn^{p+t})$ time and, with probability 0.99, outputs the support of $x$.*

Let's first consider Theorem 1 in its simplest setting where $A = 1$ and $t$ is a fixed constant. For $k \lesssim \sqrt{n}$, the theorem succeed when $\lambda \geq \widetilde{\mathcal{O}}(k^{p/2})$, thus improving over the guarantees of known tensor PCA methods which require $\lambda \geq \widetilde{\mathcal{O}}(n^{p/4})$. In addition, since support recovery is *exact*, one can obtain a good estimate[11] of the planted signal by running any known tensor PCA algorithm on the subtensor corresponding to its support. Indeed, the resulting subtensor will be of significantly smaller dimension and the requirement needed on the signal strength by single-spike tensor PCA algorithms are weaker than the requirement we impose on $\lambda$ (see Remark 9 for details). As a result, our algorithm recovers the guarantees of diagonal thresholding in the matrix ($p = 2$) setting. Our polynomial-time algorithm also improves over the result of [LZ20], which required $\lambda \gtrsim \sqrt{pk^p \log n}$, by removing the polynomial dependency of the tensor order $p$ in the signal strength $\lambda$.[12]

Consider now the limited brute force parameter $t$. From the introductory exposition, we know that one can obtain a statistically optimal algorithm by performing a brute force search over the space of $k$-sparse flat vectors in $\mathbb{R}^n$. The *limited brute force* algorithm is a natural extension that takes into account computational constraints by searching over the smaller set of $t$-sparse flat vectors, for $1 \leq t \leq k$, to maximize $\langle \boldsymbol{Y}, u^{\otimes p} \rangle$. The parametric nature of the algorithm captures both the brute force search algorithm (when $t = k$) and the idea of diagonal thresholding (when $t = 1$ and $p = 2$). As long as $t \leq k$, using a larger $t$ represents a direct trade-off between running time and the signal-to-noise ratio. Extending the result to approximately flat vectors, the dependency on $A$ in the term $\left( 2A^2 \right)^p$ can be removed by increasing the computational budget to some value $t' \geq 2A^2 t$.

### 1.2.2 Multiple spikes

**Theorem 2** (Algorithm for multi-spike sparse tensor PCA, Informal). *Let $A \geq 1$ be a constant. Consider the observation tensor*

$$\boldsymbol{Y} = \boldsymbol{W} + \sum_{q=1}^{r} \lambda_q x_{(q)}^{\otimes p}$$

*where the additive noise tensor $\boldsymbol{W} \in \otimes^p \mathbb{R}^n$ contains i.i.d. $N(0,1)$ entries and the signals $x_{(1)}, \ldots, x_{(r)} \in \mathbb{R}^n$ are $(k, A)$-sparse unit vectors with disjoint supports and corresponding signal strengths $\lambda_1 \geq \ldots \geq \lambda_r > 0$. Let $1 \leq t \leq k$ be an integer and $0 < \epsilon \leq 1/2$. Suppose that*

$$\lambda_r \gtrsim \frac{1}{\epsilon} \cdot \sqrt{t \left( \frac{2A^2 k}{t} \right)^p \ln n} \quad \text{and} \quad \lambda_r \gtrsim A^{2p} \cdot (2\epsilon)^{p-1} \cdot \lambda_1 \,.$$

*Then, there exists an algorithm that runs in $\mathcal{O}(rpn^{p+t})$ time and, with probability 0.99, outputs the individual signal supports of $x_{(\pi(1))}, \ldots, x_{(\pi(r))}$ for some unknown bijection $\pi : [r] \to [r]$.*

Theorem 2 requires two assumptions on the signals: (1) signals have disjoint support; (2) there is a bounded signal strength gap of $\lambda_r \gtrsim A^{2p} \cdot (2\epsilon)^{p-1} \cdot \lambda_1$. In the context of sparse PCA, algorithms that recover multiple spikes (e.g. [JL09, DM16]) only require the sparse vectors to be orthogonal. However, their guarantees are of the form $\lambda_r \geq \widetilde{\mathcal{O}} \left( \left| \bigcup_{q \in [r]} \operatorname{supp}\left( x_{(q)} \right) \right| \right)$. That is, when the $r$ signals have disjoint supports, they require the smallest signal to satisfy $\lambda_r \geq \widetilde{\mathcal{O}}(k \cdot r)$. In comparison, already for constant $t$, Theorem 2 successfully recovers the supports when $\lambda_r \geq \widetilde{\mathcal{O}}(k)$,

---

[11]Recovery is up to a global sign flip since $\langle u, v \rangle^p = \langle u, -v \rangle^p$ for even tensor powers $p$.

[12]The result of [LZ20] extends to the settings where $\boldsymbol{Y} = \boldsymbol{W} + \lambda \mathcal{X}$ for an approximately flat tensor $\mathcal{X} \in \otimes^p \mathbb{R}^n$. Both Theorem 1 and Theorem 2 can also be extended to these settings (see Appendix B.2).

thus removing the dependency on the number of signals and improving the bound by a $1/r$ factor[13]. Meanwhile, the bounded signal strength gap assumption is a common identifiability assumption (e.g. see [CMW13, DM16]). We remark that Theorem 2 provides a tradeoff between this signal strength gap assumption and the signal strengths: we can recover the supports with a smaller gap if the signal strengths are increased proportionally – increasing $\lambda_r$ by a multiplicative factor $\alpha$ enables the algorithm to succeed with gap that is smaller by a multiplicative factor of $1/\alpha$. As an immediate consequence, we also obtain a tradeoff between gap assumption and running time: every time we double $t$ (while ensuring $1 \leq t \leq k$), $\lambda_r$ increases by a factor of $(1/\sqrt{2})^{p-1}$ and thus the algorithm can succeed with a smaller gap. Finally, as in the single spike case, the exact support recovery allow us to obtain good estimate of each signal by running known tensor PCA algorithms.

**Remark** We remark that these results can be extended to the general tensor settings

$$\boldsymbol{Y} = \boldsymbol{W} + \sum_{q=1}^{r} \lambda_q \mathcal{X}_{(q)}$$

where for $q \in [r]$, $\mathcal{X}_{(q)} = x_{(q,1)} \otimes \cdots \otimes x_{(q,p)} \in \otimes^p \mathbb{R}^n$ in a natural way. See Appendix B.2.

### 1.2.3 An exponential gap between lower bounds and algorithms

SSTM generalizes both sparse PCA and tensor PCA. Hence, a tight hardness result for the model is interesting as it may combine and generalize the known bounds for these special cases. Here, we give a lower bound for the restricted computational model captured by *low-degree polynomials*. Originally developed in the context of the sum of squares hierarchy, this computational model appears to accurately predict the current best-known guarantees for problems such as sparse PCA, tensor PCA, community detection, and planted clique (e.g. see [HS17, HKP+17, Hop18, BHK+19, DKWB19, KWB19, dKNS20]).

**Theorem 3** (Lower bound for low-degree polynomials, Informal). *Let $1 \leq D \leq 2n/p$ and $\nu$ be the distribution of $\boldsymbol{Z} \in \otimes^p \mathbb{R}^n$ with i.i.d. entries from $N(0,1)$. Then, there exists a distribution $\mu$ over tensors $\boldsymbol{Y} \in \otimes^p \mathbb{R}^n$ of the form*

$$\boldsymbol{Y} = \boldsymbol{W} + \lambda \boldsymbol{x}^{\otimes p}$$

*where $\boldsymbol{W} \in \otimes^p \mathbb{R}^n$ is a noise tensor with i.i.d. $N(0,1)$ entries, the marginal distribution of $\boldsymbol{x}$ is supported on vectors with entries $\left\{ \pm 1/\sqrt{k}, 0 \right\}^n$, and $\boldsymbol{x}$ and $\boldsymbol{W}$ are distributionally independent, such that whenever*

$$\lambda \lesssim \frac{\sqrt{D}}{2^p} \min \left\{ \left( \frac{n}{pD} \right)^{p/4}, \left( \frac{k}{pD} \left( 1 + \left| \ln \left( \frac{npD}{ek^2} \right) \right| \right) \right)^{p/2} \right\},$$

*$\mu$ is indistinguishable[14] from $\nu$ with respect to all polynomials of degree at most $D$.*

Theorem 3 states that for certain values of $\lambda$, low-degree polynomials cannot be used to distinguish between the distribution of $\boldsymbol{Y}$ and $\boldsymbol{W}$ as typical values of low-degree polynomials are the same (up to a vanishing difference) under both distributions. The theorem captures known results in both sparse and tensor PCA settings. When $p = 2$, our bound reduces to $\lambda \lesssim \min \left\{ \sqrt{n}, \frac{k}{\sqrt{D}} \left( 1 + \left| \ln \left( \frac{2nD}{ek^2} \right) \right| \right) \right\}$, matching known low-degree bounds of [DKWB19] in the sparse PCA setting. Meanwhile, in the tensor PCA settings ($p \geq 2$, $k = n$), Theorem 3 implies a bound of the form $\lambda \lesssim \sqrt{D} \left( \frac{n}{pD} \right)^{p/4}$, thus recovering the results of [KWB19].

---

[13]It is an intriguing question whether an improvement of $1/r$ can be achieved in the more general settings of orthogonal spikes. Our approach relies on the signals having disjoint support and we expect it to *not* be generalizable to orthogonal signals. This can be noticed in the simplest settings with brute-force parameter $t = 1$ and $p = 2$ where the criteria of Algorithm 3 for finding an entry of a signal vector is to look at the diagonal entries of the data matrix. In this case, the algorithm may be fooled since the largest diagonal entry can depend on more than one spike. Nevertheless, we are unaware of any fundamental barrier suggesting that such guarantees are computationally hard to achieve.

[14]In the sense that for any low-degree polynomial $p(\boldsymbol{Y})$ we have $\frac{\mathbb{E}_\mu p(\boldsymbol{Y}) - \mathbb{E}_\nu p(\boldsymbol{Y})}{\sqrt{\mathbb{V} p(\boldsymbol{Y})}} \in o(1)$. See Appendix A.4.2.

For constant power $p$ and $k \lesssim \sqrt{n}$, our lower bound suggests that no estimator captured by polynomials of degree $D \lesssim t \log n$ can improve over our algorithmic guarantees by more than a logarithmic factor. However, for $p \in \omega(1)$, an exponential gap appears between the bounds of Theorem 3 and state-of-the-art algorithms (both in the sparse settings as well as in the dense settings).[15] As a concrete example, let us consider the setting where $p = n^{0.1} < k$. The polynomial time algorithm of Theorem 1 requires $\lambda \geq \tilde{\mathcal{O}}(k^{p/2})$ while according to Theorem 3 it may be enough to have $\lambda \geq \tilde{\mathcal{O}} \left( k/n^{0.1} \right)^{p/2}$. Similarly, for $k \gtrsim \sqrt{np}$, known tensor algorithms recovers the signal for $\lambda \geq \tilde{\mathcal{O}}(n^{p/4})$ while our lower bound only rules out algorithms for $\lambda \leq \tilde{\mathcal{O}} \left( n^{0.9 \cdot p/4} \right)$.

Surprisingly, for the distinguishing problem considered in Theorem 3, these bounds appear to be tight. For a wide range of parameters (in both the dense and sparse settings) there exists polynomial time algorithms that can distinguish the distributions $\nu$ and $\mu$ right at the threshold considered in Theorem 3 (see Appendix C). It remains a fascinating open question whether sharper recovering algorithms can be designed or stronger lower bounds are required.

Finally, we would like to highlight that this non-trivial dependency on $p$ is a purely computational phenomenon as it does not appear in information-theoretic bounds (see Appendix D).

**Remark**  Note that Theorem 3 is *not* in itself a lower bound for the recovery problem. However, any algorithm which obtains a good estimation of the signal vector $x$ for signal strength $\lambda \geq \sqrt{k \log n}$ can be used to design a probabilistic algorithm which solve the distinguishing problem for signal strength $\mathcal{O}_p(\lambda)$. Let us elaborate. Consider an algorithm that given $\boldsymbol{Y} = \boldsymbol{W} + \lambda x^{\otimes p}$ outputs a vector $\hat{x}$ such that $|\langle \hat{x}, x \rangle| \geq 0.9$. With high probability, $\max_{|z|_2=1, |z|_0=k} |\langle \boldsymbol{W}, z^{\otimes p} \rangle| \leq \widetilde{\mathcal{O}}(\sqrt{k})$ and thus $|\langle \boldsymbol{Y}, \hat{x}^{\otimes p} \rangle| \geq \lambda \cdot (0.9)^p - \widetilde{\mathcal{O}}(\sqrt{k})$. Therefore, one can solve the distinguishing problem as follows: output "planted" if $|\langle \boldsymbol{Y}, \hat{x}^{\otimes p} \rangle| \gtrsim \sqrt{k \log n}$ and "null" otherwise.

## 1.3   Notation and outline of paper

We write random variables in boldface and the set $\{1, \ldots, n\}$ as $[n]$. We hide absolute constant multiplicative factors and multiplicative factors logarithmic in $n$ using standard notations: $\mathcal{O}(\cdot), \Omega(\cdot), \lesssim, \gtrsim,$ and $\widetilde{\mathcal{O}}(\cdot)$. We denote by $e_1, \ldots, e_n \in \mathbb{R}^n$ the standard basis vectors. For $x \in \mathbb{R}^n$, we use $\operatorname{supp}(x) \subseteq [n]$ to denote the set of support coordinates. We say that $x$ is a $(k, A)$-*sparse vector* if $k \in [n]$, constant $A \geq 1$, $|\operatorname{supp}(x)| = k$, and $\frac{1}{A\sqrt{k}} \leq |x_\ell| \leq \frac{A}{\sqrt{k}}$ for $\ell \in \operatorname{supp}(x)$. When $A = 1$, we say that $x$ is a $k$-*sparse flat vector* and may omit the parameter $A$. For general $A \geq 1$, we say that $x$ is *approximately flat*. For an integer $t \geq 1$, we define $U_t = \left\{ u \in \left\{ -\frac{1}{\sqrt{t}}, 0, \frac{1}{\sqrt{t}} \right\}^n : |\operatorname{supp}(u)| = t \right\}$ as the set of $t$-sparse flat vectors. For a tensor $T \in \otimes^p \mathbb{R}^n$ and a vector $u \in \mathbb{R}^n$, their inner product is defined as $\langle T, u^{\otimes p} \rangle = \sum\limits_{i_1, \ldots, i_p \in [n]} T_{i_1, \ldots, i_p} u_{i_1} \ldots u_{i_p}$.

The rest of the paper is organized as follows: In Section 2, we introduce the main ideas behind Theorem 1 and Theorem 2. In Section 3, we flesh out some concrete unresolved research questions. Appendix A contains preliminary notions. We formally prove Theorem 1 and Theorem 2 in Appendix B. The lower bound Theorem 3 is given in Appendix C. We present an information theoretic bound in Appendix D. Appendix E discusses the planted sparse densest sub-hypergraph model. Finally, Appendix F contains any deferred technical proofs required throughout the paper.

## 2   Recovering signal supports via limited brute force searches

We describe here the main ideas behind our limited brute force algorithm. We consider the model

**Model 4** (Sparse spiked tensor model)**.** *For $A \geq 1, r \geq 1, k \leq n$ we observe a tensor of the form*

$$\boldsymbol{Y} = \boldsymbol{W} + \sum_{q=1}^{r} \lambda_q x_{(q)}^{\otimes p} \in \otimes^p \mathbb{R}^n$$

---

[15]In particular, in the sparse settings $k \leq \sqrt{np}$, the $p^{-p/2}$ factor could not be seen in the restricted case of sparse PCA (as this factor is a constant when $p = 2$).

*where $\boldsymbol{W} \in \otimes^p \mathbb{R}^n$ is a noise tensor with i.i.d. $N(0, 1)$ entries, $\lambda_1 \geq \ldots \geq \lambda_r > 0$ are the signal strengths, and $x_{(1)}, \ldots, x_{(r)}$ are $k$-sparse flat unit length signal vectors.*

We first look at the simplest setting of a single flat signal (i.e. $A = 1$ and $r = 1$). Then, we explain how to extend the analysis to multiple flat signals. For a cleaner discussion, we assume here that all the non-zero entries of the sparse vector $x$ and vectors in the set $U_t$ have positive sign. Our techniques also easily extend to approximately flat vectors (where $A \geq 1$) and general signal tensors $x_{(1)} \otimes \cdots \otimes x_{(p)} \in \otimes^p \mathbb{R}^n$. We provide details for these extensions in Appendix B.

## 2.1 Single flat signal

**Limited-brute force**   As already mentioned in the introduction, a brute force search over $U_k$ for the vector maximizing $\langle \boldsymbol{Y}, u^{\otimes p} \rangle$ returns the signal vector $x$ (up to a global sign flip) with high probability whenever $\lambda \gtrsim \sqrt{k \log n}$. This algorithm provides provably optimal guarantees but requires exponential time (see Appendix D for an information-theoretic lower bound). The idea of a *limited brute force search* is to search over a smaller set $U_t$ ($1 \leq t \leq k$) instead, and use the maximizer $\boldsymbol{v}_*$ to determine the signal support $\mathrm{supp}\,(x)$. The hope is that for a sufficiently large signal-to-noise ratio, this $t$-sparse vector $\boldsymbol{v}_*$ will still be non-trivially correlated with the hidden vector $x$. Indeed as $t$ grows, the requirement on $\lambda$ decreases towards the information-theoretic bound, at the expense of increased running time.

As a concrete example, consider the matrix settings ($p = 2$). It is easy to generalize the classic diagonal thresholding algorithm ([JL09]) into a limited brute-force algorithm. Recall that diagonal thresholding identifies the support of $x$ by picking the indices of the largest $k$ diagonal entries of $\boldsymbol{Y}$. In other words, the algorithm simply computes $\langle \boldsymbol{Y}, e_i^{\otimes 2} \rangle$ for all $i \in [n]$ and returns the largest $k$ indices. From this perspective, the algorithm can be naturally extended to $t > 1$ by computing the $\binom{k}{t}$ vectors $u \in U_t$ maximizing $\langle \boldsymbol{Y}, u^{\otimes 2} \rangle$ and reconstructing the signal from them. For $t = k$, the algorithm corresponds to exhaustive search.

With this intuition in mind, we now introduce our family of algorithms, heavily inspired[16] by [DKWB19]. We first apply a preprocessing step to obtain two independent copies of the data.

---

**Algorithm 1** Preprocessing

**Input**: $\boldsymbol{Y}$.
Sample a Gaussian tensor $\boldsymbol{Z} \in \otimes^p \mathbb{R}^n$ where each entry is an i.i.d. standard Gaussian $N(0, 1)$.
**Return** two independent copies $\boldsymbol{Y}^{(1)}$ and $\boldsymbol{Y}^{(2)}$ of $\boldsymbol{Y}$ as follows:

$$\boldsymbol{Y}^{(1)} = \frac{1}{\sqrt{2}} (\boldsymbol{Y} + \boldsymbol{Z}) \quad \text{and} \quad \boldsymbol{Y}^{(2)} = \frac{1}{\sqrt{2}} (\boldsymbol{Y} - \boldsymbol{Z})$$

---

Algorithm 1 effectively creates two independent copies of the observation tensor $\boldsymbol{Y}$. To handle the noise variance, the signal-to-noise ratio is only decreased by the constant factor $1/\sqrt{2}$. For simplicity, we will ignore this constant factor in the remainder of the section.

---

**Algorithm 2** Single spike limited brute force

**Input**: $k, t$ and $\boldsymbol{Y}^{(1)}, \boldsymbol{Y}^{(2)}$ obtained from Algorithm 1.
Compute $\boldsymbol{v}_* := \mathrm{argmax}_{u \in U_t} \langle \boldsymbol{Y}^{(1)}, u^{\otimes p} \rangle$.
Compute the vector $\boldsymbol{\alpha} \in \mathbb{R}^n$ with entries $\boldsymbol{\alpha}_\ell := \langle \boldsymbol{Y}^{(2)}, \boldsymbol{v}_*^{\otimes p-1} \otimes e_\ell \rangle$ for every $\ell \in [n]$.
**Return** the indices of the largest $k$ entries of $\boldsymbol{\alpha}$.

---

[16]The algorithm in [DKWB19] is a specialization of ours (with comparable guarantees) in the simplest setting of $p = 2$ and a single spike. However, looking at [DKWB19], it is a priori unclear how to generalize the result to the settings of our interest. This is especially true in the tensor settings ($p \geq 3$) with multiple spikes, where signals may interfere with each other.

The signal support recovery process outlined in Algorithm 2 has two phases. In the first phase, we search over $U_t$ to obtain a vector $\boldsymbol{v}_*$ that is correlated with the signal $x$. In the second phase, we use $\boldsymbol{v}_*$ to identify $\mathrm{supp}(x)$. The correctness of the algorithm follows from these two claims:

   (i) The $t$-sparse maximizer $\boldsymbol{v}_*$ shares a large fraction of its support coordinates with signal $x$.

   (ii) The $k$ largest entries of $\boldsymbol{\alpha}$ belong to the support $\mathrm{supp}(x)$ of signal $x$.

Crucial to our analysis is the following standard concentration bound on Gaussian tensors. We directly use Lemma 5 in our exposition here, and formally prove a more general form in Appendix F.1.

**Lemma 5.** *Let $p \leq n$, $t > 0$ be an integer, and $\boldsymbol{W} \in \otimes^p \mathbb{R}^n$ be a tensor with i.i.d. $N(0,1)$ entries. Then, with high probability, for any $u \in U_t$,*

$$\langle \boldsymbol{W}, u^{\otimes p} \rangle \lesssim \sqrt{t \log n}\,.$$

For some constant $0 < \epsilon \leq 1/2$, suppose that

$$\lambda \gtrsim \frac{1}{\epsilon \cdot (1-\epsilon)^{p-1}} \cdot \sqrt{t \left(\frac{k}{t}\right)^p \log n}\,. \tag{1}$$

For any $u \in U_t$ with support $\mathrm{supp}(u) \subseteq \mathrm{supp}(x)$, we have

$$\langle \boldsymbol{Y}^{(1)}, u^{\otimes p} \rangle = \lambda \langle x, u \rangle^p + \langle \boldsymbol{W}^{(1)}, u^{\otimes p} \rangle \geq \lambda \cdot \left(\frac{t}{k}\right)^{\frac{p}{2}} - \mathcal{O}\left(\sqrt{t \log n}\right)\,.$$

On the other hand, any $u \in U_t$ with support satisfying $|\mathrm{supp}(u) \cap \mathrm{supp}(x)| \leq (1-\epsilon) \cdot t$ has small correlation with $\boldsymbol{Y}^{(1)}$ in the sense that

$$\langle \boldsymbol{Y}^{(1)}, u^{\otimes p} \rangle = \lambda \langle x, u \rangle^p + \langle \boldsymbol{W}^{(1)}, u^{\otimes p} \rangle \leq \lambda \cdot (1-\epsilon)^p \cdot \left(\frac{t}{k}\right)^{\frac{p}{2}} + \mathcal{O}\left(\sqrt{t \log n}\right)\,.$$

By Eq. (1), with high probability, $\boldsymbol{v}_*$ will have at least a fraction $(1-\epsilon)$ of its support contained in $\mathrm{supp}(x)$, yielding the first claim. Observe that $\boldsymbol{v}_*$ does not completely overlap with $x$. A priori, this might seem to be an issue. However, it turns out that we can still use $\boldsymbol{v}_*$ to exactly reconstruct the support of $x$. Indeed, for all $\ell \in \mathrm{supp}(x)$,

$$\begin{aligned}
\boldsymbol{\alpha}_\ell &= \lambda \cdot x_\ell \cdot \langle x, \boldsymbol{v}_* \rangle^{p-1} + \langle \boldsymbol{W}^{(2)}, \boldsymbol{v}_*^{\otimes p-1} \otimes e_\ell \rangle \\
&\geq \lambda \cdot \frac{(1-\epsilon)^{p-1}}{\sqrt{k}} \cdot \left(\frac{t}{k}\right)^{\frac{p-1}{2}} + \langle \boldsymbol{W}^{(2)}, \boldsymbol{v}_*^{\otimes p-1} \otimes e_\ell \rangle \\
&\gtrsim \frac{1}{\epsilon} \cdot \sqrt{\log n} + \langle \boldsymbol{W}^{(2)}, \boldsymbol{v}_*^{\otimes p-1} \otimes e_\ell \rangle\,.
\end{aligned}$$

Now, by independence of $\boldsymbol{W}^{(2)}$ and $\boldsymbol{v}_*$, $\langle \boldsymbol{W}^{(2)}, \boldsymbol{v}_*^{\otimes p-1} \otimes e_\ell \rangle$ behaves like a standard Gaussian. Thus, with high probability, $\left|\langle \boldsymbol{W}^{(2)}, \boldsymbol{v}_*^{\otimes p-1} \otimes e_\ell \rangle\right| \lesssim \sqrt{\log n}$ and $\boldsymbol{\alpha}_\ell \gtrsim \sqrt{\log n}$. Conversely, if $\ell$ is *not* in the support of the signal, then $\boldsymbol{\alpha}_\ell \lesssim \sqrt{\log n}$. So, the vector $\boldsymbol{\alpha}$ acts as indicator of the support of $x$!

**Remark 6.** In its simplest form of $t = 1$, Algorithm 2 does *not* exploit the tensor structure of the data: it performs entry-wise search for the largest (in magnitude) over a subset of $\boldsymbol{Y}$. However, this is no longer true as $t$ grows. For $t = k$, the algorithm computes the $k$-sparse flat unit vector $u$ maximizing $\langle \boldsymbol{Y}^{(1)}, u^{\otimes p} \rangle$.

## 2.2 Multiple flat signals with disjoint signal supports

Consider now the setting with $r > 1$ spikes. Recall that we assumed the vectors $x_{(1)}, \ldots, x_{(r)}$ to have non-intersecting supports. We also assumed that for any $q, q' \in [r]$ and some fixed scalar $0 \leq \kappa \leq 1$, if $\lambda_q \geq \lambda_{q'}$, then $\lambda_{q'} \geq \kappa \cdot \lambda_q$. We remark that we may *not* recover the signal supports in a known

order, but we are guaranteed to recover *all of them* exactly. For simplicity of discussion, let us assume here that we recover the vector $x_{(i)}$ at iteration $i$.

The idea to recover the $r$ spikes is essentially to run Algorithm 2 $r$ times. At first, we compute the $t$-sparse vector $\boldsymbol{v}_*$ by maximizing the product $\langle \boldsymbol{Y}^{(1)}, \boldsymbol{v}_*^{\otimes p} \rangle$. Then, using $\boldsymbol{v}_*$, we compute the vector $\boldsymbol{\alpha}$ to obtain a set $\mathcal{I}_1 \subseteq [n]$. With high probability, we will have $\mathcal{I}_1 = \mathrm{supp}\left(x_{(1)}\right)$ and so we will exactly recover the support of $x_{(1)}$. In the second iteration of the loop, we repeat the same procedure with the additional constraint of searching only over the $n - k$ dimensional subset of $U_t$ containing vectors with disjoint support from $\mathcal{I}_1$. Similarly, at iteration $i$, we search over the subset of $U_t$ containing vectors with disjoint support from $\bigcup_{1 \leq j < i} \mathcal{I}_j$. As before, we first preprocess the data to create two independent copies $\boldsymbol{Y}^{(1)}$ and $\boldsymbol{Y}^{(2)}$. Concretely:

---

**Algorithm 3** Multi-spike limited brute force

---

**Input**: $k, t, r$ and $\boldsymbol{Y}^{(1)}, \boldsymbol{Y}^{(2)}$ obtained from Algorithm 1.
**Repeat** for $i = 1$ to $r$:

Compute $\boldsymbol{v}_* := \mathrm{argmax}_{u \in U_t} \langle \boldsymbol{Y}^{(1)}, u^{\otimes p} \rangle$ subject to $\mathrm{supp}\left(\boldsymbol{v}_*\right) \cap \left( \bigcup_{1 \leq j < i} \mathcal{I}_j \right) = \emptyset$.

Compute the vector $\boldsymbol{\alpha} \in \mathbb{R}^n$ with entries $\boldsymbol{\alpha}_\ell := \langle \boldsymbol{Y}^{(2)}, \boldsymbol{v}_*^{\otimes p-1} \otimes e_\ell \rangle$ for every $\ell \in [n]$.
Let $\mathcal{I}_i$ be the set of indices of the largest $k$ entries of $\boldsymbol{\alpha}$.

**Return** $\mathcal{I}_1, \ldots, \mathcal{I}_r$.

---

The proof structure is similar to that of Algorithm 2 and essentially amounts to showing that the claims (i) and (ii) described in Section 2.1 hold in each iteration.

Let $\lambda_{\min} = \min_{q \in [r]} \lambda_q$ and $\lambda_{\max} = \max_{q \in [r]} \lambda_q$. For some $0 < \epsilon \leq 1/2$, let $\kappa \gtrsim \left( \frac{\epsilon}{1-\epsilon} \right)^{p-1}$ such that $\lambda_{\min} \geq \kappa \cdot \lambda_{\max}$. Suppose that

$$\lambda_{\min} \gtrsim \frac{1}{\epsilon \cdot (1-\epsilon)^p} \cdot \sqrt{t \left( \frac{k}{t} \right)^p \log n} \quad \text{and} \quad \lambda_{\min} \gtrsim \left( \frac{\epsilon}{1-\epsilon} \right)^{p-1} \cdot \lambda_{\max}. \tag{2}$$

Consider an arbitrary iteration $i$ and suppose that we exactly recovered the support of one signal in each of the previous iterations. Without loss of generality, assume that $\lambda_{\max}$ is the largest signal strength among the yet to be recovered signals, and let $x_{(\max)}$ be one such corresponding signal.

For $u \in U_t$ satisfying $\mathrm{supp}(u) \subseteq \mathrm{supp}(x_{(\max)})$, we have

$$\langle \boldsymbol{Y}^{(1)}, u^{\otimes p} \rangle = \lambda_{\max} \langle x_{(\max)}, u \rangle^p + \langle \boldsymbol{W}^{(1)}, u^{\otimes p} \rangle \geq \lambda_{\max} \cdot \left( \frac{t}{k} \right)^{\frac{p}{2}} - \mathcal{O}\left( \sqrt{t \log n} \right).$$

On the other hand, for any $u \in U_t$ such that $\left| \mathrm{supp}(u) \cap \mathrm{supp}\left(x_{(q)}\right) \right| \leq (1 - \epsilon) \cdot t$ for all $q \in [r]$,

$$\langle \boldsymbol{Y}^{(1)}, u^{\otimes p} \rangle = \sum_{q \in [r]} \lambda_q \langle x_{(q)}, u \rangle^p + \langle \boldsymbol{W}^{(1)}, u^{\otimes p} \rangle$$

$$\leq \lambda_{\max} \cdot \left( \frac{t}{k} \right)^{\frac{p}{2}} \cdot ((1 - \epsilon)^p + \epsilon^p) + \mathcal{O}\left( \sqrt{t \log n} \right)$$

$$\leq \lambda_{\max} \cdot \left( \frac{t}{k} \right)^{\frac{p}{2}} \cdot (1 - \epsilon)^{p-1} + \mathcal{O}\left( \sqrt{t \log n} \right).$$

Thus, as in Section 2.1, it follows that $\boldsymbol{v}_*$ satisfies $\left| \mathrm{supp}\left(\boldsymbol{v}_*\right) \cap \mathrm{supp}\left(x_{(i)}\right) \right| \geq (1 - \epsilon) \cdot t$ for some signal $x_{(i)}$. Note that $x_{(i)}$ may not be $x_{(\max)}$. Even though $\boldsymbol{v}_*$ does not exactly overlap with any of the signal vectors, we will not accumulate an error at each iteration. This is because, analogous to the single spike setting, we can exactly identify the support of a signal through $\boldsymbol{\alpha}$. For any

$\ell \in \mathrm{supp}\left(x_{(i)}\right)$, it holds that $\boldsymbol{\alpha}_\ell \gtrsim \sqrt{\log n}$ as before because $\left|\langle \boldsymbol{W}^{(2)}, \boldsymbol{v}_*^{\otimes p-1} \otimes e_\ell \rangle\right| \lesssim \sqrt{\log n}$. Conversely, since signal supports are disjoint, we see that for $\ell \notin \mathrm{supp}\left(x_{(i)}\right)$,

$$
\begin{aligned}
\boldsymbol{\alpha}_\ell &= \sum_{q \in [r]} \lambda_q \cdot x_{(q),\ell} \cdot \langle x_{(q)}, \boldsymbol{v}_* \rangle^{p-1} + \langle \boldsymbol{W}^{(2)}, \boldsymbol{v}_*^{\otimes p-1} \otimes e_\ell \rangle \\
&\leq \lambda_{\max} \cdot \frac{\epsilon^{p-1}}{\sqrt{k}} \cdot \left(\frac{t}{k}\right)^{\frac{p-1}{2}} + \mathcal{O}\left(\sqrt{\log n}\right) \\
&\leq \frac{\lambda_{\min}}{\kappa} \cdot \frac{\epsilon^{p-1}}{\sqrt{k}} \cdot \left(\frac{t}{k}\right)^{\frac{p-1}{2}} + \mathcal{O}\left(\sqrt{\log n}\right) \\
&\lesssim \sqrt{\log n}\,.
\end{aligned}
$$

So, once again, $\boldsymbol{\alpha}$ exactly identifies the support of $x_{(i)}$ with high probability.

**Remark 7** (On the strength of the assumption on $\kappa$). As already briefly discussed in Section 1.2, the algorithm provides a three-way trade-off between signal gap $\kappa$, signal-to-noise ratio $\lambda$ and running time. By appropriately choosing the constant $\epsilon > 0$, the algorithm can tolerate different values of $\kappa$. Indeed, the above analysis holds as long as $\kappa \gtrsim \left(\frac{\epsilon}{1-\epsilon}\right)^{p-1}$. This suggests two ways in which we can loosen the requirement $\lambda_{\min} \geq \kappa \cdot \lambda_{\max}$ and still successfully recover the signals through Algorithm 3. One is increase the running time, so that we can decrease $\epsilon$ without increasing the signal-to-noise ratio $\lambda_{\min}$. The other is to decrease $\epsilon$ and increase the value of $\lambda_{\min}$ accordingly.

**Remark 8** (On independent copies of $\boldsymbol{Y}$). To clarify why it suffices to have 2 independent copies of $\boldsymbol{Y}$ even for multiple iterations, observe that at each iteration $i$, the choice of the set $\mathcal{I}_i$ depends only on the vector $\boldsymbol{v}_*$ with high probability. Consider the following thought experiment where we are given a fresh copy $\boldsymbol{Y}^{(i)}$ of $\boldsymbol{Y}$ in the second phase of each iteration $i$ of the algorithm (while still using only a single copy $\boldsymbol{Y}^{(1)}$ for *all* the first phases). Even with fresh randomness, the result is the same as Algorithm 3 with high probability because at each iteration the choice of maximizer $\boldsymbol{v}_*$ causes the same output.

**Remark 9** (Reconstructing the signals from their supports). After recovering individual signal supports, one can reconstruct signals using known tensor PCA algorithms (e.g. [MR14, HSS15]) on the subtensor defined by each recovered support. The signal strength required for this new subproblem is weaker and is satisfied by our recovery assumptions. For instance, by concatenating our algorithm with [HSS15, Theorem 7.1], one obtains vectors $\widehat{x}_{(1)}, \ldots, \widehat{x}_{(r)}$ such that $\left|\langle \widehat{x}_{(i)}, x_{(i)} \rangle\right| \geq 0.99$, for any $i \in [r]$, with probability 0.99.

## 3   Open questions

**Open question 1.** *Theorem 2 improves over existing sparse PCA multi-spike recovery algorithms (which only assumed orthogonal spikes) by a factor of $1/r$ in the case where these planted signals have disjoint support. Can one still obtain an improvement of $1/r$ if we only assume orthogonality?*

**Open question 2.** *For $p \in \omega(1)$, there is an exponential gap between the bounds of Theorem 3 and state-of-the-art algorithms. A natural question is whether one can design better recover algorithms or prove stronger lower bounds for this range of tensor power $p \in \omega(1)$?*

**Remark** (Societal impact). *This work does not present any foreseeable negative societal consequence.*

## Acknowledgments and Disclosure of Funding

This project has received funding from the European Research Council (ERC) under the European Union's Horizon 2020 research and innovation programme (grant agreement No 815464). This research/project is supported by the National Research Foundation, Singapore under its AI Singapore Programme (AISG Award No: AISG-PhD/2021-08-013). We thank David Steurer for several helpful conversations. We thank Luca Corinzia and Paolo Penna for useful discussions about the planted sparse densest sub-hypergraph model.

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
