# A  Background

## A.1  Packings and nets

Packings and nets are useful in helping us discretize a possibly infinite metric space. Let $\mathcal{X}$ be a set of points and $d : \mathcal{X} \times \mathcal{X} \to \mathbb{R}^+$ be a (pseudo)metric[17] for $\epsilon > 0$. That is, $(\mathcal{X}, d)$ is a (pseudo)metric space. An $\epsilon$-packing $\mathcal{X}' \subseteq \mathcal{X}$ is a subset where any two distinct points $x, y \in \mathcal{X}'$ have distance $d(x, x') > \epsilon$. An $\epsilon$-net[18] $\mathcal{X}' \subseteq \mathcal{X}$ is a subset such that for any point $x \in \mathcal{X}$, there exists some point $x' \in \mathcal{X}'$ (possibly itself) where $d(x, x') \leq \epsilon$. Under these notions, the covering number $N(\mathcal{X}, d, \epsilon)$ and packing number $P(\mathcal{X}, d, \epsilon)$ are defined as the size of the *smallest $\epsilon$-net* of $\mathcal{X}$ and *largest $\epsilon$-packing* of $\mathcal{X}$ respectively. It is known[19] that

$$P(\mathcal{X}, d, 2\epsilon) \leq N(\mathcal{X}, d, \epsilon) \leq P(\mathcal{X}, d, \epsilon)$$

## A.2  Hermite polynomials

In this section, we introduce Hermite polynomials and state some properties used in our low-degree analysis. For further details, see [O'D14, Section 11.2].

**Definition 10** (Inner product of functions). *For a pair of functions $f$ and $g$ operating on the same domain $\mathcal{D}$, their inner product is defined by $\langle f, g \rangle = \mathbb{E}_{\boldsymbol{z} \sim \mathcal{D}}[f(\boldsymbol{z})g(\boldsymbol{z})]$.*

The set $\{1, \boldsymbol{z}, \boldsymbol{z}^2, \ldots, \boldsymbol{z}^D\}$ is a basis for the set of polynomials with maximum degree $D$ on Gaussian variable $\boldsymbol{z} \sim N(\mu, 1)$. By applying the Gram-Schmidt process and noting that odd functions in $\boldsymbol{z}$ have expectation 0, we can diagonalize this set to obtain an orthogonal basis: $H_{e_0}(\boldsymbol{z}) = 1$, $H_{e_1}(\boldsymbol{z}) = \boldsymbol{z}$, $H_{e_2}(\boldsymbol{z}) = \boldsymbol{z}^2 - 1$, $H_{e_3}(\boldsymbol{z}) = \boldsymbol{z}^3 - 3\boldsymbol{z}$, etc.

The orthogonal basis for polynomials of maximum degree $D$ $\{H_{e_n}\}_{n \in [D]}$ is also called the *probabilists' Hermite polynomials*. It is known that

$$\mathbb{E}_{\boldsymbol{z} \sim N(\mu, 1)}[H_{e_n}(\boldsymbol{z})] = \mu^n \quad \text{and} \quad \mathbb{E}_{\boldsymbol{z} \sim N(\mu, 1)}[(H_{e_n}(\boldsymbol{z}))^2] = n!$$

As we are interested in *orthonormal* bases, we use the *normalized* probabilists' Hermite polynomials $\{h_n\}_{n \in \mathbb{N}}$ where $h_n = \frac{1}{\sqrt{n!}} H_{e_n}$. One can check that

$$\mathbb{E}_{\boldsymbol{z} \sim N(\mu, 1)}[h_n(\boldsymbol{z})] = \frac{1}{\sqrt{n!}} \mu^n \quad \text{and} \quad \mathbb{E}_{\boldsymbol{z} \sim N(\mu, 1)}[(h_n(\boldsymbol{z}))^2] = 1$$

## A.3  Information theory

Techniques from the statistical minimax theory, such as the Fano method, allow us to *lower bound* the worst case behavior of *any* estimator. In the following discourse, we borrow some notation from [Duc16]. Given a single tensor observation $\boldsymbol{Y} = \boldsymbol{W} + \lambda x^{\otimes p}$ generated from an underlying signal $x \in U_t$ (i.e. the parameter of the observation is $\theta(\boldsymbol{Y}) = x$), an estimator $\widehat{\theta}(\boldsymbol{Y})$ outputs some unit vector in $\widehat{x} \in U_k$. For two vectors $x$ and $x'$, we use the pseudometric[20] $\rho(x, x') = \min\{\|x - x'\|_2, \|x + x'\|_2\}$ and the loss function $\Phi(t) = t^2/2$. Thus, $\Phi(\rho(x, x')) = 1 - |\langle x, x' \rangle|$ with the corresponding minimax risk being

$$\inf_{\widehat{\theta}} \sup_{x \in U_k} \mathbb{E}_{\boldsymbol{Y}} \left[ \Phi\left( \rho\left( \widehat{\theta}(\boldsymbol{Y}), \theta(\boldsymbol{Y}) \right) \right) \right] = \inf_{\widehat{x} \in U_k} \sup_{x \in U_k} \mathbb{E}_{\boldsymbol{Y}} \left[ 1 - |\langle \widehat{x}, x \rangle| \right]$$

A common way to *lower bound* the minimax risk function is to look at it from the lens of a *finite* testing problem. The *canonical hypothesis testing problem*[21] (in our context) is as follows. Let $\mathcal{X} \subseteq U_k$ be

---

[17] A metric satisfies 3 properties: (1) $d(x, y) = 0$ if and only if $x = y$; (2) $d(x, y) = d(y, x)$; (3) $d(x, y) \leq d(x, z) + d(z, y)$. A pseudometric may violate (1) by allowing $d(x, y) = 0$ for distinct $x \neq y$. Pseudometrics are also sometimes called semimetrics.

[18] Nets are also referred to as coverings.

[19] e.g. See resources such as [Tao14] and [Ver18, Section 4.2].

[20] Instead of the "standard" $\|x - x'\|_2$ loss, we want a loss function that captures the "symmetry" that $\langle x, x' \rangle^p = \langle x, -x' \rangle^p$ for even tensor powers $p$. Clearly, $\rho(x, y) = \rho(y, x)$ and one can check that $\rho(x, y) \leq \rho(x, z) + \rho(z, y)$. Observe that $\rho$ is a pseudometric (and not a metric) because $\rho(x, y) = 0$ holds for $x = -y$.

[21] See Section 13.2.1 in [Duc16].

an $\epsilon$-packing of $U_k$ of size $|\mathcal{X}| = m \geq P(U_k, \rho, \epsilon)$. That is, $\min_{x_i, x_j \in \mathcal{X}, i \neq j} \rho(x_i, x_j) > \epsilon$. Then, (1) Nature chooses a unit vector $\boldsymbol{x} \in \mathcal{X}$ *uniformly at random*; (2) We observe tensor $\boldsymbol{Y} = \boldsymbol{W} + \lambda \boldsymbol{x}^{\otimes p}$; (3) A test $\Psi : \mathcal{Y} \to \mathcal{X}$ determines what is the planted unit vector. Applying Fano's inequality (Lemma 11) and the data processing inequality (Lemma 12) to the above-mentioned canonical hypothesis testing problem, one can show[22] that

$$\inf_{\widehat{x} \in U_k} \sup_{x \in U_k} \mathbb{E}_{\boldsymbol{Y}} \left[ 1 - |\langle \widehat{x}, x \rangle| \right] \geq \inf_{\widehat{x} \in U_k} \sup_{x \in \mathcal{X}} \mathbb{E}_{\boldsymbol{Y}} \left[ 1 - |\langle \widehat{x}, x \rangle| \right] \geq \frac{\epsilon^2}{4} \cdot \left( 1 - \frac{I(x; \boldsymbol{Y}) + 1}{\log m} \right)$$

where $I(x; \boldsymbol{Y})$ is the mutual information between $x$ and $\boldsymbol{Y}$. Since it is known[23] that for $x \in \mathcal{X}$, one can upper bound $I(x; \boldsymbol{Y})$ by $I(x; \boldsymbol{Y}) \leq \max_{u,v \in \mathcal{X}} D_{KL} \left( \mathbb{P}_{\boldsymbol{Y} \sim \mathcal{Y}|u} \, \middle\| \, \mathbb{P}_{\boldsymbol{Y} \sim \mathcal{Y}|v} \right)$, we have

$$\inf_{\widehat{x} \in U_k} \sup_{x \in U_k} \mathbb{E}_{\boldsymbol{Y}} \left[ 1 - |\langle \widehat{x}, x \rangle| \right] \geq \frac{\epsilon^2}{4} \cdot \left( 1 - \frac{\max_{u,v \in \mathcal{X}} D_{KL} \left( \mathbb{P}_{\boldsymbol{Y} \sim \mathcal{Y}|u} \, \middle\| \, \mathbb{P}_{\boldsymbol{Y} \sim \mathcal{Y}|v} \right) + 1}{\log m} \right) \quad (3)$$

where $D_{KL}(\cdot \| \cdot)$ is the KL-divergence function and $\mathbb{P}_{\boldsymbol{Y} \sim \mathcal{Y}|u}$ is the probability distribution of observing $\boldsymbol{Y}$ from signal $u$ with additive standard Gaussian noise tensor $\boldsymbol{W}$.

We now state standard facts regarding Fano's inequality without proof. For an introductory exposition on Fano's inequality and its applications, we refer readers to [SC19].

**Lemma 11** (Fano's inequality (Uniform input distribution)). *Let $X, Y \in \mathcal{X}$ denote the (hidden) input and (observed) output. Given $Y$, let $\widehat{X} \in \mathcal{X}$ be the estimated version of $X$ by* any *estimator, and $P_e = \mathbb{P}[X \neq \widehat{X}]$ be the event that the estimation is wrong. If $X$ is uniformly distributed over $\mathcal{X}$, then*

$$P_e \geq 1 - \frac{I(X; \widehat{X}) + 1}{\log |\mathcal{X}|}$$

*where $I(X; \widehat{X}) = H(X) - H(X \mid \widehat{X})$ is the mutual information function.*

The following inequality makes the Fano's inequality more user-friendly since it replaces the $I(X; \widehat{X})$ term with $I(X; Y)$. In statistical learning, it is typically easier to bound $I(X; Y)$ as we know how $Y$ is generated given $X$.

**Lemma 12** (Data processing inequality). *Suppose variables $X$, $Y$ and $\widehat{X}$ form a Markov chain relation $X \to Y \to \widehat{X}$. That is, $X$ and $\widehat{X}$ are independent given $Y$. Then, $I(X; Y) \geq I(X; \widehat{X})$.*

### A.4 Low-degree method

The low-degree likelihood ratio is a proxy to model efficiently computable functions. It is closely related to the pseudo-calibration technique and it has been developed in a recent line of work on the Sum-of-Squares hierarchy ([BHK+19, HS17, HKP+17, Hop18]). In this section, we will only introduce the basic idea and encourage interested readers to see [Hop18, BKW20] for further details.

The objects of study are distinguishing versions of planted problems: given two distributions and an instance, the goal is to decide from which distribution the instance was sampled. For us, the distinguishing formulation takes the form of deciding whether the tensor $\boldsymbol{Y}$ was sampled according to the (planted) distribution as described in Model 4, or if it was sampled from the (null) distribution where $\boldsymbol{W} \in \otimes^p \mathbb{R}^n$ has i.i.d. entries sampled from $N(0, 1)$. In general, we denote with $\nu$ the null distribution and with $\mu$ the planted distribution with the hidden structure.

#### A.4.1 Background on Classical Decision Theory

From the point of view of classical Decision Theory, the optimal algorithm to distinguish between two distribution is well-understood. Given distributions $\nu$ and $\mu$ on a measurable space $\mathcal{S}$, the likelihood ratio $L(\boldsymbol{Y}) := d\mathbb{P}_\mu(\boldsymbol{Y})/d\mathbb{P}_\nu(\boldsymbol{Y})$[24] is the optimal function to distinguish whether $\boldsymbol{Y} \sim \nu$ or $\boldsymbol{Y} \sim \mu$ in the following sense.

---

[22]E.g. see [Duc16, Proposition 13.10], adapted to our context. Recall that we had $\Phi(t) = t^2/2$.

[23]E.g. see [SC19, Lemma 4].

[24]The Radon-Nikodym derivative.

**Proposition 13** ([NP33]). *If $\nu$ is absolutely continuous with respect to $\mu$, then the unique solution of the optimization problem*

$$\max \mathbb{E}_\mu\left[f(\boldsymbol{Y})\right] \qquad \textit{subject to } \mathbb{E}_\nu\left[f^2(\boldsymbol{Y})\right] = 1$$

*is the normalized likelihood ratio $L(\boldsymbol{Y})/\mathbb{E}_\nu\left[L(\boldsymbol{Y})^2\right]$ and the optimum value is $\mathbb{E}_\nu\left[L(\boldsymbol{Y})^2\right]$.*

Arguments about statistical distinguishability are also well-understood. Unsurprisingly, the likelihood ratio plays a major role here as well and a key concept is Le Cam's contiguity.

**Definition 14** ([Cam60]). *Let $\underline{\mu} = (\mu_n)_{n\in\mathbb{N}}$ and $\underline{\nu} = (\nu_n)_{n\in\mathbb{N}}$ be sequences of probability measures on a common probability space $\mathcal{S}_n$. Then $\underline{\mu}$ and $\underline{\nu}$ are contiguous, written $\underline{\mu} \triangleleft \underline{\nu}$, if as $n \to \infty$, whenever for $A_n \in \mathcal{S}_n$, $\mathbb{P}_{\underline{\mu}}(A_n) \to 0$ then $\mathbb{P}_{\underline{\nu}}(A_n) \to 0$.*

Contiguity allows us to capture the idea of indistinguishability of probability measures. Two contiguous sequences $\underline{\mu}, \underline{\nu}$ of probability measures are said to be indistinguishable if there is no function $f : \mathcal{S}_n \to \{0, 1\}$ such that $f(\boldsymbol{Y}) = 1$ with high probability whenever $\boldsymbol{Y} \sim \underline{\mu}$ and $f(\boldsymbol{Y}) = 0$ with high probability whenever $\boldsymbol{Y} \sim \underline{\nu}$. The *second moment method* allows us to establish contiguity through the likelihood ratio.

**Proposition 15.** *If $\mathbb{E}_\nu\left[L_n(\boldsymbol{Y})^2\right]$ remains bounded as $n \to \infty$, then $\underline{\mu} \triangleleft \underline{\nu}$.*

This discussion allows us to argue whether a given function can be used to distinguish between our planted and null distributions. In particular, for probability measures $\mu$ and $\nu$ over $\mathcal{S}$, and a given function $f : \mathcal{S} \to \mathbb{R}$, we can say that $f$ cannot distinguish between $\mu$ and $\nu$ if it satisfies the following bound on the $\chi^2$-divergence:

$$\frac{|\mathbb{E}_\mu(f(\boldsymbol{Y})) - \mathbb{E}_\nu(f(\boldsymbol{Y}))|}{\sqrt{\mathbb{V}_\nu(f(\boldsymbol{Y}))}} \leq o(1).$$

### A.4.2 Background on the Low-degree Method

The main problem with the likelihood ratio is that it is hard to compute in general, thus the analysis has to be restricted to the space of efficiently computable functions. Concretely, we use low-degree multivariate polynomials in the entries of the observation $\boldsymbol{Y}$ as a proxy for efficiently computable functions. By denoting the space of degree $\leq D$ polynomials in $\boldsymbol{Y}$ with $\mathbb{R}_{\leq D}[\boldsymbol{Y}]$, we can establish a low-degree version of the Neyman-Pearson lemma.

**Proposition 16** (e.g. [Hop18]). *The unique solution of the optimization problem*

$$\max_{f\in\mathbb{R}_{\leq D}[\boldsymbol{Y}]} \mathbb{E}_\mu\left[f(\boldsymbol{Y})\right] \qquad \textit{subject to } \mathbb{E}_\nu\left[F(\boldsymbol{Y})\right] = 1$$

*is the normalized orthogonal projection $L^{\leq D}(\boldsymbol{Y})/\mathbb{E}_\nu\left[L^{\leq D}(\boldsymbol{Y})^2\right]$ of the likelihood ratio $L(\boldsymbol{Y})$ onto $\mathbb{R}_{\leq D}[\boldsymbol{Y}]$ and the value of the optimization problem is $\mathbb{E}_\nu\left[L^{\leq D}(\boldsymbol{Y})^2\right]$.*

With the reasoning above in mind, it is then natural to argue that a polynomial $p(\boldsymbol{Y}) \in \mathbb{R}_{\leq D}[\boldsymbol{Y}]$ cannot distinguish between $\mu$ and $\nu$ if

$$\frac{|\mathbb{E}_\mu(p(\boldsymbol{Y})) - \mathbb{E}_\nu(p(\boldsymbol{Y}))|}{\sqrt{\mathbb{V}_\nu(p(\boldsymbol{Y}))}} \leq o(1). \tag{4}$$

It is important to remark that, at the heart of our discussion, there is the belief that in the study of planted problems, low-degree polynomials capture the computational power of efficiently computable functions. This can be phrased as the following conjecture.

**Conjecture 17** (Informal[25]). *For "nice" sequences of probability measures $\underline{\mu}$ and $\underline{\nu}$, if there exists $D = D(d) \geq O(\log d)$ for which $\mathbb{E}_\nu\left[L^{\leq D}(\boldsymbol{Y})^2\right]$ remains bounded as $d \to \infty$, then there is no polynomial-time algorithm that distinguishes in the sense described in Appendix A.4.1[26].*

A large body of work support this conjecture (see citations mentioned) by providing evidence of an intimate relation between polynomials, sum of squares algorithms, and lower bounds.

---

[25]See [BHK$^+$19, HS17, HKP$^+$17, Hop18].

[26]We do not explain what "nice" means (e.g. see [Hop18]) and remark that the most general formulation of the conjecture above (i.e. a broad definition of "nice" distributions) has been rejected ([HW20]).

### A.4.3 Chi-squared divergence and orthogonal polynomials

From a technical point of view, the key observation used to prove bounds for low-degree polynomials is the fact that the polynomial which maximizes the ratio in Eq. (4) has a convenient characterization in terms of orthogonal polynomials with respect to the null distribution.

Formally, for any linear subspace of polynomials $\mathcal{S}_{\leq D} \subseteq \mathbb{R}[\boldsymbol{Y}]_{\leq D}$ and any absolutely continuous probability distribution $\nu$ such that all polynomials of degree at most $2D$ are $\nu$-integrable, one can define an inner product in the space $\mathcal{S}_{\leq D}$ as follows

$$\forall p, q \in \mathcal{S}_{\leq D} \quad \langle p, q \rangle = \mathbb{E}_{\boldsymbol{Y} \sim \nu} p(\boldsymbol{Y}) q(\boldsymbol{Y}) .$$

Hence we can talk about orthonormal basis in $\mathcal{S}_{\leq D}$ with respect to this inner product.

**Proposition 18** (See [dKNS20] for a proof). *Let $\mathcal{S}_{\leq D} \subseteq \mathbb{R}[\boldsymbol{Y}]_{\leq D}$ be a linear subspace of polynomials of dimension $N$. Suppose that $\nu$ and $\mu$ are probability distributions over $\boldsymbol{Y} \in \mathbb{R}^{n \times d}$ such that any polynomial of degree at most $D$ is $\mu$-integrable and any polynomial of degree at most $2D$ is $\nu$-integrable. Suppose also that $\nu$ is absolutely continuous. Let $\{\psi_i(\boldsymbol{Y})\}_{i=1}^{N}$ be an orthonormal basis in $\mathcal{S}_{\leq D}[\boldsymbol{Y}]$ with respect to $\nu$. Then*

$$\max_{p \in \mathcal{S}_{\leq D}} \frac{(\mathbb{E}_\mu p(\boldsymbol{Y}))^2}{\mathbb{E}_\nu p^2(\boldsymbol{Y})} = \sum_{i=1}^{N} (\mathbb{E}_\mu \psi_i)^2 .$$

In the case of Gaussian noise, a useful orthonormal basis in $\mathbb{R}[\boldsymbol{Y}]_{\leq D}$ is the system of Hermite polynomials $\{H_\alpha(\boldsymbol{Y})\}_{|\alpha| \leq D}$ (see Appendix A.2). By applying Proposition 18 to the subspace of polynomials such that $\mathbb{E}_\nu p(\boldsymbol{Y}) = 0$, we get

**Corollary 19.** *Let $\nu$ be Gaussian. Suppose that the distribution $\mu$ is so that any polynomial of degree at most $D$ is $\mu$-integrable. Then*

$$\max_{p \in \mathbb{R}[\boldsymbol{Y}]_{\leq D}} \frac{(\mathbb{E}_\mu p(\boldsymbol{Y}) - \mathbb{E}_\nu p(\boldsymbol{Y}))^2}{\mathbb{V}_\nu p(\boldsymbol{Y})} = \sum_{0 < |\alpha| \leq D} (\mathbb{E}_\mu H_\alpha(Y))^2 .$$

### A.5 Sparse norm bounds

Denote $\mathcal{S}^{n-1} = \{x \in \mathbb{R}^n : \|x\|_2 = 1\}$ as the $n$-dimensional unit sphere and $A \in \mathbb{R}^{m \times n}$ be a matrix. Then, the matrix norm of $A$ is defined as

$$\|A\| = \max_{x \in \mathcal{S}^{n-1}} \|Ax\|_2 = \max_{x \in \mathcal{S}^{m-1}, y \in \mathcal{S}^{n-1}} x^\top A y = \max_{x \in \mathcal{S}^{m-1}, y \in \mathcal{S}^{n-1}} \sum_{i=1}^{m} \sum_{j=1}^{n} A_{i,j} x_i y_j$$

More generally, the tensor norm of an order $p \geq 2$ tensor $T \in \mathbb{R}^{n_1 \times \cdots \times n_p}$ is defined as

$$\|T\| = \max_{x_{(1)} \in \mathcal{S}^{n_1 - 1}, \ldots, x_{(p)} \in \mathcal{S}^{n_p - 1}} T\left(x_{(1)}, x_{(2)}, \ldots, x_{(p)}\right)$$

$$= \max_{x_{(1)} \in \mathcal{S}^{n_1 - 1}, \ldots, x_{(p)} \in \mathcal{S}^{n_p - 1}} \sum_{i_1 = 1}^{n_1} \cdots \sum_{i_p = 1}^{n_p} T_{i_1, \ldots, i_p} x_{(1), i_1} x_{(2), i_2} \cdots x_{(p), i_p}$$

In the following, let all dimensions be equal (i.e. $n = m = n_1 = \ldots = n_p$). Without any sparsity conditions, it is known[27] that $\|\boldsymbol{A}\| \leq \mathcal{O}(\sqrt{n} + t)$ with high probability for matrix $\boldsymbol{A}$ with i.i.d. standard Gaussian entries. For tensors, [TS14] proved that $\|\boldsymbol{T}\| \leq \mathcal{O}\left(\sqrt{np \log(p) + \log(1/\gamma)}\right)$ with probability at least $1 - \gamma$. These results are typically proven using $\epsilon$-net arguments over the unit sphere $\mathcal{S}^{n-1}$. However, to the best of our knowledge, there is no known result for bounding the norm of $\boldsymbol{A}$ or $\boldsymbol{T}$ when interacting with $r$ distinct (at most) $s$-sparse unit vectors from the set $\mathcal{S}_s^{n-1} = \{x \in \mathbb{R}^n : \|x\|_2 = 1, |\mathcal{I}_x| \leq s \leq n\}$.

Using $\epsilon$-net arguments, we bound $\left|\boldsymbol{T}\left(x_{(1)}, \ldots, x_{(p)}\right)\right|$ when $x_{(1)}, \ldots, x_{(p)}$ are $r$ distinct vectors from $\mathcal{S}_s^{n-1}$. Our result recovers known bounds, up to constant factors, when $s = n$ and $r = p$.

---

[27]e.g. See [Ver18, Section 4.4.2], [Tro15, Section 4.2.2] and [Tao12, Section 2.3.1].

**Lemma 20** (Tensor sparse bound). *Let $\boldsymbol{T} \in \otimes^p \mathbb{R}^n$ be an order $p \geq 2$ tensor with i.i.d. standard Gaussian entries and $x_{(1)}, \ldots, x_{(p)} \in \mathcal{S}_s^{n-1}$ be $r$ distinct (at most) $s$-sparse unit vectors. Then, for $1 \leq s \leq n$, $1 \leq r \leq p$, and $\gamma \in (0, 1)$,*

$$\mathbb{P}\left[\left|\boldsymbol{T}(x_{(1)}, \ldots, x_{(p)})\right| \geq \sqrt{8 \cdot \left(4rs \ln\left(\frac{np}{s}\right) + \ln\left(\frac{1}{\gamma}\right)\right)}\right] \leq 2\gamma$$

*Proof sketch of Lemma 20 (described for $p = 2$).* We fix an $\epsilon$-net $\mathcal{N}(\mathcal{S}_s^{n-1})$ of size $N(\mathcal{S}_s^{n-1}, \epsilon)$ over sparse vectors, bound the norm for an arbitrary point in $\mathcal{N}(\mathcal{S}_s^{n-1}) \times \mathcal{N}(\mathcal{S}_s^{n-1})$, and apply union bound over all points in $\mathcal{N}(\mathcal{S}_s^{n-1}) \times \mathcal{N}(\mathcal{S}_s^{n-1})$. Since $\mathcal{S}_s^{n-1} \times \mathcal{S}_s^{n-1}$ is compact, there is a maximizing point that attains the norm. We complete the proof by relating the maximizer (and hence the norm) to its closest point in $\mathcal{N}(\mathcal{S}_s^{n-1}) \times \mathcal{N}(\mathcal{S}_s^{n-1})$. See Appendix F.1 for the formal proof. $\square$

# B    Limited Brute Force Recovery Algorithm

We first discuss how to extend the discussions from Section 2 to the case where signals are approximately flat. Then, we will prove the most general form of our algorithmic result[28] (Theorem 21) that our limited brute force search algorithm that recovers exact individual signal supports under some algorithmic assumptions. Finally, we explain how to extend our techniques to handle single-spike general tensors where the signal takes the form $x_{(1)} \otimes \ldots \otimes x_{(p)}$ involving $1 \leq \ell \leq p$ distinct $k$-sparse tensors in Appendix B.2. As in Section 2, our exposition will ignore the constant factors introduced by Algorithm 1.

## B.1    Approximately flat signals

By factoring $A \geq 1$ into our assumptions on minimal signal strength $\lambda_{\min}$ and relative strength ratio $\kappa$, we can extend the above analyses (using the same proof outline as in Section 2.2) so that Algorithm 3 recovers the individual supports of multiple approximately flat $(k, A)$-sparse signals. Besides accounting for $A$ factors, the only significant change in the analysis is in how we lower bound $\langle \boldsymbol{Y}^{(1)}, \boldsymbol{v}_* \rangle$. Consider the first iteration (by our discussion in Section 2.2, other iterations are similar) and let $u_* \in U_t$ be the vector satisfying

$$\lambda \langle x, u_* \rangle^p = \max_{q \in [r], u \in U_t} \lambda_q \langle x_{(q)}, u \rangle^p .$$

This choice allows us to account for skewed signals. Let $\lambda_{\min} = \min_{q \in [r]} \lambda_q$ and $\lambda_{\max} = \max_{q \in [r]} \lambda_q$. For some $0 < \epsilon \leq 1/2$. Suppose that

$$\lambda_{\min} \gtrsim \frac{A^p}{\epsilon \cdot (1-\epsilon)^p} \cdot \sqrt{t \left(\frac{k}{t}\right)^p \log n} \quad \text{and} \quad \lambda_{\min} \gtrsim A^{2p} \cdot \left(\frac{\epsilon}{1-\epsilon}\right)^{p-1} \cdot \lambda_{\max} . \qquad (5)$$

By definition of $\boldsymbol{v}_*$, we have

$$\langle \boldsymbol{Y}^{(1)}, \boldsymbol{v}_* \rangle \geq \langle \boldsymbol{Y}^{(1)}, u_* \rangle = \lambda \langle x, u_* \rangle^p + \langle \boldsymbol{W}^{(1)}, u_*^{\otimes p} \rangle . \qquad (6)$$

Conversely, consider an arbitrary $u \in U_t$ such that $\left|\text{supp}(u) \cap \text{supp}\left(x_{(q)}\right)\right| < (1 - \epsilon) \cdot t$ for all $q \in [r]$. Intuitively, the largest attainable value for $\langle \boldsymbol{Y}^{(1)}, u^{\otimes p} \rangle$ is obtained removing $\epsilon \cdot t$ entries from $u_*$ and placing them on some highly skewed signal. Using Eq. (5) and Eq. (6), it is possible to show[29]

$$\langle \boldsymbol{Y}^{(1)}, u^{\otimes p} \rangle \leq \lambda \langle x, u_* \rangle^p \cdot \left(1 - \frac{\epsilon}{A^2}\right)^{p-1} + \langle \boldsymbol{W}^{(1)}, u^{\otimes p} \rangle .$$

Thus, it follows that $\boldsymbol{v}_*$ satisfies $\left|\text{supp}\left(\boldsymbol{v}_*\right) \cap \text{supp}\left(x_{(i)}\right)\right| \geq (1 - \epsilon)t$ for some $i \in [r]$. We can now repeat the same analysis as Section 2.2 to argue that $\boldsymbol{\alpha}$ behaves as an indicator vector for $\text{supp}\left(x_{(i)}\right)$, taking account of $A$ factors.

---

[28]Theorem 1 and Theorem 2 from the main text are direct consequences of Theorem 21.

[29]For the full derivation, see the proof of Lemma 23 in Appendix F.2. A $\sqrt{2}$ factor appears due to Algorithm 1.

**Theorem 21** (Multi-spike recovery for approximately flat signals). *Consider Model 4. Suppose*

$$\lambda_r \gtrsim \frac{\kappa}{(A\epsilon)^p} \sqrt{t \left(\frac{k}{t}\right)^p \ln\left(\frac{n}{\delta}\right)}, \quad \lambda_r \geq \kappa \cdot \lambda_1, \quad and \quad \kappa \geq 5A^{2p} \left(\frac{\epsilon}{1-\epsilon}\right)^{p-1}.$$

*Then, Algorithm 3 that runs in $\mathcal{O}(rpn^{p+t})$ time and, with probability at least $1 - \delta$, outputs the individual signal supports* $\mathrm{supp}\left(x_{(\pi(1))}\right), \ldots, \mathrm{supp}\left(x_{(\pi(r))}\right)$ *with respect to some unknown bijection* $\pi : [r] \to [r]$.

We remark that the constant factors are chosen to make the analysis clean; smaller factors are possible. As discussed in Remark 9, after recovering the support coordinates, one can then run known tensor PCA recovery methods on the appropriate sub-tensor to obtain a good approximation of each signal.

For convenience, let us restate Algorithm 1 and Algorithm 3.

---

**Algorithm 1** Preprocessing

---

**Input**: $Y$.
Sample a Gaussian tensor $Z \in \otimes^p \mathbb{R}^n$ where each entry is an i.i.d. standard Gaussian $N(0, 1)$.
**Return** two independent copies $Y^{(1)}$ and $Y^{(2)}$ of $Y$ as follows:

$$Y^{(1)} = \frac{1}{\sqrt{2}} (Y + Z) \quad and \quad Y^{(2)} = \frac{1}{\sqrt{2}} (Y - Z)$$

---

**Algorithm 3** Multi-spike limited brute force

---

**Input**: $k, t, r$ and $Y^{(1)}, Y^{(2)}$ obtained from Algorithm 1.
**Repeat** for $i = 1$ to $r$:

Compute $v_* := \mathrm{argmax}_{u \in U_t} \langle Y^{(1)}, u^{\otimes p} \rangle$ subject to $\mathrm{supp}(v_*) \cap \left( \bigcup_{1 \leq j < i} \mathcal{I}_j \right) = \emptyset$.

Compute the vector $\alpha \in \mathbb{R}^n$ with entries $\alpha_\ell := \langle Y^{(2)}, v_*^{\otimes p-1} \otimes e_\ell \rangle$ for every $\ell \in [n]$.
Let $\mathcal{I}_i$ be the set of indices of the largest $k$ entries of $\alpha$.

**Return** $\mathcal{I}_1, \ldots, \mathcal{I}_r$.

---

Let us begin with a simple running time analysis.

**Lemma 22.** *Algorithm 3 runs in $\mathcal{O}(rpn^{p+t})$ time.*

*Proof of Lemma 22.* Sampling $Z$ and creating copies $Y^{(1)}$ and $Y^{(2)}$ take $\mathcal{O}(n^p)$ time. Fix an arbitrary round. Observe that $|U_t| = \binom{n}{t} 2^t \leq ((2e)/t)^t n^t \leq e^2 n^t$. Each computation of $\langle Y^{(1)}, u^{\otimes p} \rangle$ can be naively performed in $\mathcal{O}(pn^p)$ time while checking whether for disjoint support can be done naively in additional $\mathcal{O}(n^2)$ time for each $u \in U_t$. Similarly, the computation of $\alpha$ can be done in $\mathcal{O}(pn^p)$ time and we can perform a linear scan in $\mathcal{O}(n)$ time to obtain the largest $k$ entries. So, an arbitrary round runs in $\mathcal{O}(pn^{p+t})$ time. We perform the entire process $r$ times. $\square$

Algorithm 3 recovers the exact $k$-sparse support of some signal $x_{(\pi(i))}$ in each round. That is, $\mathcal{I}_{x_{(\pi(i))}} = \mathrm{supp}\left(x_{(\pi(i))}\right)$ for $i \in [r]$. It succeeds when these two claims hold for any round $i \in [r]$:

(I) The $t$-sparse maximizer $v_*$ shares $\geq (1 - \epsilon) \cdot t$ support coordinates with some signal $x_{(\pi(i))}$.

(II) The $k$ largest entries of $\alpha$ belong to the support $\mathrm{supp}\left(x_{(\pi(i))}\right)$ of $x_{(\pi(i))}$.

Lemma 23 and Lemma 24 address these claims respectively. See Appendix F.2 for their proofs.

**Lemma 23.** *Consider Model 4 and an arbitrary round $i \in [r]$. Suppose*

$$\lambda_r \geq \frac{32\kappa}{(A\epsilon)^p} \sqrt{t \left(\frac{k}{t}\right)^p \ln(n)}, \quad \lambda_r \geq \kappa \cdot \lambda_1, \quad and \quad \kappa \geq 5A^{2p} \left(\frac{\epsilon}{1-\epsilon}\right)^{p-1}.$$

*Then,*

$$\mathbb{P}\left[\left|\operatorname{supp}\left(\boldsymbol{v}_*\right) \cap \operatorname{supp}\left(x_{(\pi(i))}\right)\right| \geq (1-\epsilon) \cdot t\right] \geq 1 - 4\exp\left(-\lambda_r^2 \frac{\epsilon^2}{128A^4}\left(\frac{t}{k}\right)^p\right)$$

**Lemma 24.** *Consider Model 4 and an arbitrary round $i \in [r]$. Suppose*

$$\lambda_r \geq \frac{32\kappa}{(A\epsilon)^p}\sqrt{t\left(\frac{k}{t}\right)^p \ln(n)}, \quad \lambda_r \geq \kappa \cdot \lambda_1, \quad and \quad \kappa \geq 5A^{2p}\left(\frac{\epsilon}{1-\epsilon}\right)^{p-1}.$$

*Further suppose that $\left|\operatorname{supp}\left(\boldsymbol{v}_*\right) \cap \operatorname{supp}\left(x_{(\pi(i))}\right)\right| \geq (1-\epsilon) \cdot t$. Then, the largest (in magnitude) coordinates of $\boldsymbol{\alpha}$ are $\operatorname{supp}\left(x_{(\pi(i))}\right)$ with probability at least $1 - 2n\exp\left(-\lambda_r^2 \frac{A^{2p}\epsilon^{2p-2}}{16\kappa^2 t}\left(\frac{t}{k}\right)^p\right)$.*

We now prove Theorem 21 using Lemma 22, Lemma 23, and Lemma 24.

*Proof of Theorem 21.* Lemma 22 gives the running time. The correctness of Algorithm 3 hinges on Lemma 23 and Lemma 24 always succeeding. In each round, we need Lemma 23 to succeed once and Lemma 24 to succeed at most $n$ times. There are a total of at most $r(1+n)$ events, each failing with probability at most $4n\exp\left(-\lambda_r^2 \frac{A^{2p-4}\epsilon^{2p-2}}{128\kappa^2 t}\left(\frac{t}{k}\right)^p\right)$. By union bound, the probability of *any* event fails is at most $r(1+n) \cdot 4n\exp\left(-\lambda_r^2 \frac{A^{2p-4}\epsilon^{2p-2}}{128\kappa^2 t}\left(\frac{t}{k}\right)^p\right)$. By the disjoint signal support assumption, we have $r \leq n$. So, when $\lambda_r \gtrsim \frac{\kappa}{(A\epsilon)^p}\sqrt{t\left(\frac{k}{t}\right)^p \ln\left(\frac{n}{\delta}\right)}$, Algorithm 3 succeeds with probability at least $1 - \delta$. $\square$

## B.2    General tensors for single spike

We now briefly describe how to extend the model of Model 4 to the case where the single tensor signal could be made up of $1 \leq \ell \leq p$ distinct $k$-sparse vectors[30]: instead of the signal being $x^{\otimes p}$, it is $x_{(1)} \otimes \ldots \otimes x_{(p)}$ involving $\ell$ distinct vectors. The discussions in this section can be further generalized to the case of multiple approximately flat spikes using the techniques from Section 2.2 and Appendix B.1.

Given $\ell$, we can modify Algorithm 2 to search over $U_t^{\otimes \ell}$ and compute $\boldsymbol{v}_*$ that maximizes

$$\langle \boldsymbol{Y}^{(1)}, u_{(1)} \otimes \ldots \otimes u_{(p)}\rangle$$

where there are $\binom{p-1}{\ell-1}$ possible ways[31] to form the signal using $\ell$ distinct $t$-sparse vectors. By Lemma 20, $\langle \boldsymbol{W}^{(1)}, u_{(1)} \otimes \ldots \otimes u_{(p)}\rangle \lesssim \sqrt{\ell t \log(n)}$ whenever one (or more) of the $t$-sparse vectors used to form $u_{(1)} \otimes \ldots \otimes u_{(p)}$ is *not* part of the actual signal. Suppose the maximizer $\boldsymbol{v}_*$ involves $\ell$ distinct vectors $\boldsymbol{v}_{*,(1)}, \ldots, \boldsymbol{v}_{*,(\ell)}$. For notational convenience, let us write $\boldsymbol{v}_* \setminus \boldsymbol{v}_{*,(q)}$ to mean the tensor of order $p-1$ derived by removing one copy of $\boldsymbol{v}_{*,(q)}$ from $\boldsymbol{v}_*$. Then, for each distinct vector $\boldsymbol{v}_{*,(q)}$ in the maximizer, define $\boldsymbol{\alpha}_{(q)} \in \mathbb{R}$ where $\alpha_{(q),i} = \langle \boldsymbol{Y}^{(2)}, (\boldsymbol{v}_* \setminus \boldsymbol{v}_{*,(q)}) \otimes e_i\rangle$ and output the $k$ largest entries of $\boldsymbol{\alpha}_{(q)}$ as the support of $\boldsymbol{v}_{*,(q)}$.

This modified algorithm will run in time $\mathcal{O}\left(\ell(pe)^\ell n^{p+\ell t}\right)$.[32] Adapting our analysis for the single spike accordingly (by using Lemma 20 with $\ell$ distinct vectors) will show that we can recover the signal supports of each $u_{(q)}$ whenever $\lambda \gtrsim \sqrt{\ell t(k/t)^p \log n}$.[33] Notice that this is an improvement over the algorithm of [LZ20] for $\ell \in o(p)$ or $t \geq 2$ when $p \in \omega(1)$.

---

[30]This model has been studied by [LZ20]. To be precise, they actually allow different *known* sparsity levels for each $x_{(q)}$ vector. Here, we assume that all of them are $k$-sparse. It is conceptually straightforward (but complicated and obfuscates the key idea) to extend the current discussion to allow different sparsity values.

[31]There are $\binom{p-1}{\ell-1}$ ways to obtain integer solutions to $x_1 + \ldots + x_\ell = p$ assuming $x_1 \geq 1, \ldots, x_\ell \geq 1$.

[32]The $\ell$ factor is due to using $\ell$ copies of $\boldsymbol{\alpha}$. The increase from $p$ to $p^\ell$ is due to trying $\binom{p-1}{\ell-1}$ combinations. The increase of $n^t$ to $n^\ell e^\ell$ is due to searching over $U_t^{\otimes \ell}$.

[33]The extra $\sqrt{\ell}$ factor follows from Lemma 20 to accomodate $\ell$ distinct vectors in the maximization.

# C  Computational Low-Degree Bounds

In this section, we formalize our results on the computational low-degree bounds for sparse tensor PCA. We will first show a low-degree lower bound on the distinguishing problem using $k$-sparse scaled Rademacher unit vectors and then will give a low-degree distinguishing algorithm showing that the lower-bound is tight in certain parameter regimes.

Following the discussion in Appendix A.4, we design the following distinguishing problem.

**Problem 25** (Hypothesis testing for single-spiked $k$-sparse scaled Rademacher vectors). *Given an observation tensor $\boldsymbol{Y} \in \otimes^p \mathbb{R}^n$, decide whether:*

$$\text{Null distribution } H_0 : \boldsymbol{Y} = \boldsymbol{W}$$

$$\text{Planted distribution } H_1 : \boldsymbol{Y} = \boldsymbol{W} + \lambda \boldsymbol{x}^{\otimes p}$$

*where $\boldsymbol{W}$ is a noise tensor with i.i.d. $N(0,1)$ entries and $\boldsymbol{x}$ is a $k$-sparse scaled Rademacher unit vector whose entries are independently drawn as follows:*

$$\boldsymbol{x}_i = \begin{cases} 1/\sqrt{k} & \text{with probability } k/(2n), \\ -1/\sqrt{k} & \text{with probability } k/(2n), \\ 0 & \text{with probability } 1 - k/n. \end{cases}$$

Formally speaking, the vector $\boldsymbol{x}$ in Problem 25 is not necessarily a unit vector, as compared to the planted signal in the single-spike case of Model 4. However, since $\boldsymbol{x}$ is $k(1 + o(1))$-sparse with high probability, a lower bound given by Problem 25 implies a distinguishing lower bound for single-spike sparse tensor model with a planted $k(1 + o(1))$-sparse vector and signal strength $\frac{\lambda}{1+o(1)}$. We study Problem 25 through the lens of low-degree polynomials. Since $\boldsymbol{W}$ is Gaussian noise, we use the set of normalized probabilists' Hermite polynomials $\{h_\alpha\}_\alpha$ as our orthogonal basis. Our strategy is similar to prior works such as [HKP$^+$17, HS17, DKWB19]: By examining the low-degree analogue of the $\chi^2$-divergence between probability measures, we will show that low-degree polynomial estimators cannot distinguish $H_0$ and $H_1$.

We now state the two main theorems that we will prove in the following subsections. For a cleaner exposition, we defer some proofs to Appendix F.3.

**Theorem 26** (Single-spike low-degree distinguishability lower bound). *Let $p \geq 2$, $1 \leq D \leq 2n/p$, $\boldsymbol{Y} \in \otimes^p \mathbb{R}^n$ be an observation tensor, $\boldsymbol{x}$ be a $k$-sparse scaled Rademacher vector, and $\{h_\alpha\}_\alpha$ be the set of normalized probabilists' Hermite polynomials. If $0 \leq \epsilon \leq 1/2$ and*

$$\lambda \leq \sqrt{\frac{\epsilon D}{e 4^p}} \min \left\{ \left(\frac{n}{pD}\right)^{p/4}, \left(\frac{k}{pD}\left(1 + \left|\ln\left(\frac{npD}{ek^2}\right)\right|\right)\right)^{p/2} \right\},$$

*then*

$$\chi^2(H_1 \| H_0) = \sup_{|\alpha| \leq D} \frac{(\mathbb{E}_{H_1} h_\alpha(\boldsymbol{Y}) - \mathbb{E}_{H_0} h_\alpha(\boldsymbol{Y}))^2}{Var_{H_0} h_\alpha(\boldsymbol{Y})} = \sum_{|\alpha| \leq D} (\mathbb{E}_{H_1} h_\alpha(\boldsymbol{Y}))^2 \leq 2\epsilon.$$

**Theorem 27** (Single-spike low-degree distinguishability lower bound). *Let $p \geq 2$, $1 \leq D \leq 2n/p$, $\boldsymbol{Y} \in \otimes^p \mathbb{R}^n$ be an observation tensor, $\boldsymbol{x}$ be a $k$-sparse scaled Rademacher vector, and $\{h_\alpha\}_\alpha$ be the set of normalized probabilists' Hermite polynomials. If either of the following holds:*

1. *If $D$ is even and*

$$\lambda \geq \epsilon^{\frac{1}{2D}} e^{\frac{p}{2}} \sqrt{D} \left(\frac{n}{pD}\right)^{\frac{p}{4}}$$

2. *If $p \leq n$, $D \leq \frac{\ln^2(n/p)}{4e^2}$ is even, $\sqrt{np} \cdot \left(\frac{e\sqrt{D}}{\ln(n/k)}\right) \leq k \leq \sqrt{np}$, and*

$$\lambda \geq \epsilon^{\frac{1}{2D}} \sqrt{D} \left(\frac{k}{pD} \ln\left(\frac{n}{k}\right)\right)^{\frac{p}{2}}$$

*then*

$$\chi^2(H_1 \| H_0) = \sup_{|\alpha| \leq D} \frac{(\mathbb{E}_{H_1} h_\alpha(\boldsymbol{Y}) - \mathbb{E}_{H_0} h_\alpha(\boldsymbol{Y}))^2}{Var_{H_0} h_\alpha(\boldsymbol{Y})} = \sum_{|\alpha| \leq D} (\mathbb{E}_{H_1} h_\alpha(\boldsymbol{Y}))^2 \geq \epsilon.$$

### C.1 Low-degree lower bound

To prove our computational lower bound, we first compute $(\mathbb{E}_{H_1} h_\alpha(\boldsymbol{Y}))^2$ explicitly in Lemma 28 using properties of the normalized probabilists' Hermite polynomials for a given degree parameter $D$. Then, we upper bound $\sum_{|\alpha| \leq D} (\mathbb{E}_{H_1}[h_\alpha(\boldsymbol{Y})])^2$ using Lemma 29 and Lemma 30. Solving for the condition on $\lambda$ such that $\sum_\alpha (\mathbb{E}_{H_1}[h_\alpha(\boldsymbol{Y})])^2 \ll \epsilon$ yields our computational lower bound Theorem 26.

**Lemma 28.** *Let $p \geq 2$, $d \geq 1$, $\boldsymbol{Y} \in \otimes^p \mathbb{R}^n$ be an observation tensor, $\boldsymbol{x}$ be a $k$-sparse scaled Rademacher vector, and $\{h_\alpha\}_\alpha$ be the set of normalized probabilists' Hermite polynomials. An entry of $\boldsymbol{Y} \in \otimes^p \mathbb{R}^n$ can be indexed by either an integer from $[n^p]$ or a $p$-tuple. Define $\phi : [n^p] \rightarrow [n]^p$, $\alpha$, $c(\alpha)$, $s(\alpha)$, and $\mathbb{1}_{even(c(\alpha))}$ as follows:*

- *$\phi(i)$ maps to a $p$-tuple indicating the $p$ (possibly repeated) entries of $\boldsymbol{x}$ that are used.*

- *$\alpha = (\alpha_1, \ldots, \alpha_{n^p})$ is an $n^p$-tuple that corresponds to a Hermite polynomial of degree $|\alpha| = \sum_{i=1}^{n^p} \alpha_i$. For each $i$, $\alpha_i$ is the number of times entry $\boldsymbol{Y}_{\phi(i)}$ was chosen, where each $\boldsymbol{Y}_{\phi(i)}$ references $p$ coordinates of $\boldsymbol{x}$.*

- *$c(\alpha) = (c_1, \ldots, c_n)$, where $c_j$ is the number of times $\boldsymbol{x}_j$ is used in $\alpha$.*

- *$s(\alpha)$ is the number of distinct non-zero $\boldsymbol{x}_j$'s in $c(\alpha)$.*

- *$\mathbb{1}_{even(c(\alpha))}$ be the indicator whether all $c_j$'s are even.*

*Under these definitions, we have the following:*

$$(\mathbb{E}_{H_1} h_\alpha(\boldsymbol{Y}))^2 = \lambda^{2d} k^{-pd} \mathbb{1}_{even(c(\alpha))} \left(\frac{k}{n}\right)^{2s(\alpha)} \left(\Pi_{i=1}^{n^p} \frac{1}{(\alpha_i)!}\right).$$

**Lemma 29.** *Let $p \geq 2$, $1 \leq D \leq 2n/p$, $\boldsymbol{Y} \in \otimes^p \mathbb{R}^n$ be an observation tensor, $\boldsymbol{x}$ be a $k$-sparse scaled Rademacher vector, and $\{h_\alpha\}_\alpha$ be the set of normalized probabilists' Hermite polynomials. Then,*

$$\sum_{|\alpha| \leq D} (\mathbb{E}_{H_1}[h_\alpha(\boldsymbol{Y})])^2 \leq \sum_{d=1}^D \frac{\lambda^{2d}}{d!} \sum_{s=1}^{pd/2} \left(\frac{ek^2}{sn}\right)^s \left(\frac{s}{k}\right)^{pd}.$$

**Lemma 30.** *For $p \geq 2$, $d \geq 1$, $1 \leq k \leq n$ and $1 \leq s \leq pd/2$, we have*

$$\left(\frac{ek^2}{sn}\right)^s \left(\frac{s}{k}\right)^{pd} \leq \left[\frac{2pd}{\min\left\{\sqrt{npd},\ k\left(1 + \left|\ln\left(\frac{npd}{ek^2}\right)\right|\right)\right\}}\right]^{pd}.$$

We are now ready to prove Theorem 26.

*Proof of Theorem 26.* Lemma 29 and Lemma 30 together tell us that

$$\sum_{|\alpha|\leq D} (\mathbb{E}_{H_1}[f_\alpha(\boldsymbol{Y})])^2 \leq \sum_{d=1}^{D} \frac{\lambda^{2d}}{d!} \frac{pd}{2} \left[ \frac{2pd}{\min\left\{ \sqrt{npd},\ k\left(1+\left|\ln\left(\frac{npd}{ek^2}\right)\right|\right)\right\}} \right]^{pd}$$

$$= \sum_{d=1}^{D} \lambda^{2d} \left[ \left(\frac{1}{2d!}\right)^{\frac{1}{pd}} (pd)^{\frac{1}{pd}} \frac{2pd}{\min\left\{ \sqrt{npd},\ k\left(1+\left|\ln\left(\frac{npd}{ek^2}\right)\right|\right)\right\}} \right]^{pd}$$

$$\leq \sum_{d=1}^{D} \lambda^{2d} \left[ \left(\frac{e}{d}\right)^{\frac{1}{p}} \frac{4pd}{\min\left\{ \sqrt{npd},\ k\left(1+\left|\ln\left(\frac{npd}{ek^2}\right)\right|\right)\right\}} \right]^{pd} \qquad (\star)$$

$$= \sum_{d=1}^{D} \lambda^{2d} \left(\frac{e}{d}\right)^{d} \frac{4^{pd}}{\min\left\{ \left(\frac{n}{pd}\right)^{pd/2},\ \left(\frac{k}{pd}\left(1+\left|\ln\left(\frac{npd}{ek^2}\right)\right|\right)\right)^{pd}\right\}}$$

$$\leq \sum_{d=1}^{D} \epsilon^{d} \left(\frac{D}{d}\right)^{d} \frac{\min\left\{ \left(\frac{n}{pD}\right)^{pd/2},\ \left(\frac{k}{pD}\left(1+\left|\ln\left(\frac{npD}{ek^2}\right)\right|\right)\right)^{pd}\right\}}{\min\left\{ \left(\frac{n}{pd}\right)^{pd/2},\ \left(\frac{k}{pd}\left(1+\left|\ln\left(\frac{npd}{ek^2}\right)\right|\right)\right)^{pd}\right\}} \qquad (*)$$

$$\leq \sum_{d=1}^{D} \epsilon^{d} \qquad (\dagger)$$

where $(\star)$ is because $\frac{1}{2d!} \leq \frac{1}{d!} \leq \left(\frac{e}{d}\right)^d$ and $(pd)^{\frac{1}{pd}} \leq 2$, $(*)$ is the theorem assumption on $\lambda$, and $(\dagger)$ is because $p \geq 2$. The statement follows since $\sum_{d=1}^{D} \epsilon^d \leq \frac{\epsilon}{1-\epsilon} \leq 2\epsilon$ for $0 \leq \epsilon \leq 1/2$. $\qquad\square$

## C.2 Low-degree distinguishing algorithm

The starting point of our distinguishing algorithm is the explicit expression from Lemma 28 and Claim 34. Assuming $D$ is even[34], we show that degree $D$ Hermite polynomials is "sufficiently large" by considering a subset of terms in the explicit summation.

*Proof of Theorem 27.* Under the assumption of $D \leq \frac{2n}{p}$ and $D$ is even, Lemma 28 and Claim 34 together tell us that

$$\sum_{|\alpha|\leq D} (\mathbb{E}_{H_1}[f_\alpha(\boldsymbol{Y})])^2 = \sum_{d=1}^{D} \lambda^{2d} k^{-pd} \frac{1}{d!} \sum_{s=1}^{\lfloor pd/2 \rfloor} \binom{n}{s}\left(\frac{k}{n}\right)^{2s} \sum_{\substack{\beta_1+\ldots+\beta_s=pd/2 \\ \beta_1\neq 0,\ldots,\beta_s\neq 0}} \binom{pd}{2\beta_1,\ldots,2\beta_s}$$

$$\geq \frac{\lambda^{2D}}{D!} k^{-pD} \sum_{s=1}^{pD/2} \binom{n}{s}\left(\frac{k}{n}\right)^{2s} \sum_{\substack{\beta_1+\ldots+\beta_s=pD/2 \\ \beta_1\neq 0,\ldots,\beta_s\neq 0}} \binom{pD}{2\beta_1,\ldots,2\beta_s}$$

$$\geq \frac{\lambda^{2D}}{D!} k^{-pD} \binom{n}{pD/2}\left(\frac{k}{n}\right)^{pD} \binom{pD}{2,\ldots,2} \qquad (\dagger)$$

$$\geq \frac{\lambda^{2D}}{D!} k^{-pD} \left(\frac{2n}{pD}\right)^{pD/2}\left(\frac{k}{n}\right)^{pD} \left(\frac{pD}{e}\right)^{pD} 2^{-pD/2} \qquad (\ddagger)$$

$$\geq \left(\frac{\lambda^2}{D}\left(\frac{kpD}{ek\sqrt{npD}}\right)^{p}\right)^{D} \qquad (\star)$$

where $(\dagger)$ is by only using $s = pD/2$ and $\beta_1 = \ldots = \beta_{pD/2} = 1$, $(\ddagger)$ is due to $\binom{n}{k} \geq (n/k)^k$ and $n! \geq (n/e)^n$, and $(\star)$ is because $D! \leq D^D$.

---

[34] For $D \geq 2$, we consider Hermite polynomials of degree either $D$ or $D-1$ (whichever is even).

When $\lambda \geq \epsilon^{\frac{1}{2D}} e^{\frac{p}{2}} \sqrt{D} \left( \frac{n}{pD} \right)^{\frac{p}{4}}$, we see that

$$\chi^2(H_1 \parallel H_0) = \sum_{|\alpha| \leq D} (\mathbb{E}_{H_1}[f_\alpha(\boldsymbol{Y})])^2 \geq \left( \frac{\lambda^2}{D} \left( \frac{kpD}{ek\sqrt{npD}} \right)^p \right)^D = \left( \frac{\lambda^2}{De^p} \left( \frac{pD}{n} \right)^{\frac{p}{2}} \right)^D \geq \epsilon$$

We now assume that $p \leq n$, $D \leq \frac{\ln^2(n/p)}{4e^2}$ and $\sqrt{np} \cdot \left( \frac{e\sqrt{D}}{\ln(n/k)} \right) \leq k \leq \sqrt{np}$. Then,

$$\left( \frac{\lambda^2}{D} \left( \frac{kpD}{ek\sqrt{npD}} \right)^p \right)^D = \left( \frac{\lambda^2}{D} \left( \frac{pD}{k\ln(\frac{n}{k})} \right)^p \left( \frac{k\ln(\frac{n}{k})}{\sqrt{e^2npD}} \right)^p \right)^D \geq \left( \frac{\lambda^2}{D} \left( \frac{pD}{k\ln(\frac{n}{k})} \right)^p \right)^D$$

where the last inequality is because $\sqrt{np} \cdot \left( \frac{e\sqrt{D}}{\ln(n/k)} \right) \leq k$. The constraints $p \leq n$ and $\sqrt{D}e \leq \frac{1}{2}\ln(n/p) = \ln(n/\sqrt{np}) \leq \ln(n/k)$ ensure that there exists valid values of $k$.

So when $\lambda \geq \epsilon^{\frac{1}{2D}} \sqrt{D} \left( \frac{k}{pD} \ln\left(\frac{n}{k}\right) \right)^{\frac{p}{2}}$, we see that

$$\chi^2(H_1 \parallel H_0) = \sum_{|\alpha| \leq D} (\mathbb{E}_{H_1}[f_\alpha(\boldsymbol{Y})])^2 \geq \left( \frac{\lambda^2}{D} \left( \frac{pD}{k\ln(\frac{n}{k})} \right)^p \right)^D \geq \epsilon$$

$\square$

# D Information-theoretic Lower Bound

In this section, we will use standard techniques[35] in information theory to lower bound the minimax risk for approximate signal recovery in the single-spike sparse tensor PCA.

**Remark** We notice that equivalent results appeared in [PWB16, NZ20]. Nevertheless we include it for completeness.

Consider the following setting: Given a single tensor observation $\boldsymbol{Y} = \boldsymbol{W} + \lambda x^{\otimes p}$ generated from an underlying signal $x \in U_t$ (i.e. the parameter of the observation is $\theta(\boldsymbol{Y}) = x$), an estimator $\widehat{\theta}(\boldsymbol{Y})$ outputs some unit vector in $\widehat{x} \in U_k$. For two vectors $x$ and $x'$, we use the pseudometric[36] $\rho(x, x') = \min\{\|x - x'\|_2, \|x + x'\|_2\}$ and the loss function $\Phi(t) = t^2/2$. Thus, $\Phi(\rho(x, x')) = 1 - |\langle x, x'\rangle|$ with the corresponding minimax risk being

$$\inf_{\widehat{\theta}} \sup_{x \in U_k} \mathbb{E}_{\boldsymbol{Y}} \left[ \Phi\left( \rho\left( \widehat{\theta}(\boldsymbol{Y}), \theta(\boldsymbol{Y}) \right) \right) \right] = \inf_{\widehat{x} \in U_k} \sup_{x \in U_k} \mathbb{E}_{\boldsymbol{Y}} \left[ 1 - |\langle \widehat{x}, x\rangle| \right]$$

Let $\mathcal{X} \subseteq U_k$ be an $\epsilon$-packing of $U_k$ of size $|\mathcal{X}| = m \geq P(U_k, \rho, \epsilon)$. Then, one can show that the minimax risk can be lower bounded as follows:

$$\inf_{\widehat{x} \in U_k} \sup_{x \in U_k} \mathbb{E}_{\boldsymbol{Y}} \left[ 1 - |\langle \widehat{x}, x\rangle| \right] \geq \frac{\epsilon^2}{4} \cdot \left( 1 - \frac{\max_{u,v \in \mathcal{X}} D_{KL}\left( \mathbb{P}_{\boldsymbol{Y} \sim \mathcal{Y}|u} \parallel \mathbb{P}_{\boldsymbol{Y} \sim \mathcal{Y}|v} \right) + 1}{\log m} \right) \quad (7)$$

where $D_{KL}(\cdot \parallel \cdot)$ is the KL-divergence function and $\mathbb{P}_{\boldsymbol{Y} \sim \mathcal{Y}|u}$ is the probability distribution of observing $\boldsymbol{Y}$ from signal $u$ with additive standard Gaussian noise tensor $\boldsymbol{W}$. The following information-theoretic lower bound is shown by lower bounding Eq. (7).

**Theorem 31** (Single-spike info-theoretic lower bound). *Given* $\boldsymbol{Y} = \boldsymbol{W} + \lambda x^{\otimes p} \in \otimes^p \mathbb{R}^n$ *where* $\boldsymbol{W}$ *is a noise tensor with i.i.d.* $N(0, 1)$ *entries and the planted signal* $x \in U_k$ *has signal strength* $\lambda$. *Let* $\widehat{x} \in U_k$ *be the recovered signal by* any *estimator. If* $n \geq 2k$ *and* $\lambda \leq \sqrt{\frac{k}{12} \log\left(\frac{n-k}{k}\right) - \frac{1}{2}}$, *then*

$$\inf_{\widehat{x} \in U_k} \sup_{x \in U_k} \mathbb{E}_{\boldsymbol{Y}} \left[ 1 - |\langle \widehat{x}, x\rangle| \right] \geq 0.05.$$

---

[35]See Appendix A.3 for a brief introduction.

[36]Instead of the "standard" $\|x - x'\|_2$ loss, we want a loss function that captures the "symmetry" that $\langle x, x'\rangle^p = \langle x, -x'\rangle^p$ for even tensor powers $p$. Clearly, $\rho(x, y) = \rho(y, x)$ and one can check that $\rho(x, y) \leq \rho(x, z) + \rho(z, y)$. Observe that $\rho$ is a pseudometric (and not a metric) because $\rho(x, y) = 0$ holds for $x = -y$.

**Remark** With $n \geq 2k$, we see that $\log(\frac{n-k}{k}) \geq \log(\frac{n}{2k})$. Then, for if $n \geq 2^{\frac{1}{1-c}} k$ for any constant $c \in (0,1)$, we see that $\log(\frac{n}{2k}) \geq c \log(\frac{n}{k})$. In particular, when $n \geq 4k$, we have $\log(\frac{n}{2k}) \geq \frac{1}{2} \log(\frac{n}{k})$ and can write $\lambda \lesssim \sqrt{k \log(n/k)}$.

To prove the result, we lower bound $m$ and upper bound the KL-divergence. Since $\|x - x'\|_2 \geq \rho(x, x')$, we see that $N(U_k, \|\cdot\|_2, \epsilon) \leq P(U_k, \|\cdot\|_2, \epsilon) \leq P(U_k, \rho, \epsilon) \leq m$. To lower bound $m$, we lower bound $N(U_k, \|\cdot\|_2, \epsilon)$ via Lemma 32. Then, we upper bound the KL-divergence in Lemma 33 by the triangle inequality and the KL-divergence of Gaussian vectors. We defer the proofs of Lemma 32 and Lemma 33 to Appendix F.4.

**Lemma 32.** *Let $U_k$ be the set of $k$-sparse flat unit vectors and $N(U_k, \|\cdot\|_2, \epsilon)$ be the $\epsilon$-covering number of $U_k$ with respect to Euclidean distance. For $\epsilon \in (0,1]$ and $n \geq 2k$,*

$$N(U_k, \|\cdot\|_2, \epsilon) \geq \left(\frac{n-k}{k}\right)^{k\left(1-\frac{\epsilon^2}{2}\right)}.$$

**Lemma 33.** *Denote $\mathcal{S}_k^{n-1}$ as the set of $k$-sparse unit vectors. Then,*

$$\max_{u,v \in \mathcal{S}_k^{n-1}} D_{KL}\left(\mathbb{P}_{\boldsymbol{Y} \sim \mathcal{Y}|u} \,\Big\|\, \mathbb{P}_{\boldsymbol{Y} \sim \mathcal{Y}|v}\right) \leq 2\lambda^2$$

*where $D_{KL}(\cdot\|\cdot)$ is the KL-divergence function and $\mathbb{P}_{\boldsymbol{Y} \sim \mathcal{Y}|u}$ is the probability distribution of observing $\boldsymbol{Y}$ from signal $u$ with additive standard Gaussian noise tensor $\boldsymbol{W}$.*

We are now ready to prove Theorem 31 by using the above two lemmata.

*Proof of Theorem 31.* The theorem follows by computing a lower bound for Eq. (7) with $\mathcal{X}$ as an $\epsilon$-packing of $U_k$ of size $|\mathcal{X}| = m \geq P(U_k, \rho, \epsilon)$.

Lemma 32 tells us that $N(U_k, \|\cdot\|_2, \epsilon) \geq \left(\frac{n-k}{k}\right)^{k\left(1-\frac{\epsilon^2}{2}\right)}$. Since $\|x - x'\|_2 \geq \rho(x, x')$, we see that $N(U_k, \|\cdot\|_2, \epsilon) \leq P(U_k, \|\cdot\|_2, \epsilon) \leq P(U_k, \rho, \epsilon) \leq m$. Thus,

$$k\left(1 - \frac{\epsilon^2}{2}\right) \log\left(\frac{n-k}{k}\right) \leq \log m$$

Meanwhile, Lemma 33 tells us that $\max_{u,v \in \mathcal{S}_k^{n-1}} D_{KL}\left(\mathbb{P}_{\boldsymbol{Y} \sim \mathcal{Y}|u} \,\big\|\, \mathbb{P}_{\boldsymbol{Y} \sim \mathcal{Y}|v}\right) \leq 2\lambda^2$. Since $U_k \subseteq \mathcal{S}_k^{n-1}$, this implies that

$$\max_{u,v \in \mathcal{X}} D_{KL}\left(\mathbb{P}_{\boldsymbol{Y} \sim \mathcal{Y}|u} \,\big\|\, \mathbb{P}_{\boldsymbol{Y} \sim \mathcal{Y}|v}\right) \leq 2\lambda^2$$

Let $\tau > 0$ be a lower bound constant (which we fix later). Putting the above bounds together, we have

$$\inf_{\widehat{x} \in U_k} \sup_{x \in U_k} \mathbb{E}_{\boldsymbol{Y}}\left[1 - |\langle \widehat{x}, x\rangle|\right] \geq \frac{\epsilon^2}{4} \cdot \left(1 - \frac{\max_{u,v \in \mathcal{X}} D_{KL}\left(\mathbb{P}_{\boldsymbol{Y} \sim \mathcal{Y}|u} \,\big\|\, \mathbb{P}_{\boldsymbol{Y} \sim \mathcal{Y}|v}\right) + 1}{\log m}\right) \qquad \text{Eq. (7)}$$

$$\geq \frac{\epsilon^2}{4} \cdot \left(1 - \frac{2\lambda^2 + 1}{k\left(1 - \frac{\epsilon^2}{2}\right)\log\left(\frac{n-k}{k}\right)}\right)$$

$$\geq \tau$$

Rearranging, we get $\lambda \leq \sqrt{\frac{(1-(4\tau)/\epsilon^2)(1-\epsilon^2/2)}{2} k \log\left(\frac{n-k}{k}\right) - \frac{1}{2}}$. The claims follows[37] by setting $\tau = 0.05$ and $\epsilon = 1/2$. $\qquad \square$

In the setting context of our interest, the works of [NWZ20, PWB+20] both papers outline a phase transition at $\lambda = \Theta(\sqrt{k \log(n/k)})$. Specifically, they prove that weak recovery is possible when $\lambda \gtrsim \sqrt{k \log(n/k)}$ and information theoretically impossible when $\lambda \lesssim \sqrt{k \log(n/k)}$. Our information theoretic bounds presented here are equivalent to these results, up to constant factors.

---

[37] Observe that $(1 - (4*0.05)/(0.5^2)) * (1 - (0.5^2)/2)/2 = 0.0875 > 1/12$.

# E    Related Model: Planted sparse densest sub-hypergraph

The planted $k$-densest sub-hypergraph model ([CPMB19, BCPS20, CPSB20]) is closely related to our sparse spiked tensor model Model 4. While not directly reducible from/to one another, techniques developed in one model can inform the other.

The planted $k$-densest sub-hypergraph model is a weighted complete hypergraph where a subset of $k$ planted vertices, denoted by $S \subseteq [n]$, is *drawn uniformly at random* and each hyperedge involves $p$ vertices, for $2 \le p \le k \le n$. Except for the $\binom{k}{p}$ *one-sided biased* hyperedges (belonging to the planted subgraph induced by the $k$ vertices in $S$) whose weights follow the Gaussian distribution $N(\beta, \sigma^2)$, the weight of all remaining $\binom{n}{p} - \binom{k}{p}$ hyperedges follow a Gaussian distribution $N(0, \sigma^2)$. In other words, the hyperedge defined by $\{i_1, \ldots, i_p\}$, for $i_1, \ldots, i_p \in [n]$, has weight

$$\boldsymbol{Y}_{i_1,\ldots,i_p} = \begin{cases} \beta + \boldsymbol{W}_{i_1,\ldots,i_p} & \text{if } i_1, \ldots, i_p \in S \\ \boldsymbol{W}_{i_1,\ldots,i_p} & \text{otherwise} \end{cases}$$

where each $\boldsymbol{W}_{i_1,\ldots,i_p} \sim N(0, \sigma^2)$ is independent and planted entries have a $\beta > 0$ bias.

As one can see, the planted $k$-densest sub-hypergraph model (PDSM) is very similar to the single-spike ($r = 1$) sparse spiked tensor model (SSTM) that we study[38]. However, there are two key model differences that one needs to be aware of. Firstly, there are $n^p$ observations in SSTM instead of $\binom{n}{p}$ in PDSM as the former is not constrained to hyperedges (e.g. $\boldsymbol{Y}_{1,\ldots,1}$ is a valid data observation in SSTM but not in PDSM). Secondly, our signal bias is not one-sided and is scaled by a factor of $k^{-p/2}$: For a planted coordinate $(i_1, \ldots, i_p)$, SSTM observes $\boldsymbol{Y}_{i_1,\ldots,i_p} = \boldsymbol{W}_{i_1,\ldots,i_p} \pm \lambda k^{-p/2}$ instead of $\boldsymbol{Y}_{i_1,\ldots,i_p} = \boldsymbol{W}_{i_1,\ldots,i_p} + \beta$ in PDSM, where the sign of bias in SSTM depends on the polarities of signal entries $x_{i_1}, \ldots, x_{i_p}$.

While the signal scaling discrepancies can be handled by replacing $\beta\sqrt{\binom{k}{p}}$ terms in PDSM bounds with $\lambda$ in SSTM[39], the one-sidedness of the signal bias has implications on the computational hardness of the two models. In a recent work, [CPSB20] showed that an Approximate Message Passing (AMP) algorithm succeeds in signal recovery for PDSM when $\lambda \gtrsim \frac{k}{p\sqrt{n}} \left(\frac{n}{k}\right)^{p/2}$. In contrast, Theorem 2 (for constant $p$) tells us that our polynomial time algorithm for SSTM succeeds when $k \le \sqrt{np}$ and $\lambda \gtrsim \sqrt{k^p \log\left(\frac{np}{k}\right)}$. Meanwhile, Theorem 3 implies that it is impossible to recover the signal using low-degree polynomials whenever the signal-to-noise ratio satisfies $\lambda \lesssim \widetilde{O}\left(\min\left\{\left(\frac{n}{p}\right)^{p/4}, \left(\frac{k}{p}\left(1 + \left|\ln\left(\frac{np}{k^2}\right)\right|\right)\right)^{p/2}\right\}\right)$. Indeed, the one-sidedness of the signal bias in PDSM appears to make the problem *computationally easier* than SSTM in some regimes[40].

Nevertheless, we believe that techniques used in either model are generally applicable in the other and we expect a variant of our limited brute force algorithm to work in PDSM. From a statistical viewpoint, [CPSB20] proved information-theoretic lower bounds for recovery in PDSM of $\lambda \lesssim \sqrt{k \log n}$ while we have $\lambda \lesssim \sqrt{k \log((n-k)/k)}$ for approximate signal recovery[41] in SSTM, which matches the PDSM bounds when $k \in o(n)$. These results in both models rely on standard techniques such as Fano's inequality.

---

[38]We believe that handling a more general $\sigma^2$ is a non-issue when comparing these models because the $\sigma^2$ factor could be propagated throughout our analysis by appropriately adjusting the sub-Gaussian concentration arguments.

[39]This discrepancy arises due to having $\binom{k}{p}$ planted hyperedges in PDSM, as opposed to $k^p$ signal entries in SSTM, and the signal strength scaling of $k^{-p/2}$ in SSTM.

[40]e.g. Large $k$ regimes such as $k = n^{0.9}$. For large $k$, a heuristic adaptation of our low-degree analysis to PDSM shows that the relationship between parameters $\lambda$, $n$, $k$ and $p$ in a low-degree bound is roughly of the form $\lambda \gtrsim \left(\frac{n}{k}\right)^{p/2}$ instead of $\lambda \gtrsim n^{p/4}$. This roughly matches the AMP bounds shown by [CPSB20] and further provides credence to our claim that techniques from one model can applied to the other.

[41]i.e. It is enough to find a strongly correlated estimate $\widehat{x}$ of the signal $x$ where $\widehat{x}$ could be "wrong on a few coordinates". The bounds for exact and approximate recovery in [CPSB20] differ by constant factors.

# F Deferred proofs and details

This section provides the formal proofs that were deferred in favor for readability. For convenience, we will restate the statements before proving them.

## F.1 Sparse norm bounds

**Lemma 20** (Tensor sparse bound). *Let $\boldsymbol{T} \in \otimes^p \mathbb{R}^n$ be an order $p \geq 2$ tensor with i.i.d. standard Gaussian entries and $x_{(1)}, \ldots, x_{(p)} \in \mathcal{S}_s^{n-1}$ be $r$ distinct (at most) $s$-sparse unit vectors. Then, for $1 \leq s \leq n$, $1 \leq r \leq p$, and $\gamma \in (0, 1)$,*

$$\mathbb{P}\left[\left|\boldsymbol{T}(x_{(1)}, \ldots, x_{(p)})\right| \geq \sqrt{8 \cdot \left(4rs \ln\left(\frac{np}{s}\right) + \ln\left(\frac{1}{\gamma}\right)\right)}\right] \leq 2\gamma$$

*Proof of Lemma 20.* We will focus on proving the following statement:

$$\mathbb{P}\left[\max_{\substack{x_{(1)}, \ldots, x_{(p)} \in \mathcal{S}_s^{n-1}, \\ r \text{ distinct vectors}}} \boldsymbol{T}(x_{(1)}, \ldots, x_{(p)}) \geq \sqrt{8 \cdot \left(4rs \ln\left(\frac{np}{s}\right) + \ln\left(\frac{1}{\gamma}\right)\right)}\right] \leq \gamma \qquad (8)$$

By a similar argument, one can obtain

$$\mathbb{P}\left[\min_{\substack{x_{(1)}, \ldots, x_{(p)} \in \mathcal{S}_s^{n-1}, \\ r \text{ distinct vectors}}} \boldsymbol{T}\left(x_{(1)}, \ldots, x_{(p)}\right) \leq -\sqrt{8 \cdot \left(4rs \ln\left(\frac{np}{s}\right) + \ln\left(\frac{1}{\gamma}\right)\right)}\right] \leq \gamma \qquad (9)$$

The lemma follows from a union bound of Eq. (8) and Eq. (9).

It now remains to prove Eq. (8). Let $\lambda, t, \epsilon$ be proof parameters which we fix later. Define $\mathcal{N}(\mathcal{S}_s^{n-1})$ as an $\epsilon$-cover of $\mathcal{S}_s^{n-1}$ of smallest cardinality $N(\mathcal{S}_s^{n-1}, \epsilon)$. It is known[42] that $\left(\frac{1}{\epsilon}\right)^n \leq N(\mathcal{S}^{n-1}, \epsilon) \leq \left(\frac{2}{\epsilon} + 1\right)^n \leq \left(\frac{3}{\epsilon}\right)^n$. Treating each unit sphere defined on $s$ coordinates independently and then taking union bound gives us $N(\mathcal{S}_s^{n-1}, \epsilon) \leq \binom{n}{s} \cdot N(\mathcal{S}^{n-1}, \epsilon) \leq \left(\frac{en}{s}\right)^s \left(\frac{3}{\epsilon}\right)^s$. So, $\left|\mathcal{N}(\mathcal{S}_s^{n-1})\right| \leq \left(\frac{3en}{\epsilon s}\right)^s$. For any $r$ distinct vectors $x_{(1)}, \ldots, x_{(p)} \in \mathcal{N}(\mathcal{S}_s^{n-1})$,

$$\mathbb{P}\left[\boldsymbol{T}(x_{(1)}, \ldots, x_{(p)}) \geq t\right]$$

$$= \mathbb{P}\left[\sum_{i_1, \ldots, i_p = 1}^{n} \boldsymbol{T}_{i_1, \ldots, i_p} x_{(1), i_1} x_{(2), i_2} \cdots x_{(p), i_p} \geq t\right]$$

$$= \mathbb{P}\left[\exp\left(\lambda \sum_{i_1, \ldots, i_p = 1}^{n} \boldsymbol{T}_{i_1, \ldots, i_p} x_{(1), i_1} x_{(2), i_2} \cdots x_{(p), i_p}\right) \geq e^{\lambda t}\right]$$

$$\leq e^{-\lambda t} \cdot \mathbb{E}\left[\exp\left(\lambda \sum_{i_1, \ldots, i_p = 1}^{n} \boldsymbol{T}_{i_1, \ldots, i_p} x_{(1), i_1} x_{(2), i_2} \cdots x_{(p), i_p}\right)\right] \qquad \text{Markov's inequality}$$

$$= e^{-\lambda t} \cdot \Pi_{i_1, \ldots, i_p = 1}^{n} \mathbb{E}\left[\exp\left(\boldsymbol{T}_{i_1, \ldots, i_p} \lambda x_{(1), i_1} x_{(2), i_2} \cdots x_{(p), i_p}\right)\right]$$

$$= e^{-\lambda t} \cdot \Pi_{i_1, \ldots, i_p = 1}^{n} \exp\left(\frac{\left(\lambda x_{(1), i_1} x_{(2), i_2} \cdots x_{(p), i_p}\right)^2}{2}\right) \qquad \boldsymbol{T}_{i_1, \ldots, i_p} \sim N(0, 1)$$

$$= \exp\left(-\lambda t + \sum_{i_1, \ldots, i_p = 1}^{n} \frac{\left(\lambda x_{(1), i_1} x_{(2), i_2} \cdots x_{(p), i_p}\right)^2}{2}\right)$$

$$= \exp\left(-\lambda t + \frac{\lambda^2}{2}\right) \qquad \sum_{i_1 = 1}^{n} x_{(1), i_1}^2 = \ldots = \sum_{i_p = 1}^{n} x_{(p), i_p}^2 = 1$$

$$\leq \exp\left(-\frac{t^2}{2}\right) \qquad \text{Maximized when } \lambda = t$$

---

[42] e.g. See [Ver18, Corollary 4.2.13].

By union bound over all $\left(N(\mathcal{S}_s^{n-1}, \epsilon)\right)^r \leq \left(\frac{3en}{\epsilon s}\right)^{rs}$ $r$ distinct points in $\otimes^p \mathcal{N}(\mathcal{S}_s^{n-1})$,

$$\mathbb{P}\left[\max_{\substack{x_{(1)},\ldots,x_{(p)} \in \mathcal{N}(\mathcal{S}_s^{n-1}), \\ r \text{ distinct vectors}}} \boldsymbol{T}(x_{(1)},\ldots,x_{(p)}) \geq t\right] \leq \sum_{\substack{x_{(1)},\ldots,x_{(p)} \in \mathcal{N}(\mathcal{S}_s^{n-1}), \\ r \text{ distinct vectors}}} \mathbb{P}\left[\boldsymbol{T}(x_{(1)},\ldots,x_{(p)}) \geq t\right]$$

$$\leq \left(\frac{3en}{\epsilon s}\right)^{rs} \exp\left(-\frac{t^2}{2}\right)$$

$$= \exp\left(rs \ln\left(\frac{3en}{\epsilon s}\right) - \frac{t^2}{2}\right)$$

As $\otimes^p \mathcal{S}_s^{n-1}$ is compact, there are $r$ distinct vectors $x_{(1)}^*, x_{(2)}^*, \ldots, x_{(p)}^* \in \mathcal{S}_s^{n-1}$ such that

$$\left(x_{(1)}^*, x_{(2)}^*, \ldots, x_{(p)}^*\right) = \operatorname{argmax}_{\substack{x_{(1)},\ldots,x_{(p)} \in \mathcal{S}_s^{n-1}, \\ r \text{ distinct vectors}}} \boldsymbol{T}\left(x_{(1)},\ldots,x_{(p)}\right)$$

By definition of $\epsilon$-cover, there are vectors $x_{(1)}, x_{(2)}, \ldots, x_{(p)} \in \mathcal{N}(\mathcal{S}_s^{n-1})$ such that $x_{(1)}^* = x_{(1)} + \delta_{(1)}, \ldots, x_{(p)}^* = x_{(p)} + \delta_{(p)}$, where $\left\|\delta_{(z)}\right\|_2 \leq \epsilon$ for $z \in \{1,\ldots,p\}$. Let $z \in \{1,\ldots,p\}$. Since $x_{(z)}^*$ and $x_{(z)}$ are $s$-sparse, $\delta_{(z)}$ is at most $(2s)$-sparse. We can express $\delta_{(z)} = \delta_{(z)}^{(1)} + \delta_{(z)}^{(2)}$ as a sum of two $s$-sparse vectors where $\frac{\delta_{(z)}^{(1)}}{\left\|\delta_{(z)}^{(1)}\right\|_2}, \frac{\delta_{(z)}^{(2)}}{\left\|\delta_{(z)}^{(2)}\right\|_2} \in \mathcal{S}_s^{n-1}$, $\left\|\delta_{(z)}^{(1)}\right\|_2 \leq \left\|\delta_{(z)}\right\|_2 \leq \epsilon$, and $\left\|\delta_{(z)}^{(2)}\right\|_2 \leq \left\|\delta_{(z)}\right\|_2 \leq \epsilon$. We can relate $\boldsymbol{T}(x_{(1)}^*, x_{(2)}^*, \ldots, x_{(p)}^*)$ to $\boldsymbol{T}(x_{(1)}, x_{(2)}, \ldots, x_{(p)})$ by expanding the definition:

$$\max_{\substack{x_{(1)},\ldots,x_{(p)} \in \mathcal{S}_s^{n-1}, \\ r \text{ distinct vectors}}} \boldsymbol{T}(x_{(1)}, x_{(2)}, \ldots, x_{(p)})$$

$$= \sum_{i_1,\ldots,i_p=1}^{n} \boldsymbol{T}_{i_1,\ldots,i_p} x_{(1),i_1}^* x_{(2),i_2}^* \cdots x_{(p),i_p}^*$$

$$= \sum_{i_1,\ldots,i_p=1}^{n} \boldsymbol{T}_{i_1,\ldots,i_p} \left(x_{(1)} + \delta_{(1)}^{(1)} + \delta_{(1)}^{(2)}\right)_{i_1} \left(x_{(2)} + \delta_{(2)}^{(1)} + \delta_{(2)}^{(2)}\right)_{i_2} \cdots \left(x_{(p)} + \delta_{(p)}^{(1)} + \delta_{(p)}^{(2)}\right)_{i_p}$$

$$\leq \boldsymbol{T}(x_{(1)},\ldots,x_{(p)}) + \left(\max_{\substack{x_{(1)},\ldots,x_{(p)} \in \mathcal{S}_s^{n-1}, \\ r \text{ distinct vectors}}} \boldsymbol{T}(x_{(1)},\ldots,x_{(p)})\right) \cdot \left(\epsilon \cdot 2\binom{p}{1} + \epsilon^2 \cdot 2^2\binom{p}{2} + \ldots + \epsilon^p 2^p \binom{p}{p}\right)$$

$$\leq \boldsymbol{T}(x_{(1)},\ldots,x_{(p)}) + \left(\max_{\substack{x_{(1)},\ldots,x_{(p)} \in \mathcal{S}_s^{n-1}, \\ r \text{ distinct vectors}}} \boldsymbol{T}(x_{(1)},\ldots,x_{(p)})\right) \cdot \left(\frac{2p\epsilon}{1!} + \frac{(2p\epsilon)^2}{2!} + \ldots + \frac{(2p\epsilon)^p}{p!}\right)$$

$$\leq \boldsymbol{T}(x_{(1)},\ldots,x_{(p)}) + \left(\max_{\substack{x_{(1)},\ldots,x_{(p)} \in \mathcal{S}_s^{n-1}, \\ r \text{ distinct vectors}}} \boldsymbol{T}(x_{(1)},\ldots,x_{(p)})\right) \cdot \left(e^{2p\epsilon} - 1\right)$$

The first inequality is by counting how the $\delta$'s group together, factoring out their norms so that they belong to $\mathcal{S}_s^{n-1}$ (so $\max_{(\ldots)} \boldsymbol{T}\left(x_{(1)},\ldots,x_{(p)}\right)$ applies), then using $\left\|\delta_{(z)}^{(1)}\right\|_2, \left\|\delta_{(z)}^{(2)}\right\|_2 \leq \epsilon$. The second inequality is due to $\binom{n}{k} \leq \frac{n^k}{k!}$. The third is due to the definition of $e^x = \sum_{n=0}^{\infty} \frac{x^n}{n!}$. Note that $1 > e^{2p\epsilon} - 1$ if and only if $\epsilon < \frac{\ln 2}{2p}$. Set $\epsilon = \frac{\ln 2}{4p}$, then $\frac{1}{2 - e^{2p\epsilon}} < 2$. Rearranging, we get

$$\max_{\substack{x_{(1)},\ldots,x_{(p)} \in \mathcal{S}_s^{n-1}, \\ r \text{ distinct vectors}}} \boldsymbol{T}\left(x_{(1)}, x_{(2)}, \ldots, x_{(p)}\right) \leq \frac{\boldsymbol{T}\left(x_{(1)}, x_{(2)}, \ldots, x_{(p)}\right)}{2 - e^{2p\epsilon}}$$

$$\leq 2 \max_{\substack{x_{(1)},\ldots,x_{(p)} \in \mathcal{N}(\mathcal{S}_s^{n-1}), \\ r \text{ distinct vectors}}} \boldsymbol{T}\left(x_{(1)}, x_{(2)}, \ldots, x_{(p)}\right)$$

Thus,

$$\mathbb{P}\left[\max_{\substack{x_{(1)},\ldots,x_{(p)}\in\mathcal{S}_s^{n-1},\\ r\text{ distinct vectors}}}\boldsymbol{T}\left(x_{(1)},x_{(2)},\ldots,x_{(p)}\right)\geq t\right]$$

$$\leq\mathbb{P}\left[\max_{\substack{x_{(1)},\ldots,x_{(p)}\in\mathcal{N}(\mathcal{S}_s^{n-1}),\\ r\text{ distinct vectors}}}\boldsymbol{T}\left(x_{(1)},x_{(2)},\ldots,x_{(p)}\right)\geq\frac{t}{2}\right]$$

$$\leq\exp\left(rs\ln\left(\frac{3en}{\epsilon s}\right)-\frac{(t/2)^2}{2}\right)\qquad\qquad\text{From above}$$

$$\leq\exp\left(4rs\ln\left(\frac{np}{s}\right)-\frac{t^2}{8}\right)\qquad\qquad\text{Since }\epsilon=\frac{\ln 2}{4p}\text{ and }\ln\left(\frac{12e}{\ln 2}\right)<4$$

Setting $t^2=8\cdot\left(4rs\ln\left(\frac{np}{s}\right)+\ln\left(\frac{1}{\gamma}\right)\right)$ yields Eq. (8). $\qquad\square$

## F.2  Proofs for recovery algorithms

**Lemma 23.** *Consider Model 4 and an arbitrary round $i\in[r]$. Suppose*

$$\lambda_r\geq\frac{32\kappa}{(A\epsilon)^p}\sqrt{t\left(\frac{k}{t}\right)^p\ln(n)},\quad\lambda_r\geq\kappa\cdot\lambda_1,\quad\text{and}\quad\kappa\geq 5A^{2p}\left(\frac{\epsilon}{1-\epsilon}\right)^{p-1}.$$

*Then,*

$$\mathbb{P}\left[\left|\mathrm{supp}\left(\boldsymbol{v}_*\right)\cap\mathrm{supp}\left(x_{(\pi(i))}\right)\right|\geq(1-\epsilon)\cdot t\right]\geq 1-4\exp\left(-\lambda_r^2\frac{\epsilon^2}{128A^4}\left(\frac{t}{k}\right)^p\right)$$

*Proof of Lemma 23.* Without loss of generality, suppose that $x_{(1)},\ldots,x_{(s)}$ are the remaining $s$ (where $1\leq s\leq r$) unrecovered signals with signal strengths $\lambda_1,\ldots,\lambda_s$ such that $\lambda_1\geq\ldots\geq\lambda_s\geq\lambda_r$. Let $u_*\in U_t$ lie completely in some signal $\hat{x}$ with signal strength $\hat{\lambda}$ such that

$$\hat{\lambda}\langle\hat{x},u_*\rangle^p\geq\max_{q\in[s]}\max_{u\in U_t}\lambda_q\langle x_{(q)},u\rangle^p\quad\text{and}\quad\langle\boldsymbol{Y}^{(1)},u_*^{\otimes p}\rangle\geq\frac{\hat{\lambda}}{\sqrt{2}}\langle\hat{x},u_*\rangle^p+\langle\boldsymbol{W}^{(1)},u_*^{\otimes p}\rangle.$$

By optimality, $\langle\boldsymbol{Y}^{(1)},\boldsymbol{v}_*^{\otimes p}\rangle\geq\langle\boldsymbol{Y}^{(1)},u_*^{\otimes p}\rangle$. So, the claim holds if we can show that $\langle\boldsymbol{Y}^{(1)},u_*^{\otimes p}\rangle>\langle\boldsymbol{Y}^{(1)},u^{\otimes p}\rangle$ for *any* $u\in U_t$ such that

$$\left|\mathrm{supp}\left(u\right)\cap\mathrm{supp}\left(x_{(1)}\right)\right|<(1-\epsilon)\cdot t,\ldots,\left|\mathrm{supp}\left(u\right)\cap\mathrm{supp}\left(x_{(s)}\right)\right|<(1-\epsilon)\cdot t.\qquad(10)$$

For any $u \in U_t$ that satisfies Eq. (10), we see that

$$\langle \boldsymbol{Y}^{(1)}, u^{\otimes p} \rangle$$

$$= \langle \boldsymbol{W}^{(1)}, u^{\otimes p} \rangle + \sum_{q=1}^{s} \frac{\lambda_q}{\sqrt{2}} \langle u, x_{(q)} \rangle^p$$

$$\leq \langle \boldsymbol{W}^{(1)}, u^{\otimes p} \rangle + \frac{\hat{\lambda}}{\sqrt{2}} \left( \langle \hat{x}, u_* \rangle - \frac{\epsilon}{A} \sqrt{\frac{t}{k}} \right)^p + \frac{\lambda_1 \epsilon^p A^p}{\sqrt{2}} \left( \frac{t}{k} \right)^{\frac{p}{2}} \qquad \textcolor{blue}{\text{Eq. (10)}}$$

$$\leq \langle \boldsymbol{W}^{(1)}, u^{\otimes p} \rangle + \frac{\hat{\lambda}}{\sqrt{2}} \left( \left( \langle \hat{x}, u_* \rangle - \frac{\epsilon}{A} \sqrt{\frac{t}{k}} \right)^p + \frac{\epsilon^p A^p}{\kappa} \left( \frac{t}{k} \right)^{\frac{p}{2}} \right) \qquad \hat{\lambda} \geq \lambda_r \geq \kappa \lambda_1$$

$$\leq \langle \boldsymbol{W}^{(1)}, u^{\otimes p} \rangle + \frac{\hat{\lambda}}{\sqrt{2}} \langle \hat{x}, u_* \rangle^p \left( \left( 1 - \frac{\epsilon}{A^2} \right)^p + \frac{\epsilon^p A^p}{\kappa} \right) \qquad \sqrt{\frac{t}{k}} \leq \langle \hat{x}, u_* \rangle \leq A \sqrt{\frac{t}{k}}$$

$$\leq \langle \boldsymbol{W}^{(1)}, u^{\otimes p} \rangle + \frac{\hat{\lambda}}{\sqrt{2}} \langle \hat{x}, u_* \rangle^p \left( \left( 1 - \frac{\epsilon}{A^2} \right)^p + \frac{\epsilon(1-\epsilon)^{p-1}}{A^p} \right) \qquad \kappa \geq A^{2p} \left( \frac{\epsilon}{1-\epsilon} \right)^{p-1}$$

$$\leq \langle \boldsymbol{W}^{(1)}, u^{\otimes p} \rangle + \frac{\hat{\lambda}}{\sqrt{2}} \langle \hat{x}, u_* \rangle^p \left( 1 - \frac{\epsilon}{A^2} \right)^{p-1} \qquad A \geq 1, \epsilon \leq \frac{1}{2}$$

Let us set parameters $(r, s, \gamma)$ as $\left( 1, t, \exp\left( -\frac{\lambda_r^2 \epsilon^2}{128 A^4} \left( \frac{t}{k} \right)^p \right) \right)$ in Lemma 20. Since

$$\frac{\lambda_r^2 \epsilon^2}{128 A^4} \left( \frac{t}{k} \right)^p \geq 8t \ln(n) \geq 4t \ln\left( \frac{np}{t} \right),$$

we see that

$$\frac{\lambda_r \epsilon}{2\sqrt{2}A} \left( \frac{t}{k} \right)^{\frac{p}{2}} \geq \sqrt{8 \left( 4t \ln\left( \frac{np}{t} \right) + \ln\left( \frac{1}{\gamma} \right) \right)}.$$

Thus, Lemma 20 gives us that, for any $u \in U_t$,

$$\mathbb{P}\left[ \max_{u \in U_t} \left| \langle \boldsymbol{W}^{(1)}, u^{\otimes p} \rangle \right| \geq \frac{\lambda_r \epsilon}{2\sqrt{2}A^2} \left( \frac{t}{k} \right)^{\frac{p}{2}} \right] \leq 2\exp\left( -\frac{\lambda_r^2 \epsilon^2}{128 A^4} \left( \frac{t}{k} \right)^p \right).$$

Then, with probability at least $1 - 4\exp\exp\left( -\frac{\lambda_r^2 \epsilon^2}{128 A^4} \left( \frac{t}{k} \right)^p \right)$,

$$\langle \boldsymbol{W}^{(1)}, u_*^{\otimes p} \rangle - \langle \boldsymbol{W}^{(1)}, u^{\otimes p} \rangle < \frac{\hat{\lambda}}{\sqrt{2}} \langle \hat{x}, u_* \rangle^p \frac{\epsilon}{A^2} \leq \frac{\hat{\lambda}}{\sqrt{2}} \langle \hat{x}, u_* \rangle^p \left( 1 - \left( 1 - \frac{\epsilon}{A^2} \right)^{p-1} \right).$$

and so $\langle \boldsymbol{Y}^{(1)}, \boldsymbol{v}_*^{\otimes p} \rangle > \langle \boldsymbol{Y}^{(1)}, u_*^{\otimes p} \rangle > \langle \boldsymbol{Y}^{(1)}, u^{\otimes p} \rangle$. for any $u \in U_t$ that satisfies Eq. (10). $\qquad \square$

**Lemma 24.** *Consider Model 4 and an arbitrary round $i \in [r]$. Suppose*

$$\lambda_r \geq \frac{32\kappa}{(A\epsilon)^p} \sqrt{t \left( \frac{k}{t} \right)^p \ln(n)}, \quad \lambda_r \geq \kappa \cdot \lambda_1, \quad and \quad \kappa \geq 5A^{2p} \left( \frac{\epsilon}{1-\epsilon} \right)^{p-1}.$$

*Further suppose that* $\left| \operatorname{supp}(\boldsymbol{v}_*) \cap \operatorname{supp}(x_{(\pi(i))}) \right| \geq (1 - \epsilon) \cdot t$. *Then, the largest (in magnitude) coordinates of $\boldsymbol{\alpha}$ are* $\operatorname{supp}(x_{(\pi(i))})$ *with probability at least* $1 - 2n \exp\left( -\lambda_r^2 \frac{A^{2p} \epsilon^{2p-2}}{16\kappa^2 t} \left( \frac{t}{k} \right)^p \right)$.

*Proof of Lemma 24.* Recall that

$$\boldsymbol{\alpha}_\ell = \sum_{q \in [r]} \frac{\lambda_q}{\sqrt{2}} x_{(q),\ell} \langle x_{(q)}, \boldsymbol{v}_* \rangle^{p-1} + \langle \boldsymbol{W}^{(2)}, \boldsymbol{v}_*^{\otimes p-1} \otimes e_\ell \rangle.$$

Since $\boldsymbol{W}^{(2)}$ is independent from $\boldsymbol{W}^{(1)}$, we can apply standard Gaussian bounds. That is,

$$\mathbb{P}\left[\left|\langle \boldsymbol{W}^{(2)}, \boldsymbol{v}_*^{\otimes p-1} \otimes e_\ell\rangle\right| \geq \lambda_r \frac{A^p \epsilon^{p-1}}{2\kappa\sqrt{2t}} \left(\frac{t}{k}\right)^{\frac{p}{2}}\right] \leq 2\exp\left(-\lambda_r^2 \frac{A^{2p}\epsilon^{2p-2}}{16\kappa^2 t} \left(\frac{t}{k}\right)^p\right).$$

Now, conditioned on

$$\left|\langle \boldsymbol{W}^{(2)}, \boldsymbol{v}_*^{\otimes p-1} \otimes e_\ell\rangle\right| < \lambda_r \frac{A^p \epsilon^{p-1}}{2\kappa\sqrt{2k}} \left(\frac{t}{k}\right)^{\frac{p-1}{2}} < \lambda_r \frac{(1-\epsilon)^{p-1}}{2A^p\sqrt{2k}} \left(\frac{t}{k}\right)^{\frac{p-1}{2}},$$

we consider cases of $\ell \in \operatorname{supp}\left(x_{(\pi(i))}\right)$ and $\ell \notin \operatorname{supp}\left(x_{(\pi(i))}\right)$ separately. To be precise, we will show the following two results:

1. $\mathbb{P}\left[|\boldsymbol{\alpha}_\ell| < \lambda_r \frac{(1-\epsilon)^{p-1}}{2A^p\sqrt{2k}} \left(\frac{t}{k}\right)^{\frac{p-1}{2}} \,\Big|\, \ell \in \operatorname{supp}\left(x_{(\pi(i))}\right)\right] \leq 2\exp\left(-\lambda_r^2 \frac{A^{2p}\epsilon^{2p-2}}{16\kappa^2 t} \left(\frac{t}{k}\right)^p\right)$

2. $\mathbb{P}\left[|\boldsymbol{\alpha}_\ell| > \lambda_r \frac{2A^p\epsilon^{p-1}}{\kappa\sqrt{2k}} \left(\frac{t}{k}\right)^{\frac{p-1}{2}} \,\Big|\, \ell \notin \operatorname{supp}\left(x_{(\pi(i))}\right)\right] \leq 2\exp\left(-\lambda_r^2 \frac{A^{2p}\epsilon^{2p-2}}{16\kappa^2 t} \left(\frac{t}{k}\right)^p\right)$

As $\kappa > 4A^{2p}\left(\frac{\epsilon}{1-\epsilon}\right)^{p-1}$, there will be a value gap in $|\boldsymbol{\alpha}_\ell|$ for $\ell \in \operatorname{supp}\left(x_{(\pi(i))}\right)$ versus $\ell \notin \operatorname{supp}\left(x_{(\pi(i))}\right)$. The result follows by taking a union bound over all $n$ coordinates.

**Case 1** $\left(\ell \in \operatorname{supp}\left(x_{(\pi(i))}\right)\right)$: Since $\left|\operatorname{supp}\left(\boldsymbol{v}_*\right) \cap \operatorname{supp}\left(x_{(\pi(i))}\right)\right| \geq (1-\epsilon)\cdot t$,

$$\left|\frac{\lambda_{\pi(i)}}{\sqrt{2}} \cdot x_{(\pi(i)),\ell} \cdot \langle \boldsymbol{v}_*, x_{(\pi(i))}\rangle^{p-1}\right| \geq \frac{\lambda_r}{A\sqrt{2k}} \cdot \left|\langle \boldsymbol{v}_*, x_{(\pi(i))}\rangle^{p-1}\right| \geq \lambda_r \frac{(1-\epsilon)^{p-1}}{A^p\sqrt{2k}} \left(\frac{t}{k}\right)^{\frac{p-1}{2}}.$$

By reverse triangle inequality, we have

$$|\boldsymbol{\alpha}_\ell| = \lambda_r \frac{(1-\epsilon)^{p-1}}{A^p\sqrt{2k}} \left(\frac{t}{k}\right)^{\frac{p-1}{2}} - \left|\langle \boldsymbol{W}^{(2)}, \boldsymbol{v}_*^{\otimes p-1} \otimes e_\ell\rangle\right| > \lambda_r \frac{(1-\epsilon)^{p-1}}{2A^p\sqrt{2k}} \left(\frac{t}{k}\right)^{\frac{p-1}{2}}.$$

**Case 2** $\left(\ell \notin \operatorname{supp}\left(x_{(\pi(i))}\right)\right)$: Since signals have disjoint support and $\left|\operatorname{supp}\left(\boldsymbol{v}_*\right) \cap \operatorname{supp}\left(x_{(\pi(i))}\right)\right| \geq (1-\epsilon)\cdot t$, we have $\left|\operatorname{supp}\left(\boldsymbol{v}_*\right) \cap \operatorname{supp}\left(x_{(j)}\right)\right| < \epsilon \cdot t$.
By triangle inequality, we have

$$|\boldsymbol{\alpha}_\ell| \leq \lambda_1 \frac{A^p\epsilon^{p-1}}{\sqrt{2k}} \left(\frac{t}{k}\right)^{\frac{p-1}{2}} + \left|\langle \boldsymbol{W}^{(2)}, \boldsymbol{v}_*^{\otimes p-1} \otimes e_\ell\rangle\right|$$

$$\leq \lambda_r \frac{A^p\epsilon^{p-1}}{\kappa\sqrt{2k}} \left(\frac{t}{k}\right)^{\frac{p-1}{2}} + \left|\langle \boldsymbol{W}^{(2)}, \boldsymbol{v}_*^{\otimes p-1} \otimes e_\ell\rangle\right|$$

$$\leq \lambda_r \frac{2A^p\epsilon^{p-1}}{\kappa\sqrt{2k}} \left(\frac{t}{k}\right)^{\frac{p-1}{2}}.$$

$\square$

### F.3 Proofs for computational bounds

Claim 34 relates the counting of $\boldsymbol{Y}$ entries with coordinates of the signal $\boldsymbol{x}$. In the claim, $s \in [n]$ is the number of entries of $\boldsymbol{x}$ that is considered in the summation. We only need to consider $s$ up to $\lfloor pd/2 \rfloor$ because the expectation is 0 if some coordinate of $\boldsymbol{x}$ is used an odd number of times. Each $\alpha$ can be viewed as $d$ consecutive chunks of $p$ entries, and each $(\beta_1, \ldots, \beta_s)$ counts the number of times $\boldsymbol{x}_j$ occurs in $\alpha$.

**Claim 34.** *For a fixed degree $d \leq 2n/p$,*

$$\sum_{|\alpha|=d} \mathbb{1}_{even(c(\alpha))} \left(\frac{k}{n}\right)^{2s(\alpha)} \left(\Pi_{i=1}^{n^p} \frac{1}{(\alpha_i)!}\right)$$

$$= \sum_{s=1}^{\lfloor pd/2 \rfloor} \binom{n}{s} \left(\frac{k}{n}\right)^{2s} \sum_{\substack{\beta_1+\ldots+\beta_s=pd/2 \\ \beta_1 \neq 0,\ldots,\beta_s \neq 0}} \binom{pd}{2\beta_1,\ldots,2\beta_s} \frac{1}{\binom{d}{\alpha_1,\ldots,\alpha_{n^p}}} \left(\Pi_{i=1}^{n^p} \frac{1}{(\alpha_i)!}\right)$$

$$= \frac{1}{d!} \sum_{s=1}^{\lfloor pd/2 \rfloor} \binom{n}{s} \left(\frac{k}{n}\right)^{2s} \sum_{\substack{\beta_1+\ldots+\beta_s=pd/2 \\ \beta_1 \neq 0,\ldots,\beta_s \neq 0}} \binom{pd}{2\beta_1,\ldots,2\beta_s}$$

*Proof.* The second equality is by definition of multinomial coefficients. To prove the first equality, we consider two equivalent ways of viewing the summation.

From the viewpoint of choosing entries of $\boldsymbol{Y}$, one chooses $d$ (possibly repeated) entries of $\boldsymbol{Y}$ and computes $\mathbb{1}_{even(c(\alpha))} \left(\frac{k}{n}\right)^{2s(\alpha)} \left(\Pi_{i=1}^{n^p} \frac{1}{(\alpha_i)!}\right)$ directly on the corresponding $\alpha$.

From the viewpoint of choosing entries from $\boldsymbol{x}$, first observe that each $\alpha$ considered actually involves $pd$ (possibly repeated) entries of $[n]$ and can be mapped to a multi-set of $pd$ numbers[43], where multiple $\alpha$'s could map to the same multi-set of $pd$ numbers[44]. Thus, one can first pick a multi-set and then go over the different $\alpha$'s corresponding to all possible permutations[45]. Under constraint of $\mathbb{1}_{even(c(\alpha))}$, a multi-set is *valid* (contributes a non-zero term to the summation) only when the multiplicity of each number is even. So, one can view the summation as a process of first choosing $s$ distinct coordinates from $[n]$ such that each coordinate is used a non-zero even number of times when forming a multi-set of $pd$ numbers. Naturally, we have $1 \leq s \leq \lfloor pd/2 \rfloor \leq n$ and $s(\alpha) = s$. For a fixed choice of $s$ coordinates, $\sum_{\substack{\beta_1+\ldots+\beta_s=pd/2 \\ \beta_1 \neq 0,\ldots,\beta_s \neq 0}} \binom{pd}{2\beta_1,\ldots,2\beta_s}$ sums over all valid multi-sets involving $s$ entries of $[n]$. However, since every permutation of a fixed multi-set corresponds to a possibly repeated $\alpha$'s, we divide by $\binom{d}{\alpha_1,\ldots,\alpha_{n^p}}$[46]. Finally, each such $\alpha$ is then scaled by $\left(\Pi_{i=1}^{n^p} \frac{1}{(\alpha_i)!}\right)$. $\square$

### Example illustrating Claim 34

We illustrate the counting process with an example where $p = 2$, $n = 2$, and $d = 3$. Denote $\alpha, \beta, \gamma \in [n]^2$ as three distinct coordinates of $\boldsymbol{Y}$. By picking entries $\{\boldsymbol{Y}_\alpha, \boldsymbol{Y}_\beta, \boldsymbol{Y}_\gamma\}$, the corresponding Hermite polynomial $h_1(\boldsymbol{Y}_\alpha)h_1(\boldsymbol{Y}_\beta)h_1(\boldsymbol{Y}_\gamma) = \boldsymbol{Y}_\alpha \boldsymbol{Y}_\beta \boldsymbol{Y}_\gamma$ is multi-linear. With repeated entries such as $\{\boldsymbol{Y}_\alpha, \boldsymbol{Y}_\alpha, \boldsymbol{Y}_\beta\}$ and $\{\boldsymbol{Y}_\alpha, \boldsymbol{Y}_\alpha, \boldsymbol{Y}_\alpha\}$, the corresponding Hermite polynomials are $h_2(\boldsymbol{Y}_\alpha)h_1(\boldsymbol{Y}_\beta)$ and $h_3(\boldsymbol{Y}_\alpha)$ respectively.

By the constraint of $\mathbb{1}_{even(c(\alpha))}$, it suffices to only consider choices such that there are an even number of 1's and 2's. Ignoring permutations, there are 10 such selections. Including permutations, there are $\binom{3}{3} \cdot 2 + \binom{3}{2,1} \cdot 6 + \binom{3}{1,1,1} \cdot 2 = 32$ such selections. Note that only the last 2 are multi-linear.

**1 distinct:** $\{\boldsymbol{Y}_{11}, \boldsymbol{Y}_{11}, \boldsymbol{Y}_{11}\}, \{\boldsymbol{Y}_{22}, \boldsymbol{Y}_{22}, \boldsymbol{Y}_{22}\}$

**2 distinct:** $\{\boldsymbol{Y}_{11}, \boldsymbol{Y}_{11}, \boldsymbol{Y}_{22}\}, \quad \{\boldsymbol{Y}_{11}, \boldsymbol{Y}_{12}, \boldsymbol{Y}_{12}\}, \quad \{\boldsymbol{Y}_{11}, \boldsymbol{Y}_{21}, \boldsymbol{Y}_{21}\}, \quad \{\boldsymbol{Y}_{11}, \boldsymbol{Y}_{22}, \boldsymbol{Y}_{22}\}, \{\boldsymbol{Y}_{12}, \boldsymbol{Y}_{12}, \boldsymbol{Y}_{22}\}, \{\boldsymbol{Y}_{21}, \boldsymbol{Y}_{21}, \boldsymbol{Y}_{22}\}$

**3 distinct:** $\{\boldsymbol{Y}_{11}, \boldsymbol{Y}_{12}, \boldsymbol{Y}_{21}\}, \{\boldsymbol{Y}_{12}, \boldsymbol{Y}_{21}, \boldsymbol{Y}_{22}\}$

---

[43] E.g. We can identify the polynomial $\boldsymbol{Y}_{11}\boldsymbol{Y}_{12}\boldsymbol{Y}_{21}\boldsymbol{Y}_{11}$ with the multi-set of its indices $\{1,1,1,1,1,1,2,2\}$.

[44] E.g. $\boldsymbol{Y}_{11}\boldsymbol{Y}_{12}\boldsymbol{Y}_{21}\boldsymbol{Y}_{11}, \boldsymbol{Y}_{11}\boldsymbol{Y}_{12}\boldsymbol{Y}_{12}\boldsymbol{Y}_{11}$, and $\boldsymbol{Y}_{11}\boldsymbol{Y}_{11}\boldsymbol{Y}_{11}\boldsymbol{Y}_{22}$ all map to $\{1,1,1,1,1,1,2,2\}$.

[45] E.g. $(1,1,1,2,2,1,1,1) \equiv \boldsymbol{Y}_{11}\boldsymbol{Y}_{12}\boldsymbol{Y}_{21}\boldsymbol{Y}_{11}$ and $(1,1,1,2,1,2,1,1) \equiv \boldsymbol{Y}_{11}\boldsymbol{Y}_{12}\boldsymbol{Y}_{12}\boldsymbol{Y}_{11}$ are counted differently.

[46] E.g. Suppose $\beta_1 = 3$, $\beta_2 = 1$ and $pd = 8$ in the combinatorial summation. $\binom{pd}{2\beta_1,\ldots,2\beta_s}$ will include permutations such as $(1,1,1,2,1,2,1,1)$ and $(1,1,1,1,1,2,1,2)$. However, both of $(1,1,1,2,1,2,1,1)$ and $(1,1,1,1,1,2,1,2)$ actually refer to the same $\alpha$ term since $\boldsymbol{Y}_{11}\boldsymbol{Y}_{12}\boldsymbol{Y}_{12}\boldsymbol{Y}_{11} = \boldsymbol{Y}_{11}\boldsymbol{Y}_{11}\boldsymbol{Y}_{12}\boldsymbol{Y}_{12}$.

We first compute the summation on the left hand side of Claim 34. An $\alpha$ with 1 distinct entry such as $\{Y_{11}, Y_{11}, Y_{11}\}$ contributes $\left(\frac{k}{n}\right)^2 \frac{1}{3!}$ to the summation. With 2 distinct entries, such as $\{Y_{11}, Y_{12}, Y_{12}\}$, we get $\left(\frac{k}{n}\right)^4 \frac{1}{1!2!}$. Finally, each multi-linear polynomial contributes $\left(\frac{k}{n}\right)^4 \frac{1}{1!1!1!}$. So,

$$\sum_{|\alpha|=d} \mathbb{1}_{even(c(\alpha))} \left(\frac{k}{n}\right)^{2s(\alpha)} \left(\Pi_{i=1}^{n^p} \frac{1}{(\alpha_i)!}\right) = \frac{2}{3!}\left(\frac{k}{n}\right)^2 + \frac{6}{1!2!}\left(\frac{k}{n}\right)^4 + \frac{2}{1!1!1!}\left(\frac{k}{n}\right)^4 = \frac{1}{3}\left(\frac{k}{n}\right)^2 + 5\left(\frac{k}{n}\right)^4$$

On the right hand side of Claim 34, we count by viewing the selection of 3 entries of $Y$ as filling up $pd = 6$ slots with values from $\{1, 2\}$:

$$\frac{1}{d!} \sum_{s=1}^{\lfloor pd/2 \rfloor} \binom{n}{s} \left(\frac{k}{n}\right)^{2s} \sum_{\substack{\beta_1+\ldots+\beta_s=pd/2 \\ \beta_1 \neq 0, \ldots, \beta_s \neq 0}} \binom{pd}{2\beta_1, \ldots, 2\beta_s}$$

$$= \frac{1}{3!} \sum_{s=1}^{2} \binom{2}{s} \left(\frac{k}{n}\right)^{2s} \sum_{\substack{\beta_1+\ldots+\beta_s=3 \\ \beta_1 \neq 0, \ldots, \beta_s \neq 0}} \binom{6}{2\beta_1, \ldots, 2\beta_s}$$

$$= \frac{1}{6} \binom{2}{1} \left(\frac{k}{n}\right)^2 \binom{6}{6} + \frac{1}{6} \binom{2}{2} \left(\frac{k}{n}\right)^4 \left[\binom{6}{2,4} + \binom{6}{4,2}\right]$$

$$= \frac{1}{3}\left(\frac{k}{n}\right)^2 + 5\left(\frac{k}{n}\right)^4$$

**Claim 35.** *For $x > 0$ and $0 < a < 1$, we have $xa^x \leq \min\left\{x, \frac{1}{e\ln(1/a)}\right\}$.*

*Proof.* When $0 < a < 1$, we have $xa^x \leq x$ trivially. For $x > 0$ and $0 < a < 1$, we see that $(1/a)^x > 0$. So,

$$\left(\frac{1}{a}\right)^x \geq e\ln\left(\frac{1}{a}\right)^x = ex\ln\left(\frac{1}{a}\right) \iff xa^x \leq \frac{1}{e\ln\left(\frac{1}{a}\right)}$$

Thus, $xa^x \leq \min\left\{x, \frac{1}{e\ln(1/a)}\right\}$. $\qquad\qquad\square$

**Lemma 28.** *Let $p \geq 2$, $d \geq 1$, $Y \in \otimes^p \mathbb{R}^n$ be an observation tensor, $x$ be a $k$-sparse scaled Rademacher vector, and $\{h_\alpha\}_\alpha$ be the set of normalized probabilists' Hermite polynomials. An entry of $Y \in \otimes^p \mathbb{R}^n$ can be indexed by either an integer from $[n^p]$ or a $p$-tuple. Define $\phi : [n^p] \to [n]^p$, $\alpha$, $c(\alpha)$, $s(\alpha)$, and $\mathbb{1}_{even(c(\alpha))}$ as follows:*

- *$\phi(i)$ maps to a $p$-tuple indicating the $p$ (possibly repeated) entries of $x$ that are used.*

- *$\alpha = (\alpha_1, \ldots, \alpha_{n^p})$ is an $n^p$-tuple that corresponds to a Hermite polynomial of degree $|\alpha| = \sum_{i=1}^{n^p} \alpha_i$. For each $i$, $\alpha_i$ is the number of times entry $Y_{\phi(i)}$ was chosen, where each $Y_{\phi(i)}$ references $p$ coordinates of $x$.*

- *$c(\alpha) = (c_1, \ldots, c_n)$, where $c_j$ is the number of times $x_j$ is used in $\alpha$.*

- *$s(\alpha)$ is the number of distinct non-zero $x_j$'s in $c(\alpha)$.*

- *$\mathbb{1}_{even(c(\alpha))}$ be the indicator whether all $c_j$'s are even.*

*Under these definitions, we have the following:*

$$(\mathbb{E}_{H_1} h_\alpha(Y))^2 = \lambda^{2d} k^{-pd} \mathbb{1}_{even(c(\alpha))} \left(\frac{k}{n}\right)^{2s(\alpha)} \left(\Pi_{i=1}^{n^p} \frac{1}{(\alpha_i)!}\right).$$

*Proof of Lemma 28.* For fixed multi-index $\alpha = (\alpha_1, \ldots, \alpha_{n^p})$ such that $|\alpha| = d$, we now compute $\mathbb{E}_{H_1} h_\alpha(\boldsymbol{Y})$.

$$
\begin{aligned}
&\mathbb{E}_{H_1} h_\alpha(\boldsymbol{Y}) \\
=&\mathbb{E}_{H_1} \Pi_{i=1}^{n^p} h_{\alpha_i}(\boldsymbol{Y}_{\phi(i)}) && \text{Product of Hermite polys} \\
=&\mathbb{E}_{\boldsymbol{x}} \mathbb{E}_{\boldsymbol{W}_{\phi(i)} \sim N(0,1)} \Pi_{i=1}^{n^p} h_{\alpha_i}(\boldsymbol{Y}_{\phi(i)}) && \text{Definition of } H_1 \\
=&\mathbb{E}_{\boldsymbol{x}} \Pi_{i=1}^{n^p} \mathbb{E}_{\boldsymbol{W}_{\phi(i)} \sim N(0,1)} h_{\alpha_i}(\boldsymbol{Y}_{\phi(i)}) && \text{Independence of } \boldsymbol{W} \text{ entries} \\
=&\mathbb{E}_{\boldsymbol{x}} \Pi_{i=1}^{n^p} \mathbb{E}_{\boldsymbol{W}_{\phi(i)} \sim N(0,1)} h_{\alpha_i}(\boldsymbol{W}_{\phi(i)} + \lambda \Pi_{j \in \phi(i)} x_j) && \text{Definition of } \boldsymbol{Y}_{\alpha_i} \\
=&\mathbb{E}_{\boldsymbol{x}} \Pi_{i=1}^{n^p} \mathbb{E}_{\boldsymbol{z} \sim N(\lambda \Pi_{j \in \phi(i)} x_j, 1)} h_{\alpha_i}(z) && \text{Translation property of Hermite} \\
=&\mathbb{E}_{\boldsymbol{x}} \Pi_{i=1}^{n^p} \sqrt{\frac{1}{(\alpha_i)!}} (\lambda \Pi_{j \in \phi(i)} x_j)^{\alpha_i} && \text{Expectation of deg } \alpha_i \text{ Hermite on } \boldsymbol{z} \sim (\mu, 1) \\
=&\left( \Pi_{i=1}^{n^p} \sqrt{\frac{1}{(\alpha_i)!}} \right) \lambda^d \mathbb{E}_{\boldsymbol{x}} \Pi_{i=1}^{n^p} \Pi_{j \in \phi(i)} x_j^{\alpha_i} && \sum_i^{n^p} \alpha_i = |\alpha| = d \\
=&\left( \Pi_{i=1}^{n^p} \sqrt{\frac{1}{(\alpha_i)!}} \right) \lambda^d \mathbb{E}_{\boldsymbol{x}} \Pi_{j=1}^{n} x_j^{c_j} && \text{Definition of } c(\alpha) = (c_1, \ldots, c_n) \\
=&\left( \Pi_{i=1}^{n^p} \sqrt{\frac{1}{(\alpha_i)!}} \right) \lambda^d \mathbb{1}_{even(c(\alpha))} \left( \frac{k}{n} \right)^{s(\alpha)} k^{-\frac{pd}{2}}
\end{aligned}
$$

The last equality is because $\mathbb{E}_{\boldsymbol{x}} \Pi_{j=1}^{n} x_j^{c_j} = 0$ if there is an odd $c_j$. So, for $|\alpha| = d$,

$$
(\mathbb{E}_{H_1} h_\alpha(\boldsymbol{Y}))^2 = \lambda^{2d} k^{-pd} \mathbb{1}_{even(c(\alpha))} \left( \frac{k}{n} \right)^{2s(\alpha)} \left( \Pi_{i=1}^{n^p} \frac{1}{(\alpha_i)!} \right)
$$

$\square$

**Lemma 29.** *Let $p \geq 2$, $1 \leq D \leq 2n/p$, $\boldsymbol{Y} \in \otimes^p \mathbb{R}^n$ be an observation tensor, $\boldsymbol{x}$ be a $k$-sparse scaled Rademacher vector, and $\{h_\alpha\}_\alpha$ be the set of normalized probabilists' Hermite polynomials. Then,*

$$
\sum_{|\alpha| \leq D} (\mathbb{E}_{H_1}[h_\alpha(\boldsymbol{Y})])^2 \leq \sum_{d=1}^{D} \frac{\lambda^{2d}}{d!} \sum_{s=1}^{pd/2} \left( \frac{ek^2}{sn} \right)^s \left( \frac{s}{k} \right)^{pd}.
$$

*Proof of Lemma 29.* To upper bound $\sum_{|\alpha| \leq D} (\mathbb{E}_{H_1}[f_\alpha(\boldsymbol{Y})])^2$, we use an equality that relates the counting of $\boldsymbol{Y}$ entries with coordinates of the signal $\boldsymbol{x}$. For a fixed $d$, it can be shown (see Claim 34) that

$$
\sum_{|\alpha|=d} \mathbb{1}_{even(c(\alpha))} \left( \frac{k}{n} \right)^{2s(\alpha)} \left( \Pi_{i=1}^{n^p} \frac{1}{(\alpha_i)!} \right) = \frac{1}{d!} \sum_{s=1}^{\lfloor pd/2 \rfloor} \binom{n}{s} \left( \frac{k}{n} \right)^{2s} \sum_{\substack{\beta_1 + \ldots + \beta_s = pd/2 \\ \beta_1 \neq 0, \ldots, \beta_s \neq 0}} \binom{pd}{2\beta_1, \ldots, 2\beta_s}
$$

This allows us to perform combinatoric arguments on the coordinates of the signal $\boldsymbol{x}$ instead of over the tensor coordinates of $\boldsymbol{Y}$.

$$\sum_{|\alpha| \leq D} (\mathbb{E}_{H_1}[f_\alpha(\boldsymbol{Y})])^2$$

$$= \sum_{d=1}^{D} \sum_{|\alpha|=d} \lambda^{2d} k^{-pd} \mathbb{1}_{even(c(\alpha))} \left(\frac{k}{n}\right)^{2s(\alpha)} \left(\Pi_{i=1}^{n^p} \frac{1}{(\alpha_i)!}\right) \qquad \text{From above}$$

$$= \sum_{d=1}^{D} \lambda^{2d} k^{-pd} \frac{1}{d!} \sum_{s=1}^{\lfloor pd/2 \rfloor} \binom{n}{s} \left(\frac{k}{n}\right)^{2s} \sum_{\substack{\beta_1+\ldots+\beta_s=pd/2 \\ \beta_1 \neq 0,\ldots,\beta_s \neq 0}} \binom{pd}{2\beta_1,\ldots,2\beta_s} \qquad \textcolor{blue}{\text{Claim 34}}$$

$$\leq \sum_{d=1}^{D} \frac{\lambda^{2d} k^{-pd}}{d!} \sum_{s=1}^{pd/2} \binom{n}{s} \left(\frac{k}{n}\right)^{2s} \sum_{\substack{\beta_1+\ldots+\beta_s=pd/2 \\ \beta_1 \neq 0,\ldots,\beta_s \neq 0}} \binom{pd}{2\beta_1,\ldots,2\beta_s} \qquad \text{Drop floor}$$

$$\leq \sum_{d=1}^{D} \frac{\lambda^{2d} k^{-pd}}{d!} \sum_{s=1}^{pd/2} \binom{n}{s} \left(\frac{k}{n}\right)^{2s} \sum_{\beta_1+\ldots+\beta_s=pd/2} \binom{pd}{2\beta_1,\ldots,2\beta_s} \qquad \text{Drop } \beta_i \neq 0$$

$$\leq \sum_{d=1}^{D} \frac{\lambda^{2d} k^{-pd}}{d!} \sum_{s=1}^{pd/2} \binom{n}{s} \left(\frac{k}{n}\right)^{2s} \sum_{\gamma_1+\ldots+\gamma_s=pd} \binom{pd}{\gamma_1,\ldots,\gamma_s} \qquad \text{Drop "evenness constraint"}$$

$$= \sum_{d=1}^{D} \frac{\lambda^{2d} k^{-pd}}{d!} \sum_{s=1}^{pd/2} \binom{n}{s} \left(\frac{k}{n}\right)^{2s} s^{pd} \qquad \text{Multinomial theorem}$$

$$\leq \sum_{d=1}^{D} \frac{\lambda^{2d} k^{-pd}}{d!} \sum_{s=1}^{pd/2} \left(\frac{ek^2}{sn}\right)^{s} s^{pd} \qquad \binom{n}{s} \leq \left(\frac{en}{s}\right)^s$$

$$= \sum_{d=1}^{D} \frac{\lambda^{2d}}{d!} \sum_{s=1}^{pd/2} \left(\frac{ek^2}{sn}\right)^{s} \left(\frac{s}{k}\right)^{pd}$$

$$\square$$

**Lemma 30.** *For $p \geq 2$, $d \geq 1$, $1 \leq k \leq n$ and $1 \leq s \leq pd/2$, we have*

$$\left(\frac{ek^2}{sn}\right)^{s} \left(\frac{s}{k}\right)^{pd} \leq \left[\frac{2pd}{\min\left\{\sqrt{npd},\ k\left(1 + \left|\ln\left(\frac{npd}{ek^2}\right)\right|\right)\right\}}\right]^{pd}.$$

*Proof of Lemma 30.* We will first push all terms into $[\cdots]^{pd}$ and then upper bound the terms inside[47]. We start by recalling three useful inequalities:

- For $x > 0$, we have $x^{\frac{1}{x}} \leq 2$.

- For $x > 0$ and $0 < a < 1$, we have $xa^x \leq \min\left\{x, \frac{1}{e \ln(1/a)}\right\}$.

- For $x \geq \frac{1}{e}$, we have $\min\left\{\frac{1}{2}, \frac{1}{e \ln x}\right\} \leq \frac{1}{1+|\ln(x)|}$.

Using the first inequality, we get

$$\left(\frac{ek^2}{sn}\right)^{s} \left(\frac{s}{k}\right)^{pd} = \left[\left(\frac{ek^2}{sn}\right)^{\frac{s}{pd}} \frac{s}{k}\right]^{pd} = \left[\left(\frac{ek^2}{npd}\right)^{\frac{s}{pd}} \left(\frac{pd}{s}\right)^{\frac{s}{pd}} \frac{s}{k}\right]^{pd} \leq \left[2 \left(\frac{ek^2}{npd}\right)^{\frac{s}{pd}} \frac{s}{k}\right]^{pd}$$

---

[47]This works because the terms inside are greater than 0 and $pd \geq 1$.

When $ek^2 \geq npd$, we use $s \leq pd/2$ to get

$$\left(\frac{ek^2}{npd}\right)^{\frac{s}{pd}} \frac{s}{k} \leq \left(\frac{ek^2}{npd}\right)^{\frac{pd/2}{pd}} \frac{pd/2}{k} = \sqrt{\frac{epd}{4n}} \leq \sqrt{\frac{pd}{n}}$$

When $ek^2 < npd$, we use the second and third inequalities[48] to get

$$\left(\frac{ek^2}{npd}\right)^{\frac{s}{pd}} \frac{s}{k} = \frac{s}{pd} \left(\frac{ek^2}{npd}\right)^{\frac{s}{pd}} \frac{pd}{k} \leq \min\left\{\frac{1}{2}, \frac{1}{e \ln\left(\frac{npd}{ek^2}\right)}\right\} \cdot \frac{pd}{k} \leq \frac{pd}{k\left(1 + \left|\ln\left(\frac{npd}{ek^2}\right)\right|\right)}$$

Putting together, we see that

$$\left(\frac{ek^2}{sn}\right)^s \left(\frac{s}{k}\right)^{pd} \leq \left[2 \max\left\{\sqrt{\frac{pd}{n}}, \frac{pd}{k\left(1 + \left|\ln\left(\frac{npd}{ek^2}\right)\right|\right)}\right\}\right]^{pd} = \left[\frac{2pd}{\min\left\{\sqrt{npd},\ k\left(1 + \left|\ln\left(\frac{npd}{ek^2}\right)\right|\right)\right\}}\right]^{pd}$$

$$\square$$

## F.4   Proofs for information-theoretic lower bound

**Lemma 32.** *Let $U_k$ be the set of $k$-sparse flat unit vectors and $N(U_k, \|\cdot\|_2, \epsilon)$ be the $\epsilon$-covering number of $U_k$ with respect to Euclidean distance. For $\epsilon \in (0, 1]$ and $n \geq 2k$,*

$$N(U_k, \|\cdot\|_2, \epsilon) \geq \left(\frac{n-k}{k}\right)^{k\left(1 - \frac{\epsilon^2}{2}\right)}.$$

*Proof of Lemma 32.* For $x, x' \in U_k$, let us denote $\alpha = |\{i \in [n] : i \in \mathcal{I}_x \cap \mathcal{I}_{x'} \text{ and } x_i = x_i'\}|$ be the intersecting indices with agreeing signs, $\beta = |\{i \in [n] : i \in \mathcal{I}_x \cap \mathcal{I}_{x'} \text{ and } x_i = -x_i'\}|$ be the intersecting indices with disagreeing signs, and $\gamma = |\{i \in [n] : i \notin \mathcal{I}_x \cap \mathcal{I}_{x'}\}|$ be the non-intersecting indices. By definition, $\alpha \geq 0$, $\beta \geq 0$, $\gamma \geq 0$, $\alpha + \beta = |\mathcal{I}_x \cap \mathcal{I}_{x'}|$, $\alpha + \beta + \gamma = 2k - |\mathcal{I}_x \cap \mathcal{I}_{x'}| = |\mathcal{I}_x| + |\mathcal{I}_{x'}| - |\mathcal{I}_x \cap \mathcal{I}_{x'}|$, and $\gamma = 2(k - |\mathcal{I}_x \cap \mathcal{I}_{x'}|)$. Then, for $x, x' \in U_k$,

$$\|x - x'\|_2 = \sqrt{\beta\left(\frac{2}{\sqrt{k}}\right)^2 + \gamma\left(\frac{1}{\sqrt{k}}\right)^2} = \sqrt{\frac{4\beta + \gamma}{k}} \geq \sqrt{\frac{\gamma}{k}} = \sqrt{2 - \frac{2|\mathcal{I}_x \cap \mathcal{I}_{x'}|}{k}}$$

So, $\|x - x'\|_2 \leq \epsilon$ implies that $|\mathcal{I}_x \cap \mathcal{I}_{x'}| \geq k(1 - \frac{\epsilon^2}{2})$. This means that for any *fixed* $x \in U_k$, there are at most[49] $\sum_{i=0}^{\lfloor \epsilon^2 k/2 \rfloor} \binom{k}{i}\binom{n-k}{i}$ vectors in $U_k$ (including $x$ itself) that are of distance at most $\epsilon$ from $x$. By definition of covering number, we know that

$$N(U_k, \|\cdot\|_2, \epsilon) \cdot \sum_{i=0}^{\lfloor \epsilon^2 k/2 \rfloor} \binom{k}{i}\binom{n-k}{i} \geq |U_k| = 2^k \binom{n}{k}$$

Thus, to argue that $N(U_k, \|\cdot\|_2, \epsilon) \geq \left(\frac{n-k}{k}\right)^{k\left(1 - \frac{\epsilon^2}{2}\right)}$, it suffices to show

$$2^k \binom{n}{k} \geq \left(\frac{n-k}{k}\right)^{k\left(1 - \frac{\epsilon^2}{2}\right)} \cdot \sum_{i=0}^{\lfloor \epsilon^2 k/2 \rfloor} \binom{k}{i}\binom{n-k}{i}$$

---

[48]Observe that $0 < \frac{s}{pd} \leq \frac{1}{2}$, $0 < \frac{ek^2}{npd} < 1$, and $\frac{1}{e} \leq 1 < \frac{npd}{ek^2}$.

[49]First pick $i$ out of $k$ coordinates of $x$ to be different, then pick the $i$ different coordinates amongst the $n - k$ coordinates outside of $\mathcal{I}_x$. The summation is from 0 to $\lfloor \epsilon^2 k/2 \rfloor$ because we need to have $|\mathcal{I}_x \cap \mathcal{I}_{x'}| \geq k(1 - \frac{\epsilon^2}{2})$.

Observe that since $n \geq 2k$, the term $\binom{n-k}{i}$ increases with $i$:

$$\sum_{i=0}^{\lfloor \epsilon^2 k/2 \rfloor} \binom{k}{i}\binom{n-k}{i} \leq \binom{n-k}{\lfloor \epsilon^2 k/2 \rfloor} \sum_{i=0}^{\lfloor \epsilon^2 k/2 \rfloor} \binom{k}{i} \qquad (\star)$$

$$\leq \binom{n-k}{\lfloor \epsilon^2 k/2 \rfloor} \sum_{i=0}^{k} \binom{k}{i} \qquad \lfloor \epsilon^2 k/2 \rfloor \leq k$$

$$= \binom{n-k}{\lfloor \epsilon^2 k/2 \rfloor} 2^k \qquad \text{Binomial theorem}$$

where $(\star)$ is because $n \geq 2k$ and $\epsilon \in (0,1]$ implies that $n - k \geq \epsilon^2 k$ so $\binom{n-k}{i} \leq \binom{n-k}{\lfloor \epsilon^2 k/2 \rfloor}$ for $0 \leq i \leq \lfloor \epsilon^2 k/2 \rfloor$. Thus, it suffices to show

$$\binom{n}{k} \geq \left(\frac{n-k}{k}\right)^{k\left(1-\frac{\epsilon^2}{2}\right)} \cdot \binom{n-k}{\lfloor \epsilon^2 k/2 \rfloor}$$

We will now show that $\frac{\binom{n}{k}}{\binom{n-k}{\lfloor \epsilon^2 k/2 \rfloor}} \geq \left(\frac{n-k}{k}\right)^{k\left(1-\frac{\epsilon^2}{2}\right)}$:

$$\frac{\binom{n}{k}}{\binom{n-k}{\lfloor \epsilon^2 k/2 \rfloor}} = \frac{n!}{k!(n-k)!} \frac{(\lfloor \epsilon^2 k/2 \rfloor)!(n-k-\lfloor \epsilon^2 k/2 \rfloor)!}{(n-k)!}$$

$$= \frac{n!}{(n-k)!} \frac{(\lfloor \epsilon^2 k/2 \rfloor)!}{k!} \frac{(n-k-\lfloor \epsilon^2 k/2 \rfloor)!}{(n-k)!}$$

$$= [(n) \cdot \ldots \cdot (n-k+1)] \cdot \left[\frac{1}{(k) \cdot \ldots \cdot (k-\lfloor \epsilon^2 k/2 \rfloor + 1)}\right] \cdot$$

$$\left[\frac{1}{(n-k) \cdot \ldots \cdot (n-k-\lfloor \epsilon^2 k/2 \rfloor + 1)}\right]$$

$$\geq (n-k)^k \left(\frac{1}{k}\right)^{\lfloor \epsilon^2 k/2 \rfloor} \left(\frac{1}{n-k}\right)^{\lfloor \epsilon^2 k/2 \rfloor}$$

$$\geq (n-k)^k \left(\frac{1}{k}\right)^{\frac{\epsilon^2 k}{2}} \left(\frac{1}{n-k}\right)^{\frac{\epsilon^2 k}{2}}$$

$$\geq (n-k)^k \left(\frac{1}{k}\right)^{k\left(1-\frac{\epsilon^2}{2}\right)} \left(\frac{1}{n-k}\right)^{\frac{\epsilon^2 k}{2}}$$

$$= \left(\frac{n-k}{k}\right)^{k\left(1-\frac{\epsilon^2}{2}\right)}$$

where the last inequality is because $\epsilon \leq 1$ implies that $1 - \frac{\epsilon^2}{2} \geq \frac{\epsilon^2}{2}$. $\qquad \square$

**Lemma 33.** *Denote $\mathcal{S}_k^{n-1}$ as the set of $k$-sparse unit vectors. Then,*

$$\max_{u,v \in \mathcal{S}_k^{n-1}} D_{KL}\left(\mathbb{P}_{\mathbf{Y} \sim \mathcal{Y}|u} \,\Big\|\, \mathbb{P}_{\mathbf{Y} \sim \mathcal{Y}|v}\right) \leq 2\lambda^2$$

*where $D_{KL}(\cdot \| \cdot)$ is the KL-divergence function and $\mathbb{P}_{\mathbf{Y} \sim \mathcal{Y}|u}$ is the probability distribution of observing $\mathbf{Y}$ from signal $u$ with additive standard Gaussian noise tensor $\mathbf{W}$.*

*Proof of Lemma 33.* Define $vec(T)$ as vectorization of a tensor from $\otimes^p \mathbb{R}^n$ to $\mathbb{R}^{n^p}$. Then, for $u \in \mathcal{S}_k^{n-1}$, we see that $\|vec(\lambda u^{\otimes p})\|_2^2 = \lambda^2$ and the distribution $\mathbb{P}_{\mathbf{Y} \sim \mathcal{Y}|u}$ follows the distribution of a Gaussian vector $\mathbf{g} \sim N(vec(\lambda u^{\otimes p}), I_{n^p})$.

For two Gaussian vectors $\mathbf{g} \sim N(\mu_0, I_{n^p})$ and $\mathbf{h} \sim N(\mu_1, I_{n^p})$, we know that $D_{KL}(\mathbf{g} \| \mathbf{h}) = \frac{1}{2}(\mu_1 - \mu_0)^\top (\mu_1 - \mu_0) = \frac{1}{2}\|\mu_1 - \mu_0\|_2^2 \leq \frac{1}{2}(\|\mu_1\|_2 + \|\mu_0\|_2)^2$ by triangle inequality[50].

---

[50] We get an equality if $\mu_1 = -\mu_0$.

Thus, $D_{KL}\left(\mathbb{P}_{\boldsymbol{Y}\sim\mathcal{Y}|u} \,\middle\|\, \mathbb{P}_{\boldsymbol{Y}\sim\mathcal{Y}|v}\right) \leq \frac{1}{2}(\lambda^2 + \lambda^2)^2 = 2\lambda^2.$ $\qquad\square$