# OpenReview forum: "The Complexity of Sparse Tensor PCA"
_NeurIPS.cc/2021/Conference — NeurIPS 2021 Poster_

### Official Review · Reviewer_BF19 · 2021-07-01

**Rating:** 7
**Confidence:** 4

**Summary:**

This paper studies the problem of sparse tensor PCA (a generalization of sparse PCA and tensor PCA). The main result is an algorithm that -- for the highly sparse regime in which the sparsity is at most the square root of the signal dimension -- naturally interpolates between a poly-time efficient algorithm and exponential exhaustive search. This result is complemented by lower bounds for the rather popular computational model captured by low-degree polynomials. The lower bounds match the algorithm guarantees (although for the different problem of distinguishability) in the regime of constant tensor order $p$.

The results of this paper recover several results of the literature, and provide a non-trivial extension.

**Limitations And Societal Impact:**

I do not foresee potential negative societal impact for this contribution.

**Main Review:**

The authors propose a model that generalizes sparse PCA and tensor PCA -- two problems that have received a fair amount of attention in the recent theoretical literature. The idea of the algorithm is quite natural, and the proposed algorithm can be summarised as follows:

i) Create two independent copies of the tensor by adding and subtracting an independent Gaussian tensor (at the cost of a factor $1/\sqrt{2}$ in the SNR).

ii) Perform an exhaustive search over all $t$-sparse vectors and find the one that maximizes the scalar product between the first copy of the tensor and the $p$-th Kronecker power of the candidate. When $t=k$ -- $k$ being the signal sparsity -- this corresponds to exhaustive search. Choosing a smaller $t$ allows the trade-off between complexity and performance.

iii) Use the second copy of the tensor and the maximizer from (ii) to estimate the support of the signal.

The authors also propose an algorithm for the recovery of multiple signals with disjoint support, which goes along similar lines (although the technical details do become more involved).

The proof of the low-degree polynomial lower bound is not discussed in the main paper, and an inspection of the appendix seems to reveal that it follows a similar strategy of e.g.  [HKP$^+$17, HS17, DKWB19].

These results generalize existing ones, i.e., [DKWB19, LZ20], and offer a unified view of the two problems involving sparsity and tensors. However, the approach seems to resemble quite closely the one in [DKWB19] (which looks at sparse PCA) and it is not entirely clear what kind of new ingredients are needed to tackle the tensor case. The paper is overall clear and rather well written. The algorithms are described in an intuitive way, and the authors provide a comparison with similar results in the literature -- these are two features that I highly appreciated. At the same time, there are a few issues that I will detail below.

Although I am rather positive about this submission, I have the following concerns:

1) Novelty. As briefly described above, what's the difference with respect to [DKWB19]? What new ingredients are needed to handle tensors instead of matrices? The authors just mention that the family of algorithms is "heavily inspired by [DKWB19]", but a more thorough discussion is needed here.

2) One of the selling points is that, for $r$ distinct $k$-sparse signals, the proposed algorithm requires $\lambda\ge\tilde{\mathcal O}(k)$ ($\lambda$ is the SNR), while known algorithms require $\lambda\ge\tilde{\mathcal O}(k\cdot r)$. However, this paper requires disjoint supports while the existing works just need orthogonality. Is there any way to adapt the proposed approach to cover the case where the signals are just orthogonal? Is there any reason why one would expect that the improvement of the factor $r$ will remain also when the signals are just orthogonal?

3) Lines 67-68. The authors mention that [DKWB19, HSV20] provide a smooth trade-off between signal strength and running time. Is it possible to compare the results of this manuscript to the ones of [DKWB19, HSV20]? I could not find a discussion of this in the main body (the comparison seems to be only with [LZ20]).

4) What's the running time of the algorithm? In the abstract, the running time seems to be of order $n^{p+t}$, while in the theorem statements it becomes $n^{p+t+3}$ (I am looking at the case where $p, r$ are constants). Actually, looking at the proof of Lemma 22 in the appendix, I do not quite see why the running time is $n^{p+t+3}$ and not $n^{p+t}$. You have a factor $n^t$ from the cardinality of $U_t$, and a factor $n^p$ from the computation of the inner product between $Y^{(1)}$ and $u^{\otimes p}$. It seems to me that the additional $n^2$ (from checking the disjoint support) should be summed to $n^p$ (instead of taking the product). Also, why one needs to check for disjoint support for the single signal case? Same comment about the extra $n$ factor coming from obtaining the largest $k$ entries of the vector $\alpha$ (it seems to me that there should be a sum in the calculation of the running time, and not a product).

5) The statement of Theorem 2 is missing some words/lines (see line 142).

6) Appendix, lines 654-655. How do you get this formula? Please discuss or refer to the corresponding equations in Appendix F.

7) Appendix, formulas on top of page 30. How is $\tilde{x}$ defined? I could not follow here.


------

*Post-rebuttal update.* The authors addressed my concerns in a satisfactory way, and I have raised my score.

**Time Spent Reviewing:**

3

---

> ### Author Response · Authors · 2021-08-10
> **Regarding the reviewer's concerns**
>
> **1:** The algorithm in [DKWB19] is a specialization of ours (with comparable guarantees) to the simplest settings of $p=2$ and a single spike.
>     However, looking at [DKWB19], it is a priori unclear how to generalize the result to the settings of our interest.
>     This is especially true in the tensor settings ($p\geq 3$) with multiple spikes, where signals interfere with each other.
>
> **2:** It is an intriguing question whether this improvement can be achieved in the more general settings of orthogonal spikes. The approach we use relies on the signals having disjoint support and we expect it to not be generalizable to orthogonal signals. This can be  noticed in the simplest settings with brute-force parameter $t=1$ and $p=2$ where the criteria of Algorithm 3 for finding a an entry of a signal vector is to look at the diagonal entries of the data matrix (along the lines of Diagonal Thresholding).
> Here the largest diagonal entry may depend on more than one spike, fooling the algorithm. Nevertheless, we are not aware of any fundamental barrier that may suggest such guarantees are computationally hard to achieve.
>
> **3:** For the special case of sparse PCA (where $p=2$), [DKWB19, HSV20] also provide a smooth trade-off between signal strength and running time. [HSV20] only considers the Wishart model and thus its guarantees cannot be compared to ours. In these settings, their guarantees match those of [DKWB19], that is they observe the same tradeoff between running time and signal-to-noise ratio.
>     In [DKWB19], the authors also considered the *single spike* Wigner model ($Y= \lambda xx^\top + W$, where $x$ is a $k$-sparse unit vector and $W\sim N(0,1)^{n\times n}$). In these matrix settings, our algorithm and the algorithm of [DKWB19] offer the same smooth-trade off between running time and signal strength, up to universal constants. This will be clarified in the final version of the manuscript.
>
> **4:** The additional factor $n^3$ is indeed not necessary, we thank the reviewer for pointing this out. The correct running time is then $\tilde{O}(n^{p+t})$.
>
> **5-6-7:** We thank the reviewer for pointing out these typos. They will be fixed in the final version of the paper.
>
> We would like to thank the anonymous reviewer for her/his valuable feedback.

---

### Official Review · Reviewer_n5Ja · 2021-07-06

**Rating:** 8
**Confidence:** 4

**Summary:**

This paper studies the estimation of a rank-one (or more generally a low-rank) tensor $x^{\otimes p}$ ($x \in \mathbb{R}^n$ is assumed to have unit norm) given $Y = \lambda x^{\otimes p} + W$, where $\lambda > 0$ and $W$ is a Gaussian noise tensor. Here the signal vector $x$ is assumed to have only $k$ non-zero entries.

The main contribution of the paper is an algorithm that recovers with high probability the support of $x$ in time $O(n^{p+t})$ provided that $\lambda \geq O(\sqrt{t} (k/t)^{p/2})$. The number $t$ can be arbitrarily chosen between $1$ and $k$, illustrating a tradeoff between statistical constraints on the signal-to-noise ratio $\lambda$ and runtime constraints on the estimation algorithm.
This generalizes to the 'tensor settings' previous results obtained about sparse PCA.

The authors also prove an information-theoretic lower bound for low-degree polynomials. More precisely, they prove that no test based on polynomial of degree at most $D$ can distinguish between the planted model $Y = \lambda x^{\otimes p} + W$ and the pure noise model $Y=W$, when $\lambda$ is smaller than some quantity depending on $D, n, k, p$. This bound generalizes previously obtained results for PCA and tensor PCA. This bound is tight for low-degree polynomials. However there is a gap with the  bound $\lambda \geq O(\sqrt{t} (k/t)^{p/2})$ needed for the algorithm that recovers the support.




**Limitations And Societal Impact:**

This point has been clearly and firmly addressed by the authors at line 324 with a dedicated remark asserting: 'This work does not present any foreseeable negative societal consequence'. I strongly agree with them.

**Main Review:**

This is a very good paper, in terms of results, clarity and significance. It is particularly well written, making it interesting and enjoyable to read.

The proposed support-recovery algorithm is simple and elegant, while achieving strong guarantees. Such a simple method is particularly refreshing when compared to other heavy/sophisticated approach that have used for low-rank tensor recovery (SDP, tensor unfolding, gradient methods ...). I would be very curious to see if this approach of 'limited brute force' can be extended to other problems! Also it would be of natural interest to see if this approach is optimal, or if one can achieve support recovery at smaller signal-to-noise ratios.

The lower bound for low-degree polynomial is also relevant. However this result is only about the hypothesis problem and low-degree polynomials, while the first result is about support recovery. Hence it seems to me that this result does not really complement the first algorithmic result since:
a) there could a priori exists polynomial time algorithms that works even when low-degree polynomial don't.
b) support estimation may be strictly harder than testing.
For these reasons I do not feel a strong connection between the two main results of the paper.

**Time Spent Reviewing:**

3h

---

> ### Author Response · Authors · 2021-08-10
> **On the connection between the algorithmic results and the lower bound**
>
> The reviewer observations are indeed on point.
>
> Informally speaking, current techniques to prove computational hardness results for average-case problems fall in two broad categories: (i) find a reduction from a (conjecturally) hard problem, (ii) provide a lower bound against a restricted model of computation.
>
> The significance of a result of the second type clearly depends on the power of the computational model considered.
>     For a large family of problems (e.g. planted clique, sparse pca), low-degree polynomials seem to accurately predict the guarantees of algorithmic approaches such as SoS, message passing algorithms and others.
>     Based on this evidence, Theorem 3 suggests that  current techniques should not be able to efficiently solve the distinguish problem  beyond a certain signal-to-noise ratio.
>
> Concerning the relation between recovery and distinguishing. It is important to notice that for low order tensors  (e.g. when $p \in O(1)$), solving the  distinguishing problem appears to be $-$ up to constant factors $-$ as hard as recovering the signal vector. In this regime, the lower bound accurately predicts the guarantees of state-of-the-art algorithms. It is then for higher order tensors where we start observing a gap between the distinguishing lower bound and our recovery algorithms. The existence of this gap is intriguing because it leaves open the possibility of improved efficient algorithms.
>
> We would like to conclude by thanking the anonymous reviewer for her/his valuable feedback.

---

> > ### Comment · Reviewer_n5Ja · 2021-08-15
> > **About the author's response**
> >
> > I would like to thank the authors for these explanations. I think I didn't really noticed this distinction between bounded p / p -> infty. I wonder if the case $p \to \infty$ may be easier to analyze than the bounded p case using the statistical physics wisdom saying that Derida's random energy model is the limit of the p spin for large p... but I am probably a bit naive here.
> > Thanks again for your response and this nice paper. I advocate for acceptance.

---

> > > ### Author Response · Authors · 2021-08-30
> > > **About the reviewer's response**
> > >
> > > Thanks to the reviewer for the response and her/his suggestion of studying first the case $p\rightarrow \infty$.
> > >
> > > It is indeed  interesting to understand how to "fill" the gap between efficient algorithms and computational lower-bounds observed in this manuscript.

---

### Official Review · Reviewer_xHQE · 2021-07-16

**Rating:** 6
**Confidence:** 4

**Summary:**

This paper focuses on the tradeoff between computational complexity and conditions for recovery in a symmetric sparse tensor estimation problem. There has been a great deal of work on related problems recently in the case of matrices (order 2 tensors) and the case of tensors (without necessarily imposing sparsity constraints). The main achievability result (Theorem 1) provides sufficient condition for single spike model in terms of the ambient dimension, the order the tensors, the degree of sparsity, the signal strength, and an integer parameter the interpolates between a ````"low complexity" algorithm  "high complexity" algorithm (essentially brute force search). Similar results of for a multiple spike model are given in Theorem 2.

Necessary conditions for a certain testing problem are considered in Theorem 3 which provides necessary conditions such that a polynomial of bounded degree can "distinguish" between a planted model (with a signal of a given strength) and a null mull of only Gaussian noise. Here the criterion for distinguishability is a certain L2 test.

While similar results have been obtained in the literature, the authors argue that their results improve upon the existing ones in a number of ways.

**Limitations And Societal Impact:**

The paper is theoretical and so I think the discussion is appropriate.

**Main Review:**

On the positive side, I think there are some good ideas in the work. The basic approach outlined at the start of Section 2 is very nice in describing a hierarchy of algorithms based on searching over increasingly more complex signal classes (parameterized by the degree of sparsity). To the extent that the results improve significantly upon the prior work, I think this paper should be published.

One concern with this paper is that it was difficult for me to read and parse the statements. It would have been easier for me if the authors had defined the notation and various recovery criterion and then stated formal results. The current approach gives only informal theorem statements in section 1 (before the notation and recovery objective are clarified) and I found it very difficult to interpret these results. (By contrast the material in section 2 and the appendix was easier to follow).

Related to the above, I do not know the difference between the big O notation and the "less-than-approximately-equal-to" relation. The authors say they use "standard" notation on this front, but as far as I can tell, there is no "standard interpretation" for the less-than-approximately-equal-to relation. The context of the usage did not provide clues either (in some cases the authors combined the less than approximately sign with the big O notation). With respect to the presentation, I will also provide the feedback that the heavy use of footnotes was unpleasant. Even though I am quite interested in the topic and think the results are interesting, I did not enjoy reading this paper because of the lack of flow.

Finally, my main technical concern is about the interpretation of the "Lower bound" in Theorem 3. From my understanding, the lower bound is based on a certain polynomially constrained approximation to the chi-square divergence. A classical result, which the authors review in the appendix, is that boundedness of the chi-square divergence implies an impossibility result for testing (in the standard sense of type I and type II errors) with arbitrarily small error. Based on this fact, prior work has suggested the the behavior of the polynomially constrained approximation to the chi-square divergence could be used as an heuristic proxy for the difficulty of testing using polynomials. But this interpretation is only heuristic -- by itself, the argument does not rule out the possibility that a polynomial could achieve arbitrarily small testing error and/or recovery error. The discussion following Theorem 3 (e.g, line 183) seems to imply the inacheivability results holds in a stronger sense.  This is a case where I think its crucial to describe the precise meaning of the low-degree recovery criterion before stated the result.







**Time Spent Reviewing:**

3

---

> ### Author Response · Authors · 2021-08-10
> **Concerning the interpretation of the lower bound (Theorem 3)**
>
> **Concerning Theorem 3:** We say that a low-degree polynomial distinguishes between two distributions if the difference between the expectations is greater than both standard deviations.
>     This is a natural definition as it means that for typical instances of the planted distribution, the value of the polynomial is far from its value on typical instances of the null distribution.
> So if the low-degree $\chi^2$-divergence is small, then it must be that at least one of the variances is larger than the difference between the two expectations and thus low-degree polynomials cannot distinguish (in the sense described above).
>
> It is not at all clear how a low-degree polynomial $p(\mathbf{Y})$ with vanishing low-degree $\chi^2$-divergence (in the sense $\frac{|E_{\nu}[p(\mathbf{Y})]-E_{\mu} [p(\mathbf{Y})] |}{\sqrt{\mathbb{V}_{\nu}(p(\mathbf{Y}))}}\leq o(1)$) could achieve small testing error. We do not have any examples where such a polynomial has been constructed, or where it has been shown to even exist. Moreover, even among degree-$\omega(\log n)$ polynomials, we do not have such an example.
>
> Nevertheless, as pointed out by the reviewer, line 183 might be misleading and will be rewritten appropriately. Furthermore, a discussion of this aspect will be added to the  subsequent paragraph.
>
> **Notation:** As noticed by the reviewer, the "less-than-approximately-equal-to" symbol is not appropriately introduced.  We will address this issue in the final version of the paper. Similarly, we will more carefully use footnotes in the final version of this work.
>
> Finally, we would like to thank the anonymous reviewer for her/his valuable feedback.

---

> > ### Comment · Reviewer_xHQE · 2021-08-20
> > **polynomial testing**
> >
> > With respect to my comments polynomials testing, I see that what I wrote could have been stated better. What I should have clarified is that if the low-degree chi-squared divergence is vanishing, it does not rule out the possibility that a polynomial *applied to a truncated version of the date* could achieve arbitrarily small testing error and/or recovery error.  For example, adding a ``small'' be heavy-tailed noise term to the data would mean the expectations do not exists, but after truncation, a polynomial would have no difficulty. So,  testing with truncation + polynomial is easy but only polynomial is impossible. Of course "truncation" can be replaced with any simple preprocessing step (e.g., taking the logarithm).
> >
> > Is this example meaningful for the spiked tensor problem? I don't know...

---

> > > ### Author Response · Authors · 2021-08-27
> > > **Concerning polynomial testing**
> > >
> > > Thanks to the reviewer for her/his reply.
> > >
> > > With the appropriate preprocessing step, it may indeed be possible to use a simple polynomial to distinguish. That is, for the processed data the low-degree $\chi^2$-divergence may no longer vanish.
> > > However, the key here is the word "appropriate".
> > > For example, consider the case of using an information-theoretically optimal distinguishing algorithm as preprocessing step and then a trivial polynomial on the processed data.
> > >
> > > So, with the respect to the significance of the lower bound in Theorem 3, the question is really whether a given preprocessing step (e.g. truncation, taking the maximum,..) is captured by the computational model considered.
> > >
> > > If a specific algorithm  (preprocessing + polynomial) *is captured* by low-degree polynomials, then Theorem 3 should bound its guarantees.
> > >
> > > Conversely, if a certain preprocessing step *is not captured* by this restricted computational model, then for such an  algorithm this lower bound will not be meaningful.
> > >
> > > From this point of view, it is important to notice that for related average case problems we do not have polynomial-time algorithms that achieve guarantees beyond what is predicted using this low-degree $\chi^2$-divergence.  In other words,  (as far as we are aware) there are no natural related examples where simple-preprocessing steps are known to not be captured by low-degree polynomials.
> > >
> > > As a concrete example, consider the related problem of Sparse PCA for the Wishart model. In the highly sparse settings, a state-of-the-art algorithm$^*$ is Covariance Thresholding [DM14]:
> > > 1. threshold the entries of the empirical covariance matrix,
> > > 2. return its leading eigenvector.
> > >
> > > Covariance Thresholding applies a truncation as a preprocessing step. However, its guarantees are captured  by low-degree polynomials [dKNS20]. There is a known low-degree polynomial which obtain comparable guarantees (as well as a lower bound predicted by the low-degree $\chi^2$-divergence).
> > >
> > >
> > > We thank the anonymous reviewer again for the remarks, this feedback will be used to improve the related parts of the manuscript.
> > >
> > >
> > > $^*[$To be precise, the low-degree polynomial algorithm in [dKNS20] provides improvements over Covariance Thresholding in certain regimes. To avoid unnecessary details we are omitting these settings from the discussion.$]$

---

> > > > ### Comment · Reviewer_xHQE · 2021-08-29
> > > > **interesting discussion**
> > > >
> > > > I think we are on the same page with respect to the scope of the method. Indeed, the low-degree chi-square divergence is meaningful for the setting of preprocessing + polynomial if and only if the preprocessing step is captured by the polynomials. Also, I think the low-degree method is very interesting and the results in your analysis are worthwhile, so please do not take this as a criticism of your results
> > > >
> > > > However, I am not convinced that all computationally ``simple'' pre-precessing steps can be approximated by polynomials. With respect to your statement about no known cases where polynomial-time algorithms succeed beyond the boundaries predicted by the low-degree chi-square divergence, I think the heavy-tailed noise + thresholding example I mentioned above is a counterexample. Consider the spiked Wishart model in a regime where polynomials perform well. Now, change the model by adding another noise matrix whose entries are IID Cauchy (with a small scale factor tending to zero rapidly as the dimension increases). In this regime, I think one can argue that componentwise thresholding + polynomials still succeeds even though the low-degree chi-square divergence is not defined because the matrix entries do not have finite moments. The problem is that componentwise thresholding (which is computationally easy) cannot be well approximated on an unbounded interval by a polynomial of bounded degree.

---

> > > > > ### Author Response · Authors · 2021-08-30
> > > > > **Thresholding and low-degree polynomial**
> > > > >
> > > > > > I think we are on the same page with respect to the scope of the method. Indeed, the low-degree chi-square divergence is meaningful for the setting of preprocessing + polynomial if and only if the preprocessing step is captured by the polynomials. Also, I think the low-degree method is very interesting and the results in your analysis are worthwhile, so please do not take this as a criticism of your results
> > > > >
> > > > > Not at all! I highly appreciate the constructive criticism arising from this discussion (and the time spent writing it).
> > > > >
> > > > > Regarding the reviewer's example, it is indeed true that under such a distribution there is a regime in which thresholding + polynomials works while the $\chi^2$-divergence is not meaningful.
> > > > >
> > > > > I apologise if I glossed over this example.
> > > > > However, I still believe this example has a limitation: one could obtain a distribution with finite moment, that (to the eyes of a distinguisher) looks like the actual distribution, namely by thresholding at some appropriately large value.
> > > > >
> > > > > If, the truncated distribution is close to the actual distribution (for example, the total variation is upper bounded by some $\epsilon>0$ exponentially small) then computing the low-degree $\chi^2$-divergence between the null distribution and the truncated distribution still provides  formal evidence regarding the computational hardness of the problem.
> > > > >
> > > > > In any case, the one outlined by the reviewer is a clear limitation of this $\chi^2$-divergence technique for providing formal evidence of lower bounds. Needless to say, this is painfully clear in the context of inference problems with adversarial or heavy-tailed noise (still, there are some examples in this area where this approach yields satisfying results).
> > > > >
> > > > > Thanks again to the reviewer for this discussion.
> > > > > This thread will be used to clarify the relevant paragraphs in the paper.

---

### Official Review · Reviewer_rbms · 2021-07-16

**Rating:** 7
**Confidence:** 3

**Summary:**

The authors propose a family of algorithms for solving the sparse tensor PCA problems, which is based on smooth interpolation between a polynomial-time algorithm and exponential-time search algorithm. Improved results are achieve for sparce PCA in the lower signal/noise ratio regime in distinct sparse signal cases, compared with existing guarantees. Lower bound analysis is also provided for the proposed algorithm.

**Limitations And Societal Impact:**

Yes

**Main Review:**

The proposed algorithm is analyzed in several sparse models. In the single spike model, the proposed algorithm improves over existing guarantees in the sparser cases for sparse vectors, and achieves the same same guarantees for matrix cases. In the multiple spikes models, the proposed algorithm recovers the sparse PCA case by additional requirement of orthogonality on the sparse vectors. The trade-off is that the proposed algorithm allows weaker signals compared to the noise. Similar results hold when extended to the tensor cases.

The hardness of the result is also demonstrated, where the lower bound for the low-degree polynomials is provided that is same with the sparse PCA and tensor PCA cases. It is also tight in the analyzed case. Overall, the theoretical analysis for the proposed algorithm is rather comprehensive and covers most interesting scenarios.

The paper is relatively easy to follow and clearly presented. The presented proof is mostly clear and seems correct, though I did not check all detailed analysis for the proof. Overall, I think the paper overall shows an interesting result.

**Time Spent Reviewing:**

6

---

> ### Author Response · Authors · 2021-08-10
> **Thanks for the feedback**
>
> We thank the anonymous reviewer for her/his valuable feedback. We will make use of her/his comments and improve the final version of the paper.

---

### Official Review · Reviewer_y5Ce · 2021-07-16

**Rating:** 7
**Confidence:** 4

**Summary:**

# Review for "The Complexity of Sparse Tensor PCA"

This paper studies the "Sparse Tensor PCA" problem, which is a common generalization of two well-studied problems: sparse principal component analysis and tensor principal component analysis.
The problem is as follows: given a "single-spike" tensor of the form

$T = \lambda \cdot x^{\otimes p} + W$

where $W$ has i.i.d. Gaussian entries, $\lambda > 0$ is the "signal strength", and $x \in R^n$ is $k$-sparse for some $k \ll n$, find $x$.
(The problem can be generalized to more than one spike $x$; this is treated carefully in the paper.)

The paper contains interesting new algorithms and computational lower bounds for this problem.


**Limitations And Societal Impact:**

Limitations are considered carefully. This is a theory paper with no imminent direct societal impact.

**Main Review:**

Tensor methods are useful in high-dimensional statistics, with key applications to learning latent variable models, multi-way factor analysis, etc.
The single-spike model (tensor PCA without the sparsity assumption on $x$) is a well-studied model (with major developments in the last 5 years) of noisy tensor problems, whose computational complexity is by now largely well understood.
However, without a sparsity assumption on $x$, tensor PCA requires $\lambda$ to scale as $n^{\Omega(p)}$ for polynomial-time algorithms: this is expensive.

As in many settings, this may be alleviated if the underlying objects to be estimated from data are sparse.
In the matrix setting (i.e. $p = 2$), this has been extensively studied via the "sparse PCA" problem.
The sparse single spike tensor model above is a nice and simple "toy" model in which to understand just how much easier sparsity makes noisy tensor problems.
Thus, the main question addressed by the paper is: what is the computational complexity of sparse tensor PCA, as a function of tensor order $p$, ambient dimension $n$,  signal strength $\lambda$, and sparsity $k$?

The paper has two main results.


(1) The first is an algorithm for the sparse spike tensor model which recovers $x$  in polynomial time when $\lambda \gg k^{p/2}$ -- this a tensor analogue of the well known "diagonal thresholding" algorithm for sparse PCA.
This algorithm improves on the state of the art for the sparse tensor pca problem: previous algorithms required $\lambda \gg \sqrt{p k^p}$ -- this is a meaningful difference for high-order tensors $p \gg 1$.
(Whether such tensors are practically interesting is a little unclear to me, but the problem is mathematically fundamental enough that I think this is not a big deal.)

The paper actually offers a family of algorithms with a tune-able running-time parameter, giving running times which interpolate from polynomial to exponential time.
At the exponential-time end of the scale, the algorithm becomes information-theoretically optimal, capturing the natural brute-force search procedure for recovering $x$.
As the running time increases, the signal-strength $\lambda$ needed by the algorithm naturally decreases: this gives a nice "information-computation tradeoff".

Tune-able algorithms like this were already known for both sparse PCA and tensor PCA, so it is not entirely surprising that there is a common generalization.
(For the record, the algorithm here seems more along the lines of the tune-able algorithm for sparse PCA than the one for tensor PCA.)

(2) A computational lower bound which (almost) shows optimality of the algorithm above among so-called "low-degree" algorithms. This is a style of lower bound which has recently become popular and gives strong evidence that existing algorithmic techniques cannot improve very far on the algorithm given in this paper.

However, there is a small twist here compared to the analogous situations for sparse and tensor PCA (where similar lower bounds are also known): there *is* a gap between the lower bound given here and the algorithm *in the case of $p >> 1$*.
I don't view this as a significant failing of the paper: rather, it highlights the following interesting issue.
The natural way to prove such lower bounds for sparse and tensor PCA yields tight lower bounds -- this approach is to consider a certain "nice" prior on the signal vector $x$.
This same approach appears not to yield a tight lower bound for sparse tensor PCA, which leaves open the intriguing possibility of improved algorithms.

Overall, this paper mainly seems to adapt existing algorithmic and lower bound techniques to the sparse tensor PCA problem.
But it does so ably and comprehensively, and it was by no means obvious a priori what results one would get by applying these techniques in this setting.
It also highlights an interesting gap between the algorithms and lower bound which seem to be accessible with known techniques.
I therefore recommend acceptance.

(I did not carefully check any mathematics; however, the stated guarantees seem at a high level to be in line with what I would expect from the techniques outlined by the authors.)

**Time Spent Reviewing:**

2

---

> ### Author Response · Authors · 2021-08-10
> **Thanks for the feedback**
>
> We thank the anonymous reviewer for her/his valuable feedback. We will make use of her/his comments and improve the final version of the paper.

---

### Decision · Program_Chairs · 2021-09-27

**Decision:**

Accept (Poster)

**Comment:**

This paper consider the "Sparse Tensor PCA" problem,  an interesting generalisation  of the spike-matrix model. It is a theoretical paper focusing on computational complexity, and the main results (Theorem 1, 2) discussed the performance of a family of algorithms that smoothly interpolates between polynomial-time and the exponential-time exhaustive search algorithm.

Overall, there is a clear agreement on the reviewers side to accept the paper for publication at Neurips. The consensus is that this is a good and solid paper, in terms of results and clarity.  While the paper  adapt existing algorithmic and lower bound techniques to the sparse tensor PCA problem, it has been judged well written, and enjoyable to read. Some of the reviewers actually increased their score after the rebuttal, acknowledging that the authors successfully answered their comments.